# The Gyro-Structure of Some Matrix Manifolds

**Xuan Son Nguyen**
ETIS, UMR 8051, CY Cergy Paris Université, ENSEA, CNRS, Cergy, France
`xuan-son.nguyen@ensea.fr`

## Abstract

In this paper, we study the gyrovector space structure (*gyro-structure*) of matrix manifolds. Our work is motivated by the success of hyperbolic neural networks (HNNs) that have demonstrated impressive performance in a variety of applications. At the heart of HNNs is the theory of gyrovector spaces that provides a powerful tool for studying hyperbolic geometry. Here we focus on two matrix manifolds, i.e., Symmetric Positive Definite (SPD) and Grassmann manifolds, and consider connecting the Riemannian geometry of these manifolds with the basic operations, i.e., the binary operation and scalar multiplication on gyrovector spaces. Our work reveals some interesting facts about SPD and Grassmann manifolds. First, SPD matrices with the Affine-Invariant (AI) and Log-Euclidean (LE) geometries have rich structure with strong connection to hyperbolic geometry. Second, linear subspaces, when equipped with our proposed basic operations, form what we call *gyrocommutative and gyrononreductive gyrogroups*. Furthermore, they share remarkable analogies with gyrovector spaces. We demonstrate the applicability of our approach for human activity understanding and question answering.

## 1 Introduction

Data lying on matrix manifolds are commonly encountered in various applied areas such as medical imaging [3, 37], shape analysis [41], drone classification [6], image recognition [11], and human behavior analysis [10, 15, 16, 20, 21, 22, 31, 40]. These data arise from constraint sets of the problem, for which there is a natural representation of elements in the form of matrix arrays [2]. Due to the non-Euclidean nature of these data, traditional optimization algorithms usually fail to obtain good results in the matrix manifold setting. While a large body of works [6, 7, 8, 10, 20, 21, 22, 31, 33, 48] has been developed to generalize traditional optimization algorithms to this setting, there is still a lack of works that translate the language of differential geometry to basic operations on matrix manifolds so that they can be used in computational building blocks of neural network models on these manifolds just as basic operations on Euclidean spaces, e.g., matrix-matrix addition and scalar-matrix multiplication are used in deep neural networks (DNNs).

To address the above issue, we propose a novel framework based on the theory of gyrovector spaces [44, 45, 46] that has been successfully applied in the context of HNNs [12, 39]. Our aim is to uncover hidden analogies between the target manifolds and Euclidean spaces in the same way that we uncover hidden analogies between hyperbolic and Euclidean spaces [12, 39, 44, 45, 46]. Although there are some works [1, 17, 24, 25, 26, 27, 29] showing the gyro-structure of SPD manifolds with the AI geometry, none of them provides a rigorous mathematical formulation for the connection between the basic operations of [1, 17, 18, 24, 25, 26, 27] and the AI geometry of SPD manifolds. In this paper, we show how the basic operations can be constructed from the Riemannian geometry of matrix manifolds, and derive their compact expressions for SPD and Grassmann manifolds. In the case of SPD manifolds with the LE geometry [3], one recovers precisely the operations of [3] that give these manifolds a vector space structure. In the case of Grassmann manifolds, we obtain gyrocommutative and gyrononreductive gyrogroups that share remarkable analogies with gyrovector spaces. To the

36th Conference on Neural Information Processing Systems (NeurIPS 2022).

best of our knowledge, our work is the first that studies the structure of Grassmann manifolds under the framework of gyrovector spaces. Our main contributions are (1) We propose a method for constructing some basic operations, i.e., matrix-matrix addition and scalar-matrix multiplication on SPD and Grassmann manifolds; (2) We derive compact expressions of these operations for the considered manifolds; (3) We verify the gyro-structure of SPD manifolds, and some axioms of gyrovector spaces for Grassmann manifolds; (4) We showcase our approach on the tasks of human activity understanding and question answering.

## 2 Background

### 2.1 Gyrovector Spaces

Gyrovector spaces form the setting for hyperbolic geometry in the same way that vector spaces form the setting for Euclidean geometry [44, 45, 46]. We first recap the definitions of gyrogroups and gyrocommutative gyrogroups proposed in [44, 45, 46]. For greater mathematical detail and in-depth discussion, we refer the interested reader to these papers.

**Definition 2.1** (**Gyrogroups [46]**). *A pair $(G, \oplus)$ is a groupoid in the sense that it is a nonempty set, $G$, with a binary operation, $\oplus$. A groupoid $(G, \oplus)$ is a gyrogroup if its binary operation satisfies the following axioms for $a, b, c \in G$:*

*(G1) There is at least one element $e \in G$ called a left identity such that $e \oplus a = a$.*

*(G2) There is an element $\ominus a \in G$ called a left inverse of $a$ such that $\ominus a \oplus a = e$.*

*(G3) There is an automorphism $\mathrm{gyr}[a, b] : G \to G$ for each $a, b \in G$ such that*

$$a \oplus (b \oplus c) = (a \oplus b) \oplus \mathrm{gyr}[a, b]c \quad \textit{(Left Gyroassociative Law)}.$$

*The automorphism $\mathrm{gyr}[a, b]$ is called the gyroautomorphism, or the gyration of $G$ generated by $a, b$.*

*(G4) $\mathrm{gyr}[a, b] = \mathrm{gyr}[a \oplus b, b]$ (Left Reduction Property).*

**Definition 2.2** (**Gyrocommutative Gyrogroups [46]**). *A gyrogroup $(G, \oplus)$ is gyrocommutative if it satisfies*

$$a \oplus b = \mathrm{gyr}[a, b](b \oplus a) \quad \textit{(Gyrocommutative Law)}.$$

The following definition of gyrovector spaces is slightly different from Definition 3.2 in [46].

**Definition 2.3** (**Gyrovector Spaces**). *A gyrocommutative gyrogroup $(G, \oplus)$ equipped with a scalar multiplication*

$$(t, x) \to t \odot x : \mathbb{R} \times G \to G$$

*is called a gyrovector space if it satisfies the following axioms for $s, t \in \mathbb{R}$ and $a, b, c \in G$:*

*(V1) $1 \odot a = a, 0 \odot a = t \odot e = e$, and $(-1) \odot a = \ominus a$.*

*(V2) $(s + t) \odot a = s \odot a \oplus t \odot a$.*

*(V3) $(st) \odot a = s \odot (t \odot a)$.*

*(V4) $\mathrm{gyr}[a, b](t \odot c) = t \odot \mathrm{gyr}[a, b]c$.*

*(V5) $\mathrm{gyr}[s \odot a, t \odot a] = \mathrm{Id}$, where $\mathrm{Id}$ is the identity map.*

Note that the axioms of gyrovector spaces considered in our work are more strict than those proposed in [24, 25, 26, 27]. Thus many results proved in [24, 25, 26, 27] can be applied to our case, which gives rise to interesting applications.

## 3 Proposed Approach

For simplicity of exposition, we will concentrate on real matrices. Denote by $\mathrm{M}_{n,m}$ the space of $n \times m$ matrices, $\mathrm{Sym}_n^+$ the space of $n \times n$ SPD matrices, $\mathrm{Sym}_n$ the space of $n \times n$ symmetric matrices, $\mathrm{O}_n$ the space of $n \times n$ orthogonal matrices, $\mathrm{Gr}_{n,p}$ the $p$-dimensional subspaces of $\mathbb{R}^n$. Let $\mathcal{M}$ be a Riemannian homogeneous space, $T_{\mathbf{P}}\mathcal{M}$ be the tangent space of $\mathcal{M}$ at $\mathbf{P} \in \mathcal{M}$. Denote by $\exp(\mathbf{P})$ and $\log(\mathbf{P})$ the usual matrix exponential and logarithm of $\mathbf{P}$, $\mathrm{Exp}_{\mathbf{P}}(\mathbf{W})$ the exponential

map at $\mathbf{P}$ that associates to a tangent vector $\mathbf{W} \in T_{\mathbf{P}}\mathcal{M}$ a point of $\mathcal{M}$, $\mathrm{Log}_{\mathbf{P}}(\mathbf{Q})$ the logarithmic map of $\mathbf{Q} \in \mathcal{M}$ at $\mathbf{P}$. Let $\mathcal{T}_{\mathbf{P}\to\mathbf{Q}}(\mathbf{W})$ be the parallel transport of $\mathbf{W}$ from $\mathbf{P}$ to $\mathbf{Q}$ along geodesics connecting $\mathbf{P}$ and $\mathbf{Q}$. We will use superscripts for the exponential and logarithmic maps, and the parallel transport to indicate their associated Riemannian metric (in the case of SPD manifolds) or the target manifolds (in the case of Grassmann manifolds). The following definitions generalize those of [12] to the matrix manifold setting.

**Definition 3.1.** *Let $r$ be a positive real number. The binary operation $\mathbf{P} \oplus^r \mathbf{Q}$ where $\mathbf{P}, \mathbf{Q} \in \mathcal{M}$ is obtained by projecting $\mathbf{Q}^r$ in the tangent space at the identity element $\mathbf{I} \in \mathcal{M}$ with the logarithmic map, computing the parallel transport of this projection from $\mathbf{I}$ to $\mathbf{P}^r$ along geodesics connecting $\mathbf{I}$ and $\mathbf{P}^r$, and then projecting it back on the manifold with the exponential map, i.e.,*

$$\mathbf{P} \oplus^r \mathbf{Q} = \left( \mathrm{Exp}_{\mathbf{P}^r}(\mathcal{T}_{\mathbf{I}\to\mathbf{P}^r}(\mathrm{Log}_{\mathbf{I}}(\mathbf{Q}^r))) \right)^{\frac{1}{r}}. \tag{1}$$

**Definition 3.2.** *The scalar multiplication $t \otimes \mathbf{P}$ where $t \in \mathbb{R}$ and $\mathbf{P} \in \mathcal{M}$ is obtained by projecting $\mathbf{P}$ in the tangent space at the identity element $\mathbf{I} \in \mathcal{M}$ with the logarithmic map, multiplying this projection by the scalar $t$ in $T_{\mathbf{I}}\mathcal{M}$, and then projecting it back on the manifold with the exponential map, i.e.,*

$$t \otimes \mathbf{P} = \mathrm{Exp}_{\mathbf{I}}(t \, \mathrm{Log}_{\mathbf{I}}(\mathbf{P})). \tag{2}$$

In addition to the basic operations defined above, we need to determine an automorphism in order to verify the gyro-structure of a given matrix manifold. In the following, we will provide such automorphisms and derive compact expressions of the basic operations for SPD and Grassmann manifolds. The obtained expressions will ease the task of verifying the axioms of gyrovector spaces. Also, they often lead to simple and efficient implementations of neural networks on the considered manifolds. Furthermore, they enable effective generalizations of some operations on Euclidean spaces, e.g., matrix scaling to these manifolds (see Sections 4.1.2 and 4.2.2).

## 3.1 Gyrovector Spaces of SPD Matrices

We investigate the gyro-structure of SPD manifolds with the AI and LE geometries in Sections 3.1.1 and 3.1.2, respectively. The AI and LE frameworks are reviewed in the supplemental material. We refer the interested reader to that document for our theoretical results that reveal hidden analogies between SPD manifolds with the AI and LE geometries and Euclidean spaces.

### 3.1.1 AI Gyrovector Spaces

We first examine SPD manifolds with the AI geometry. Lemma 3.3 gives a compact expression of the binary operation (matrix-matrix addition).

**Lemma 3.3.** *For $\mathbf{P}, \mathbf{Q} \in \mathrm{Sym}_n^+$, the binary operation $\mathbf{P} \oplus_{ai}^r \mathbf{Q}$ is given as*

$$\mathbf{P} \oplus_{ai}^r \mathbf{Q} = (\mathbf{P}^{\frac{r}{2}} \mathbf{Q}^r \mathbf{P}^{\frac{r}{2}})^{\frac{1}{r}}. \tag{3}$$

**Proof** See the supplemental material.

An implicit assumption in Eq. (3) is that positive definite matrices are taken from the computations related to matrix powers. This assumption is used in all computations in Sections 3.1.1 and 3.1.2.

The identity element of $\mathrm{Sym}_n^+$ is the $n \times n$ identity matrix $\mathbf{I}_n$. Then, from Eq. (3), the inverse of $\mathbf{P}$ is given by

$$\ominus_{ai}^r \mathbf{P} = \mathbf{P}^{-1}.$$

**Lemma 3.4.** *For $\mathbf{P} \in \mathrm{Sym}_n^+$ and $t \in \mathbb{R}$, the scalar multiplication $t \otimes_{ai} \mathbf{P}$ is given as*

$$t \otimes_{ai} \mathbf{P} = \mathbf{P}^t. \tag{4}$$

**Proof** See the supplemental material.

**Definition 3.5** (AI Gyrovector Spaces). *Define a binary operation $\oplus_{ai}^r$ and a scalar multiplication $\otimes_{ai}$ by Eqs. (3) and (4), respectively. Define a gyroautomorphism generated by $\mathbf{P}$ and $\mathbf{Q}$ as*

$$\mathrm{gyr}_{ai}^r[\mathbf{P}, \mathbf{Q}]\mathbf{R} = \left( F_{ai}^r(\mathbf{P}, \mathbf{Q}) \mathbf{R}^r (F_{ai}^r(\mathbf{P}, \mathbf{Q}))^{-1} \right)^{\frac{1}{r}}, \tag{5}$$

*where $F_{ai}^r(\mathbf{P}, \mathbf{Q}) = (\mathbf{P}^{\frac{r}{2}} \mathbf{Q}^r \mathbf{P}^{\frac{r}{2}})^{-\frac{1}{2}} \mathbf{P}^{\frac{r}{2}} \mathbf{Q}^{\frac{r}{2}}$.*

**Theorem 3.6.** *Gyrogroups $(\mathrm{Sym}_n^+, \oplus_{ai}^r)$ with the scalar multiplication $\otimes_{ai}$ form gyrovector spaces $(\mathrm{Sym}_n^+, \oplus_{ai}^r, \otimes_{ai})$.*

**Proof** See the supplemental material.

The expressions of the binary operation, scalar multiplication, and gyroautomorphism given in Eqs. (3), (4), and (5) have appeared in [1, 17, 18]. However, these works concern with algebraic structures referred to as generalized gyrovector spaces where their considered set of axioms is different from the one in Section 2.1. The expressions given in Eq. (5) with $r = 1$ and $r = 2$ have also appeared in [24, 25, 26, 27]. However, no general form of the gyroautomorphism is given in these works.

### 3.1.2 LE Gyrovector Spaces

The LE metrics have proven to yield similar results as the AI metrics in practice, but with much simpler and faster computations [3]. This motivates us to study the gyro-structure of SPD manifolds with the LE geometry. As far as we know, this topic has not been investigated in previous works. The result given in the following lemma is not trivial as closed formulae do not exist for the parallel transport associated with the LE geometry.

**Lemma 3.7.** *For* $\mathbf{P}, \mathbf{Q} \in \mathrm{Sym}_n^+$, *the binary operation* $\mathbf{P} \oplus_{le}^r \mathbf{Q}$ *is given as*

$$\mathbf{P} \oplus_{le}^r \mathbf{Q} = \big( \exp(\log(\mathbf{P}^r) + \log(\mathbf{Q}^r)) \big)^{\frac{1}{r}}. \tag{6}$$

**Proof** See the supplemental material.

From Lemma 3.7, the inverse of $\mathbf{P}$ is given by

$$\ominus_{le}^r \mathbf{P} = \mathbf{P}^{-1}.$$

Similarly to the scalar multiplication $\otimes_{ai}$, the scalar multiplication $\otimes_{le}$ is constructed from Eq. (2). Lemma 3.8 gives an expression of this operation that is straightforward.

**Lemma 3.8.** *For* $\mathbf{P} \in \mathrm{Sym}_n^+$ *and* $t \in \mathbb{R}$, *the scalar multiplication* $t \otimes_{le} \mathbf{P}$ *is given by*

$$t \otimes_{le} \mathbf{P} = \mathbf{P}^t. \tag{7}$$

**Definition 3.9** (**LE Gyrovector Spaces**). *Define a binary operation* $\oplus_{le}^r$ *and a scalar multiplication* $\otimes_{le}$ *by Eqs. (6) and (7), respectively. Define a gyroautomorphism generated by* $\mathbf{P}$ *and* $\mathbf{Q}$ *as*

$$\mathrm{gyr}_{le}^r[\mathbf{P}, \mathbf{Q}] = \mathrm{Id}.$$

**Theorem 3.10.** *Gyrogroups* $(\mathrm{Sym}_n^+, \oplus_{le}^r)$ *with the scalar multiplication* $\otimes_{le}$ *form gyrovector spaces* $(\mathrm{Sym}_n^+, \oplus_{le}^r, \otimes_{le})$.

**Proof** See the supplemental material.

The conclusion of Theorem 3.10 agrees with [3] which shows that the space of SPD matrices with the LE geometry has a vector space structure. This vector space structure is given by the operations defined in [3] that turn out to be the binary operation and scalar multiplication on LE gyrovector spaces in the specific case where $r = 1$. Indeed, it can be shown that the mapping $\exp : (\mathrm{Sym}_n, +, .) \rightarrow (\mathrm{Sym}_n^+, \oplus_{le}^r, \otimes_{le})$ is a vector space isomorphism. Thus, for any $r$, the operations defined above also turn the space of SPD matrices with the LE geometry into a vector space. This generalizes the result of [3].

### 3.2 Grassmann Gyrocommutative and Gyrononreductive Gyrogroups

In the previous sections, we concern with the non-positively curved spaces of SPD matrices. In this section, we focus on another family of matrix manifolds, i.e., the non-negatively curved Grassmann manifolds. We adopt the following definition for Grassmann manifolds:

$$\mathrm{Gr}_{n,p} = \{\mathbf{P} \in \mathrm{M}_{n \times n} \,|\, \mathbf{P}^T = \mathbf{P}, \mathbf{P}^2 = \mathbf{P}, \mathrm{rank}(\mathbf{P}) = p\}.$$

The Riemannian geometry of Grassmann manifolds is reviewed in the supplemental material.

Let $\mathbf{I}_{n,p} = \begin{bmatrix} \mathbf{I}_p & 0 \\ 0 & 0 \end{bmatrix}$ be the identity element of $\mathrm{Gr}_{n,p}$. For the sake of convenience, we denote $\bar{\mathbf{P}} = \mathrm{Log}_{\mathbf{I}_{n,p}}^{gr}(\mathbf{P})$. For $\mathbf{P}, \mathbf{Q} \in \mathrm{Gr}_{n,p}$, assuming that $\mathbf{I}_{n,p}$ and $\mathbf{P}$ are not in each other's cut locus (see

the supplemental material for a definition of cut locus of Grassmann manifolds), and $\mathbf{I}_{n,p}$ and $\mathbf{Q}$ are not in each other's cut locus, the binary operation is defined as

$$\mathbf{P} \oplus_{gr} \mathbf{Q} = \mathrm{Exp}_{\mathbf{P}}^{gr}(\mathcal{T}_{\mathbf{I}_{n,p} \to \mathbf{P}}(\mathrm{Log}_{\mathbf{I}_{n,p}}^{gr}(\mathbf{Q}))).$$

**Lemma 3.11.** *For* $\mathbf{P}, \mathbf{Q} \in \mathrm{Gr}_{n,p}$*, the binary operation* $\mathbf{P} \oplus_{gr} \mathbf{Q}$ *is given as*

$$\mathbf{P} \oplus_{gr} \mathbf{Q} = \exp([\overline{\mathbf{P}}, \mathbf{I}_{n,p}])\mathbf{Q}\exp(-[\overline{\mathbf{P}}, \mathbf{I}_{n,p}]), \tag{8}$$

*where* $[.,.]$ *denotes the matrix commutator.*

**Proof**  See the supplemental material.

Lemma 3.12 gives a closed-form expression of the binary operation in terms of $\mathbf{P}$ and $\mathbf{Q}$.

**Lemma 3.12.** *For* $\mathbf{P}, \mathbf{Q} \in \mathrm{Gr}_{n,p}$*, the binary operation* $\mathbf{P} \oplus_{gr} \mathbf{Q}$ *is given as*

$$\mathbf{P} \oplus_{gr} \mathbf{Q} = \exp\left(\frac{1}{2}\log\left((\mathbf{I}_n - 2\mathbf{P})(\mathbf{I}_n - 2\mathbf{I}_{n,p})\right)\right)\mathbf{Q}\exp\left(-\frac{1}{2}\log\left((\mathbf{I}_n - 2\mathbf{P})(\mathbf{I}_n - 2\mathbf{I}_{n,p})\right)\right).$$

**Proof**  See the supplemental material.

For $\mathbf{P} \in \mathrm{Gr}_{n,p}$ such that $\mathbf{I}_{n,p}$ and $\mathbf{P}$ are not in each other's cut locus, the inverse $\ominus_{gr}\mathbf{P}$ of $\mathbf{P}$ is defined as

$$\ominus_{gr}\mathbf{P} = \mathrm{Exp}_{\mathbf{I}_{n,p}}^{gr}(-\mathrm{Log}_{\mathbf{I}_{n,p}}^{gr}(\mathbf{P})).$$

The scalar multiplication is defined as in Eq. (2) (subject to the assumption stated above).

**Lemma 3.13.** *For* $\mathbf{P} \in \mathrm{Gr}_{n,p}$ *and* $t \in \mathbb{R}$*, the scalar multiplication* $t \otimes_{gr} \mathbf{P}$ *is given by*

$$t \otimes_{gr} \mathbf{P} = \exp([t\overline{\mathbf{P}}, \mathbf{I}_{n,p}])\mathbf{I}_{n,p}\exp(-[t\overline{\mathbf{P}}, \mathbf{I}_{n,p}]). \tag{9}$$

**Proof**  See the supplemental material.

Lemma 3.14 gives a closed-form expression of the scalar multiplication in terms of $t$ and $\mathbf{P}$.

**Lemma 3.14.** *For* $\mathbf{P} \in \mathrm{Gr}_{n,p}$ *and* $t \in \mathbb{R}$*, the scalar multiplication* $t \otimes_{gr} \mathbf{P}$ *is given by*

$$t \otimes_{gr} \mathbf{P} = \exp\left(\frac{t}{2}\log\left((\mathbf{I}_n - 2\mathbf{P})(\mathbf{I}_n - 2\mathbf{I}_{n,p})\right)\right)\mathbf{I}_{n,p}\exp\left(-\frac{t}{2}\log\left((\mathbf{I}_n - 2\mathbf{P})(\mathbf{I}_n - 2\mathbf{I}_{n,p})\right)\right).$$

**Proof**  See the supplemental material.

Grassmann manifolds, when equipped with the above operations, do not form gyrovector spaces. However, as we will show later, they still form spaces that verify most of the axioms of gyrovector spaces. Note that since these operations can only be defined under the assumptions stated at the beginning of this section, all the axioms in Section 2.1 must be considered under these assumptions. We thus make them implicitly in the following. In order to specify the structure of Grassmann manifolds, we need to define some new algebraic structures that are generalizations of groups (Definitions 3.15 and 3.16) and vector spaces (Definition 3.17).

**Definition 3.15** (**Nonreductive Gyrogroups**). *A groupoid* $(G, \oplus)$ *is a nonreductive gyrogroup if its binary operation satisfies axioms (G1), (G2), and (G3).*

**Definition 3.16** (**Gyrocommutative and Gyrononreductive Gyrogroups**). *A nonreductive gyrogroup* $(G, \oplus)$ *is gyrocommutative and gyrononreductive if it satisfies the Gyrocommutative Law.*

**Definition 3.17** (**Nonreductive Gyrovector Spaces**). *A gyrocommutative and gyrononreductive gyrogroup* $(G, \oplus)$ *equipped with a scalar multiplication* $\odot$ *is called a nonreductive gyrovector space if it satisfies axioms (V1), (V2), (V3), (V4), and (V5).*

Definition 3.18 gives the expression of a gyroautomorphism in Grassmann manifolds.

**Definition 3.18** (**Grassmann Gyrocommutative and Gyrononreductive Gyrogroups**). *Define a binary operation* $\oplus_{gr}$ *and a scalar multiplication* $\otimes_{gr}$ *by Eqs. (8) and (9), respectively. For* $\mathbf{P}, \mathbf{Q} \in \mathrm{Gr}_{n,p}$*, assuming that* $\mathbf{I}_{n,p}$ *and* $\mathbf{P}$ *are not in each other's cut locus,* $\mathbf{I}_{n,p}$ *and* $\mathbf{Q}$ *are not in each other's cut locus,* $\mathbf{I}_{n,p}$ *and* $\mathbf{P} \oplus_{gr} \mathbf{Q}$ *are not in each other's cut locus, then a gyroautomorphism generated by* $\mathbf{P}$ *and* $\mathbf{Q}$ *can be defined as*

$$\mathrm{gyr}_{gr}[\mathbf{P}, \mathbf{Q}]\mathbf{R} = F_{gr}(\mathbf{P}, \mathbf{Q})\mathbf{R}(F_{gr}(\mathbf{P}, \mathbf{Q}))^{-1}, \tag{10}$$

*where* $F_{gr}(\mathbf{P}, \mathbf{Q})$ *is given by*

$$F_{gr}(\mathbf{P}, \mathbf{Q}) = \exp(-[\overline{\mathbf{P} \oplus_{gr} \mathbf{Q}}, \mathbf{I}_{n,p}])\exp([\overline{\mathbf{P}}, \mathbf{I}_{n,p}])\exp([\overline{\mathbf{Q}}, \mathbf{I}_{n,p}]). \tag{11}$$

The following identities are required for the verification of some axioms of nonreductive gyrovector spaces for Grassmann manifolds subject to certain conditions (Theorem 3.20). These identities are new to the best of our knowledge.

**Lemma 3.19.** *For* $\mathbf{P}, \mathbf{F} \in \mathrm{Gr}_{n,p}$, $\mathbf{O} \in \mathrm{O}_n$, $\Delta \in T_{\mathbf{P}} \mathrm{Gr}_{n,p}$, $m \in \mathbb{N}$, *and* $s \in \mathbb{R}$, *the following identities hold:*

*1.* $\mathrm{Log}^{gr}_{\mathbf{OFO}^T}(\mathbf{OPO}^T) = \mathbf{O} \, \mathrm{Log}^{gr}_{\mathbf{F}}(\mathbf{P})\mathbf{O}^T;$

*2.* $\mathrm{Exp}^{gr}_{\mathbf{OPO}^T}(\mathbf{O}\Delta\mathbf{O}^T) = \mathbf{O} \, \mathrm{Exp}^{gr}_{\mathbf{P}}(\Delta)\mathbf{O}^T;$

*3.* $[\Delta, \mathbf{P}]^m = (\mathbf{I}_n - 2\mathbf{P})(-[\Delta, \mathbf{P}])^m(\mathbf{I}_n - 2\mathbf{P});$

*4.* $\exp(s[\Delta, \mathbf{P}]) = (\mathbf{I}_n - 2\mathbf{P})\exp(-s[\Delta, \mathbf{P}])(\mathbf{I}_n - 2\mathbf{P}).$

**Proof**    See the supplemental material.

We are now ready to state the main results of this section.

**Theorem 3.20.** *Groupoids* $(\mathrm{Gr}_{n,p}, \oplus_{gr})$ *form gyrocommutative and gyrononreductive gyrogroups. Furthermore, gyrocommutative and gyrononreductive gyrogroups* $(\mathrm{Gr}_{n,p}, \oplus_{gr})$ *with the scalar multiplication* $\otimes_{gr}$ *satisfy axioms (V1) and (V4), and axioms (V2), (V3), and (V5) under the following conditions:*

*(V2)* $(s + t) \otimes_{gr} \mathbf{P} = s \otimes_{gr} \mathbf{P} \oplus_{gr} t \otimes_{gr} \mathbf{P}$ *for* $\mathbf{P} \in \mathrm{Gr}_{n,p}$, $t \in \mathbb{R}$, *and* $s \in \mathbb{R}$ *such that:*

$$|s| < \min_{\lambda_i} \left\{ \frac{\pi}{2|\mathrm{Im}(\lambda_i)|} \right\},$$

*where* $\lambda_i$ *is an eigenvalue of* $[\overline{\mathbf{P}}, \mathbf{I}_{n,p}]$, *and* $\mathrm{Im}(\lambda_i)$ *is the imaginary part of* $\lambda_i$.

*(V3)* $(st) \otimes_{gr} \mathbf{P} = s \otimes_{gr} (t \otimes_{gr} \mathbf{P})$ *for* $\mathbf{P} \in \mathrm{Gr}_{n,p}$, $s \in \mathbb{R}$, *and* $t \in \mathbb{R}$ *such that:*

$$|t| < \min_{\lambda_i} \left\{ \frac{\pi}{2|\mathrm{Im}(\lambda_i)|} \right\}.$$

*(V5)* $\mathrm{gyr}_{gr}[s \otimes_{gr} \mathbf{P}, t \otimes_{gr} \mathbf{P}] = \mathrm{Id}$ *for* $\mathbf{P} \in \mathrm{Gr}_{n,p}$, *and* $s, t \in \mathbb{R}$ *such that:*

$$\max\{|s|, |t|, |s + t|\} < \min_{\lambda_i} \left\{ \frac{\pi}{2|\mathrm{Im}(\lambda_i)|} \right\}.$$

**Proof**    See the supplemental material.

The following corollaries are useful in practice for the verification of axioms (V2), (V3), and (V5).

**Corollary 3.21.** *Gyrocommutative and gyrononreductive gyrogroups* $(\mathrm{Gr}_{n,p}, \oplus_{gr})$ *with the scalar multiplication* $\otimes_{gr}$ *satisfy axioms (V2), (V3), and (V5) if the right-hand sides of the inequalities stated in Theorem 3.20 are replaced with*

$$\frac{\pi}{2\|[\overline{\mathbf{P}}, \mathbf{I}_{n,p}]\|},$$

*where* $\|.\|$ *denotes the Hilbert-Schmidt norm.*

**Proof**    See the supplemental material.

**Corollary 3.22.** *Gyrocommutative and gyrononreductive gyrogroups* $(\mathrm{Gr}_{n,p}, \oplus_{gr})$ *with the scalar multiplication* $\otimes_{gr}$ *satisfy axioms (V2), (V3), and (V5) under the following conditions: (V2)* $|s| \leq 1$*; (V3)* $|t| \leq 1$*; (V5)* $\max\{|s|, |t|, |s + t|\} \leq 1$.

**Proof**    See the supplemental material.

It follows from Corollary 3.22 that if $\max\{|s|, |t|, |s + t|\} \leq 1$, then Grassmann manifolds, when equipped with the basic operations defined in Eqs. (8) and (9), verify all the axioms of nonreductive gyrovector spaces.

# 4    Applications

To showcase our approach, we propose new methods for human activity understanding and question answering. We refer the interested reader to the supplemental material for more applications.

## 4.1 Human Activity Understanding

In this section, we develop a class of RNNs on SPD manifolds for human activity understanding. It is worth mentioning that the operations defined in Section 3.1 as well as those constructed in Section 4.1.2 can be used to build any type of neural networks on SPD manifolds, e.g., convolutional neural networks. However, since RNNs are based on update equations that involve the basic operations on matrices, they are well suited for validating our approach.

### 4.1.1 Problem Formulation

Human activities can be recognized from low/mid level features [35, 36] or high-level poses [23, 32, 33, 34]. We use 3D skeleton data as they have shown to outperform low/mid level features for the considered task [23]. The goal is to build a model that for each sequence of 3D positions of body (hand) joints identifies the action performed by the person (or group of persons) in the sequence.

### 4.1.2 Proposed Method

We will make use of the basic operations on AI and LE gyrovector spaces and the concept of gyroderivative in these spaces (see the supplemental material) that is similar to hyperbolic derivative [4] in Möbius gyrovector spaces. We also need to generalize some operations of Euclidean RNNs to the SPD manifold setting. Here we focus on 2 operations, i.e., matrix scaling and pointwise nonlinearity.

**Matrix Scaling** If $\mathbf{P} \in \mathrm{Sym}_n^+, \mathbf{W} \in \mathbb{R}^n, \mathbf{W} > 0$, then the matrix scaling $\mathbf{W} \otimes_{spd}^v \mathbf{P}$ is given by

$$\mathbf{W} \otimes_{spd}^v \mathbf{P} = \mathbf{U} \operatorname{diag}(\mathbf{W} * \mathbf{V}) \mathbf{U}^T,$$

where $\mathbf{U} \operatorname{diag}(\mathbf{V}) \mathbf{U}^T$ is the eigenvalue decomposition of $\mathbf{P}$, and $\mathbf{W} * \mathbf{V}$ is the element-wise multiplication.

**Pointwise nonlinearity** If $\varphi$ is a pointwise nonlinear activation function, then the pointwise nonlinearity $\varphi^{\otimes_a}(\mathbf{P})$ is given by

$$\varphi^{\otimes_a}(\mathbf{P}) = \mathbf{U} \operatorname{diag}(\max(\epsilon \mathbf{I}, \varphi(\mathbf{V}))) \mathbf{U}^T,$$

where $\epsilon > 0$ is a rectification threshold, and $\mathbf{U} \operatorname{diag}(\mathbf{V}) \mathbf{U}^T$ is the eigenvalue decomposition of $\mathbf{P}$.

By adapting a class of models that are invariant to time rescaling [42] to the SPD manifold setting (see the supplemental material), we obtain the following update equations for our models:

$$\mathbf{P}_t = \varphi^{\otimes_a}(\mathbf{W}_h \otimes_{spd}^v \mathbf{H}_{t-1} + \mathbf{W}_x \otimes_{spd}^v \mathbf{X}_t), \tag{12}$$

$$\mathbf{H}_t = \mathbf{H}_{t-1} \oplus \alpha \odot ((\ominus \mathbf{H}_{t-1}) \oplus \mathbf{P}_t), \tag{13}$$

where $\mathbf{X}_t \in \mathrm{Sym}_n^+$ is the input at frame $t$, $\mathbf{H}_{t-1}, \mathbf{H}_t \in \mathrm{Sym}_n^+$ are the hidden states at frames $t-1$ and $t$, respectively, $\mathbf{W}_h, \mathbf{W}_x \in \mathbb{R}^n$, and $\alpha \in \mathbb{R}$ is a learnable parameter.

### 4.1.3 Implementation Details

In order to retain the correlation of neighboring joints [5, 49] and to increase feature interactions encoded by covariance matrices, we first identify a closest left (right) neighbor of every joint based on their distance to the hip (wrist) joint[1], and then combine the 3D coordinates of each joint and those of its left (right) neighbor to create a feature vector for the joint. For a given frame $t$, a mean vector $\boldsymbol{\mu}_t$ and a covariance matrix $\boldsymbol{\Sigma}_t$ are computed from the set of feature vectors of the frame and then combined [30] to create a SPD matrix as

$$\mathbf{Y}_t = \begin{bmatrix} \boldsymbol{\Sigma}_t + \boldsymbol{\mu}_t(\boldsymbol{\mu}_t)^T & \boldsymbol{\mu}_t \\ (\boldsymbol{\mu}_t)^T & 1 \end{bmatrix}.$$

The lower part of matrix $\log(\mathbf{Y}_t)$ is flattened to obtain a vector $\tilde{\mathbf{v}}_t$. All vectors $\tilde{\mathbf{v}}_t$ within a time window $[t, t + c - 1]$ where $c$ is a constant are used to compute a covariance matrix as $\mathbf{Z}_t = \frac{1}{c} \sum_{i=t}^{t+c-1} (\tilde{\mathbf{v}}_i - \bar{\mathbf{v}}_t)(\tilde{\mathbf{v}}_i - \bar{\mathbf{v}}_t)^T$, where $\bar{\mathbf{v}}_t = \frac{1}{c} \sum_{i=t}^{t+c-1} \tilde{\mathbf{v}}_i$. Matrix $\mathbf{Z}_t$ is then the input data at frame $t$ of the networks. Our network GyroAI-HAUNet is illustrated in Fig. 1a. For classification, the network output is projected to the tangent space at the identity matrix using the logarithmic map. The lower part of the resulting matrix is flattened and then is fed to a fully-connected layer.

---

[1]For joints having more than 2 neighbors, one of them can be chosen.

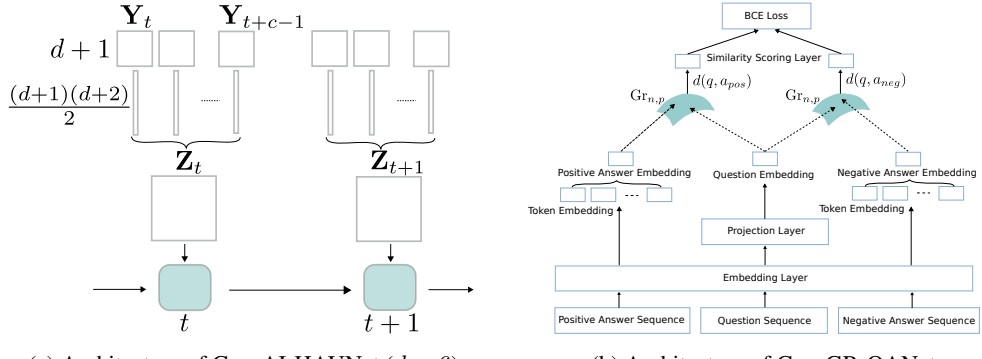

(a) Architecture of GyroAI-HAUNet ($d = 6$)      (b) Architecture of GyroGR-QANet

Figure 1: Our proposed architectures for human action understanding and question answering.

Table 1: Accuracy comparison (%) of our networks against state-of-the-art SPD neural networks.

| Dataset | SPDNet | | SPDNetBN | | SPD-SRU | | GyroAI-HAUNet | | GyroLE-HAUNet | |
|---------|--------|--------|----------|--------|---------|--------|---------------|--------|---------------|--------|
| | $M = 1$ | $M = 3$ | $M = 1$ | $M = 3$ | $M = 1$ | $M = 3$ | $M = 1$ | $M = 3$ | $M = 1$ | $M = 3$ |
| HDM05 | 58.44 | 72.75 | 62.54 | 76.25 | 42.26 | 54.79 | 61.50 | 78.14 | 57.01 | 74.53 |
| #HDM05 | 0.12 MB | 0.77 MB | 0.13 MB | 0.82 MB | 0.05 MB | 0.32 MB | 0.05 MB | 0.31 MB | 0.05 MB | 0.31 MB |
| FPHA | 87.65 | 88.17 | 88.52 | 91.83 | 78.57 | 85.16 | 89.73 | 96.00 | 83.03 | 89.94 |
| #FPHA | 0.04 MB | 0.28 MB | 0.05 MB | 0.31 MB | 0.019 MB | 0.114 MB | 0.018 MB | 0.110 MB | 0.018 MB | 0.110 MB |
| NTU60 | 73.26 | 77.38 | 75.84 | 79.52 | 66.25 | 75.34 | 83.12 | 94.72 | 77.25 | 89.44 |
| #NTU60 | 0.03 MB | 0.20 MB | 0.04 MB | 0.28 MB | 0.004 MB | 0.027 MB | 0.004 MB | 0.026 MB | 0.004 MB | 0.026 MB |

### 4.1.4 Results

We use three datasets, i.e., HDM05 [16], FPHA [13], and NTU RGB+D 60 (NTU60) [38]. These datasets include three different types of activities: body actions (HDM05), hand actions (FPHA), and interaction actions (NTU60). In the same spirit of previous works [6, 20, 40], we are interested in comparing manifold networks with a focus on SPD manifolds, and do not necessarily seek state-of-the-art performance for the target task. We use a temporal pyramid representation for each sequence. At temporal pyramid $M$, a given sequence is partitioned into $M$ subsequences of equal size. Each subsequence is then fed to a model with its own parameter set. The outputs from all the models are concatenated to create a final representation of the sequence. We run each model three times and report the best accuracy from these three runs [12]. The experimental settings can be found in the supplemental material. This document also reports the mean accuracies and standard deviations of some representative methods, an ablation study, and a comparison of our networks against Euclidean RNNs, transformers, HNNs, and graph neural networks (GNNs).

**Comparison against SPD Neural Networks**     Our networks, referred to as GyroAI-HAUNet and GyroLE-HAUNet, are compared against SPDNet [20], SPDNetBN [6], and SPD-SRU [8]. Results of these networks are obtained using their official code[2,3,4] with default parameter settings. Results for $M = 1$ and $M = 3$[5] are given in Tab. 1. On HDM05 dataset, GyroAI-HAUNet outperforms SPDNet and SPD-SRU, and performs worse than SPDNetBN when $M = 1$. However, when $M = 3$, GyroAI-HAUNet gives the best result among the competing networks. On FPHA and NTU60 datasets, GyroAI-HAUNet gives the best results for both $M = 1$ and $M = 3$. In terms of model size, GyroAI-HAUNet and GyroLE-HAUNet require far fewer parameters than SPDNet and SPDNetBN in all cases. For example, on FPHA dataset, when $M = 3$, GyroAI-HAUNet outperforms SPDNetBN by 4.17% with 2.8 times fewer parameters. Also, on NTU60 dataset, when $M = 3$, GyroAI-HAUNet outperforms SPDNetBN by 15.20% with 10.7 times fewer parameters.

---

[2]https://github.com/zhiwu-huang/SPDNet
[3]https://papers.nips.cc/paper/2019/hash/6e69ebbfad976d4637bb4b39de261bf7-Abstract.html
[4]https://github.com/zhenxingjian/SPD-SRU/tree/master
[5]For all the networks, setting $M > 3$ did not yield better results on the three datasets.

## 4.2 Question Answering

In this section, we consider learning word embeddings in $\mathrm{Gr}_{n,p}$. We also investigate the use of product manifolds $\mathrm{Gr}_{n_1,p} \times \mathrm{Sym}_{n_2}^+$ for training word embeddings. This idea is mainly motivated by the work of [14] that shows the efficacy of mixed-curvature representations for graph embeddings.

### 4.2.1 Problem Formulation

Let $\mathcal{Q}$ be a list of questions, $\mathcal{A}$ be a list of answers to the questions in $\mathcal{Q}$. Each question $q \in \mathcal{Q}$ has a list of candidate answers in $\mathcal{A}$. The candidate set comes with their relevancy judgements, where answers that are correct (positive) have labels equal to 1 and 0 otherwise. The goal is to build a model that for each query $q$ and its list of candidate answers generates an optimal ranking such that correct answers appear at top of the list [29, 43].

### 4.2.2 Proposed Method

The core idea is to learn a scoring function [29, 43] given as

$$\phi_{qa}(q, a) = -w_f d(\mathbf{Q}, \mathbf{A}) + w_b,$$

where $\mathbf{Q}$ and $\mathbf{A}$ are embeddings of question $q$ and answer $a$, respectively, $w_f, w_b \in \mathbb{R}$ are parameters of the model, and $d(.)$ is a distance function in the target manifolds. We define a matrix scaling operation as follows.

**Matrix Scaling** We parameterize a point $\mathbf{P} \in \mathrm{Gr}_{n,p}$ by a matrix $\mathbf{B} \in \mathrm{M}_{p,n-p}$ such that

$$\begin{bmatrix} 0 & \mathbf{B} \\ -\mathbf{B}^T & 0 \end{bmatrix} = [\bar{\mathbf{P}}, \mathbf{I}_{n,p}].$$

Then point $\mathbf{P}$ can be computed (see Eq. (8) and the supplemental material) by

$$\mathbf{P} = \exp([\bar{\mathbf{P}}, \mathbf{I}_{n,p}])\mathbf{I}_{n,p}\exp(-[\bar{\mathbf{P}}, \mathbf{I}_{n,p}]) = \exp\left(\begin{bmatrix} 0 & \mathbf{B} \\ -\mathbf{B}^T & 0 \end{bmatrix}\right)\mathbf{I}_{n,p}\exp\left(-\begin{bmatrix} 0 & \mathbf{B} \\ -\mathbf{B}^T & 0 \end{bmatrix}\right).$$

The matrix scaling $\otimes_{gr}^m$ is defined as

$$\mathbf{A} \otimes_{gr}^m \mathbf{P} = \exp\left(\begin{bmatrix} 0 & \mathbf{A} * \mathbf{B} \\ -(\mathbf{A} * \mathbf{B})^T & 0 \end{bmatrix}\right)\mathbf{I}_{n,p}\exp\left(-\begin{bmatrix} 0 & \mathbf{A} * \mathbf{B} \\ -(\mathbf{A} * \mathbf{B})^T & 0 \end{bmatrix}\right),$$

where $\mathbf{A} \in \mathrm{M}_{p,n-p}$. The embedding of question $q$ is computed from those of its tokens as

$$\mathbf{Q} = \mathbf{B} \oplus_{gr} \left((\mathbf{S} \otimes_{gr}^m \mathbf{T}_1) \oplus_{gr} (\mathbf{S} \otimes_{gr}^m \mathbf{T}_2) \dots \oplus_{gr} (\mathbf{S} \otimes_{gr}^m \mathbf{T}_l)\right),$$

where $\mathbf{T}_i \in \mathrm{Gr}_{n,p}, i = 1, \dots, l$ are embeddings of the tokens in question $q$, $\mathbf{S} \in \mathrm{M}_{p,n-p}$ and $\mathbf{B} \in \mathrm{Gr}_{n,p}$ are parameters of the model. The embedding of answer $a$ is the summation of those of its tokens using operation $\oplus_{gr}$. For $\mathbf{P}, \mathbf{Q} \in \mathrm{Gr}_{n,p}$, the distance function $d(.,.)$ is defined as

$$d(\mathbf{P}, \mathbf{Q}) = d_{gr}(\mathbf{P}, \mathbf{Q}) = \|\mathbf{P} - \mathbf{Q}\|_F.$$

We also train word embeddings in product manifold $\mathrm{Gr}_{n_1,p} \times \mathrm{Sym}_{n_2}^+$ using the distance function

$$d((\mathbf{P}_{gr}, \mathbf{P}_{spd}), (\mathbf{Q}_{gr}, \mathbf{Q}_{spd})) = d_{gr}(\mathbf{P}_{gr}, \mathbf{Q}_{gr}) + \tau d_{spd}^{F_1}(\mathbf{P}_{spd}, \mathbf{Q}_{spd}),$$

where $\mathbf{P}_{gr}, \mathbf{Q}_{gr} \in \mathrm{Gr}_{n_1,p}$, $\mathbf{P}_{spd}, \mathbf{Q}_{spd} \in \mathrm{Sym}_{n_2}^+$, $d_{spd}^{F_1}(.,.)$ is the Finsler distance function proposed in [29], and $\tau$ is a constant. This results in three models based on the scaling, rotation, and reflection transformations of [29]. Our Grassmann model is shown in Fig. 1b.

### 4.2.3 Results

We use two datasets, i.e., TrecQA [47] (clean version) and WikiQA [50]. The mean average precision (MAP) and mean reciprocal rank (MRR) are used as evaluation metrics [43]. We compare our models[6] against the SPD models of [29]. Since the codes of these models are not publicly available,

---

[6]Code available at https://github.com/spratmnt/qa.

Table 2: Comparison of our models against the SPD models of [29]. The SPD models learn embeddings in $\mathrm{Sym}_{14}^+$. Our Grassmann model learns embeddings in $\mathrm{Gr}_{14,7}$. Our models based on the product manifold learn embeddings in $\mathrm{Gr}_{14,7} \times \mathrm{Sym}_8^+$. Results are computed over three runs.

| DOF | Model | TrecQA | | WikiQA | | Time (TrecQA,seconds) | |
|---|---|---|---|---|---|---|---|
| | | MAP | MRR | MAP | MRR | Train/epoch | Test |
| 105 | $\mathrm{SPD}_{\mathrm{Sca}}^R$ | $47.77 \pm 0.18$ | $57.54 \pm 0.32$ | $59.42 \pm 0.04$ | $60.57 \pm 0.05$ | 15.35 | 1.41 |
| 105 | $\mathrm{SPD}_{\mathrm{Sca}}^{F_1}$ | $48.35 \pm 1.24$ | $57.64 \pm 2.23$ | $60.68 \pm 0.42$ | $62.01 \pm 0.34$ | 15.56 | 1.43 |
| 105 | $\mathrm{SPD}_{\mathrm{Rot}}^R$ | $48.07 \pm 0.89$ | $54.49 \pm 0.87$ | $59.37 \pm 1.25$ | $60.74 \pm 1.50$ | 15.98 | 1.41 |
| 105 | $\mathrm{SPD}_{\mathrm{Rot}}^{F_1}$ | $49.02 \pm 1.62$ | $58.72 \pm 1.85$ | $60.60 \pm 0.67$ | $62.28 \pm 0.74$ | 16.09 | 1.43 |
| 105 | $\mathrm{SPD}_{\mathrm{Ref}}^R$ | $48.79 \pm 0.91$ | $57.12 \pm 1.23$ | $59.07 \pm 0.53$ | $60.58 \pm 0.52$ | 15.98 | 1.41 |
| 105 | $\mathrm{SPD}_{\mathrm{Ref}}^{F_1}$ | $48.33 \pm 0.48$ | $56.18 \pm 0.97$ | $59.93 \pm 0.18$ | $61.80 \pm 0.39$ | 16.09 | 1.43 |
| 49 | GyroGR-QANet | $50.18 \pm 1.29$ | $58.19 \pm 2.59$ | $56.69 \pm 1.45$ | $58.26 \pm 1.45$ | 3.69 | 0.25 |
| 85 | GyroGR-$\mathrm{SPD}_{\mathrm{Sca}}^{F_1}$-QANet | $50.10 \pm 0.30$ | $57.70 \pm 0.93$ | $60.62 \pm 0.25$ | $62.42 \pm 0.16$ | 11.62 | 0.90 |
| 85 | GyroGR-$\mathrm{SPD}_{\mathrm{Rot}}^{F_1}$-QANet | $50.27 \pm 0.56$ | $58.62 \pm 1.35$ | $59.78 \pm 0.15$ | $61.66 \pm 0.23$ | 11.72 | 0.90 |
| 85 | GyroGR-$\mathrm{SPD}_{\mathrm{Ref}}^{F_1}$-QANet | $48.83 \pm 1.89$ | $58.11 \pm 0.87$ | $60.41 \pm 0.39$ | $61.86 \pm 0.35$ | 11.72 | 0.90 |

we implemented them by following closely the instructions in [29]. The implementation details and experimental settings can be found in the supplemental material.

Tab. 2 reports the means and standard deviations of MAP and MRR from three runs (DOF stands for degrees of freedom). Our Grassmann model GyroGR-QANet performs favorably against most of the SPD models on TrecQA dataset. Also, results of our models based on the product manifold show the efficacy of mixed-curvature representations [14] in question answering. Note that the numbers of DOF of our models are smaller than those of the SPD models. We also note that techniques in [19, 28] could potentially improve the performance of our models.

## 5   Limitations of Our Work

To develop our SPD models, we only construct the basic operations and two other operations, i.e., matrix scaling and pointwise nonlinearity (see Section 4.1.1). Other operations [39] should also be designed in order to improve the representation power of our networks. Also, the question of how to generalize a broader class of DNNs to the SPD manifold setting should be addressed in future work.

Our experiments for question answering have shown the usefulness of product manifolds $\mathrm{Gr}_{n_1,p} \times \mathrm{Sym}_{n_2}^+$ for learning word, entity, and relation embeddings. More investigation is needed to see if a product of multiple smaller-dimension SPD and Grassmann manifolds will improve performance [14]. It would also be interesting to find out patterns that show the relationship between the performance of the embeddings and the theoretical curvature of the manifolds in our problems [9].

## 6   Conclusion

We have shown that the AI and LE geometries of SPD manifolds have strong connection to hyperbolic geometry, and Grassmann manifolds share very similar properties with gyrovector spaces. We have presented new methods for generalizing Euclidean neural networks to the SPD and Grassmann manifold settings. Our experimental results on human activity understanding and question answering have demonstrated the effectiveness of the proposed approach.

### Acknowledgments

We are grateful for the constructive comments and feedback from the anonymous reviewers. We thank Amir Shahroudy, Jun Liu, Tian-Tsong Ng, Gang Wang, Guillermo Garcia-Hernando, Shanxin Yuan, Seungryul Baek, and Tae-Kyun Kim for giving us access to the NTU RGB+D and FPHA datasets.

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
