# The Gyro-Structure of Some Matrix Manifolds

**Xuan Son Nguyen**
ETIS, UMR 8051, CY Cergy Paris Université, ENSEA, CNRS, Cergy, France
`xuan-son.nguyen@ensea.fr`

This supplemental material provides the proofs for the Theorems and Lemmas presented in our paper. In Sections 7 and 8, we give more details on the proposed methods, datasets, experimental settings, and experimental results. To further demonstrate the applicability of our approach, we present a method for knowledge graph completion in Section 9. Please see the paper for references.

## 7 Human Activity Understanding

### 7.1 Datasets and Experimental Settings

**HDM05 dataset** This dataset has 2337 sequences of 3D skeleton data classified into 130 classes. Each frame contains the 3D coordinates of 31 body joints. We use all the action classes and follow the experimental protocol [15] in which 2 subjects are used for training and the remaining 3 subjects are used for testing.

**FPHA dataset** This dataset has 1175 sequences of 3D skeleton data classified into 45 classes. Each frame contains the 3D coordinates of 21 hand joints. We follow the experimental protocol [13] in which 600 sequences are used for training and 575 sequences are used for testing.

**NTU60 dataset** This dataset has 56880 sequences of 3D skeleton data classified into 60 classes. Each frame contains the 3D coordinates of 25 or 50 body joints. We use the mutual actions and follow the cross-subject experimental protocol [38] in which data from 20 subjects are used for training, and those from the other 20 subjects are used for testing.

For all the datasets, we use interpolation to create sequences of the same length.

**Comparison against SPD Neural Networks** For SPDNet and SPDNetBN, we compute a covariance matrix to represent an input sequence as in [20]. The sizes of the covariance matrices are respectively $93 \times 93$, $60 \times 60$, and $150 \times 150$ for HDM05, FPHA, and NTU60 datasets. The sizes of the transformation matrices for the experiments on FPHA dataset are set to $60 \times 50$, $50 \times 40$, $40 \times 30$, respectively. The sizes of the transformation matrices for the experiments on NTU60 dataset are set to $150 \times 100$, $100 \times 60$, $60 \times 30$, respectively.

Our networks are implemented with Tensorflow framework. They are trained using cross-entropy loss and Adadelta optimizer for 500 epochs. We use a batch size of 32 for HDM05 and FPHA datasets, and a batch size of 256 for NTU60 dataset. The number of frames in each sequence is set to 100. The learning rate and parameter $\epsilon$ for the pointwise nonlinearity are set respectively to $10^{-3}$ and $10^{-4}$. The positive real number $r$ is set to 1. The constant $c$ is set to 10. The ReLU function is used for the pointwise nonlinearity. All experiments are conducted on a machine with Intel Core i7-6700 CPU 3.40 GHz 24GB RAM.

### 7.2 More Results

**Comparison against Euclidean RNNs, transformers, and HNNs** Our networks are compared against LSTM, ST-TR [62], and HypGRU [55]. Results of LSTM are obtained using its implementation provided by the Tensorflow framework. Those of ST-TR and HypGRU are obtained using

Table 3: Accuracy comparison (%) of our networks against LSTM, ST-TR, and HypGRU.

| Dataset | LSTM | | ST-TR | | HypGRU | | GyroAI-HAUNet | | GyroLE-HAUNet | |
|---|---|---|---|---|---|---|---|---|---|---|
| | $M=1$ | $M=3$ | $M=1$ | $M=3$ | $M=1$ | $M=3$ | $M=1$ | $M=3$ | $M=1$ | $M=3$ |
| HDM05 | 49.42 | 72.82 | 69.75 | 76.12 | 51.35 | 58.50 | 61.50 | 78.14 | 57.01 | 74.53 |
| #HDM05 | 0.09 MB | 0.54 MB | 4.62 MB | 27.73 MB | 0.10 MB | 0.61 MB | 0.05 MB | 0.31 MB | 0.05 MB | 0.31 MB |
| FPHA | 66.43 | 81.22 | 86.32 | 91.34 | 56.70 | 61.42 | 89.73 | 96.00 | 83.03 | 89.94 |
| #FPHA | 0.07 MB | 0.41 MB | 4.59 MB | 27.55 MB | 0.08 MB | 0.47 MB | 0.02 MB | 0.11 MB | 0.02 MB | 0.11 MB |
| NTU60 | 85.39 | 87.27 | 92.18 | 93.78 | 86.51 | 88.03 | 83.12 | 94.72 | 77.25 | 89.44 |
| #NTU60 | 0.006 MB | 0.035 MB | 4.58 MB | 27.50 MB | 0.006 MB | 0.039 MB | 0.004 MB | 0.026 MB | 0.004 MB | 0.026 MB |

Table 4: Accuracy comparison (%) of our networks against state-of-the-art graph neural networks.

| Dataset | ST-GCN | | Shift-GCN-light | | Shift-GCN | | GyroAI-HAUNet | | GyroLE-HAUNet | |
|---|---|---|---|---|---|---|---|---|---|---|
| | $M=1$ | $M=3$ | $M=1$ | $M=3$ | $M=1$ | $M=3$ | $M=1$ | $M=3$ | $M=1$ | $M=3$ |
| HDM05 | 70.02 | 76.58 | 67.25 | 76.18 | 73.57 | 80.28 | 61.50 | 78.14 | 57.01 | 74.53 |
| #HDM05 | 2.96 MB | 17.73 MB | 0.22 MB | 1.33 MB | 0.70 MB | 4.20 MB | 0.05 MB | 0.31 MB | 0.05 MB | 0.31 MB |
| FPHA | 75.19 | 78.78 | 86.65 | 89.81 | 87.52 | 91.08 | 89.73 | 96.00 | 83.03 | 89.94 |
| #FPHA | 2.93 MB | 17.60 MB | 0.19 MB | 1.15 MB | 0.64 MB | 3.84 MB | 0.02 MB | 0.11 MB | 0.02 MB | 0.11 MB |
| NTU60 | 87.62 | 91.75 | 89.56 | 92.37 | 92.84 | 95.01 | 83.12 | 94.72 | 77.25 | 89.44 |
| #NTU60 | 2.94 MB | 17.66 MB | 0.19 MB | 1.15 MB | 0.65 MB | 3.90 MB | 0.004 MB | 0.026 MB | 0.004 MB | 0.026 MB |

their official code.[1,2] In all cases, HypGRU achieves the best results when the data are projected to hyperbolic spaces before they are fed to the network, and all its layers are based on hyperbolic geometry. To concatenate the outputs of HypGRU for temporal pyramid representations, we employ the concatenation operation proposed in the recent work [33]. The dimensions of input data at a frame are respectively 93, 60, and 150 for HDM05, FPHA, and NTU60 datasets. The hidden dimension is set to 100. Other parameters are set to their default values. Results for $M=1$ and $M=3$ are shown in Tab. 3. We can observe that our models outperform the competing models by large margins in most cases. Furthermore, our models are more advantageous than these models in terms of model size.

**Comparison against GNNs**   Our networks are compared against some state-of-the-art GNNs for 3D skeleton-based action recognition, i.e., ST-GCN [68] and Shift-GCN [51]. Results of these networks are obtained using their official code.[3,4] We also evaluate a light version of Shift-GCN referred to as Shift-GCN-light, where the numbers of input and output channels for the input and residual blocks are reduced by a factor of 2 (the number of input channels for the input block is 3). Results for $M=1$ and $M=3$ are shown in Tab. 4. As can be seen, ST-GCN does not perform well on FPHA dataset, which indicates that it is not able to capture hand joint movements on this dataset. This might be due to the fact that the underlying structure and movement of hand skeletons are different from those of body skeletons. This is confirmed by observing that Shift-GCN performs worse than GyroAI-HAUNet on FPHA. We can also see that when $M=3$, GyroAI-HAUNet outperforms Shift-GCN-light on all the datasets. Overall, when $M=3$, GyroAI-HAUNet is competitive to the best GNN model with far fewer parameters. Tab. 5 reports the mean accuracies and standard deviations of some representative methods from five runs. We note that GyroAI-HAUNet is the best method in terms of standard deviation.

**Ablation Study**   We show the impact of Eq. (13) by comparing GyroAI-HAUNet and GyroLE-HAUNet against their variant GyroNet-Baseline based only on Eq. (12). Results for $M=3$ are given in Tab. 6. We can observe that both GyroAI-HAUNet and GyroLE-HAUNet outperform GyroNet-Baseline, demonstrating the effectiveness of Eq. (13).

Results of GyroAI-HAUNet with different settings of $r$ are given in Tab. 7. Results show that $r$ can have an important impact on our network performance. For example, on HDM05, the performance gap between the two settings $r=1.0$ and $r=2.0$ is 4.79%.

---

[1] https://github.com/Chiaraplizz/ST-TR
[2] https://github.com/dalab/hyperbolic_nn
[3] https://github.com/yysijie/st-gcn
[4] https://github.com/kchengiva/Shift-GCN

Table 5: Results (mean $\pm$ standard deviation) of some representative methods (computed over 5 runs).

| Dataset | SPDNet | SPDNetBN | Shift-GCN | GyroAI-HAUNet |
|---------|--------|----------|-----------|---------------|
| HDM05 | $70.78 \pm 1.61$ | $74.37 \pm 1.54$ | $78.02 \pm 1.94$ | $77.05 \pm 1.35$ |
| FPHA | $87.53 \pm 0.39$ | $91.55 \pm 0.27$ | $90.19 \pm 0.56$ | $95.65 \pm 0.23$ |
| NTU60 | $75.94 \pm 1.45$ | $78.16 \pm 1.36$ | $93.92 \pm 1.78$ | $93.27 \pm 1.29$ |

Table 6: Ablation study for the update equations of our networks.

| Dataset | HDM05 | FPHA | NTU60 |
|---------|-------|------|-------|
| GyroNet-Baseline | 72.41 | 88.27 | 86.73 |
| GyroLE-HAUNet | 74.53 | 89.94 | 89.44 |
| GyroAI-HAUNet | 78.14 | 96.00 | 94.72 |

# 8  Question Answering

## 8.1  Proposed Method

Here we give details on how to compute the scoring function for our models based on product manifolds $\mathrm{Gr}_{n_1,p} \times \mathrm{Sym}_{n_2}^+$. The embedding of answer $a$ is the summation of those of its tokens using operation $\oplus_{grai}$ defined as

$$(\mathbf{T}_1^{gr}, \mathbf{T}_1^{spd}) \oplus_{grai} \ldots \oplus_{grai} (\mathbf{T}_l^{gr}, \mathbf{T}_l^{spd}) = \Big( \mathbf{T}_1^{gr} \oplus_{gr} \big( \ldots (\mathbf{T}_{l-1}^{gr} \oplus_{gr} \mathbf{T}_l^{gr}) \ldots \big),$$

$$\mathbf{T}_1^{spd} \oplus_{ai} \big( \ldots (\mathbf{T}_{l-1}^{spd} \oplus_{ai} \mathbf{T}_l^{spd}) \ldots \big) \Big),$$

where $(\mathbf{T}_i^{gr}, \mathbf{T}_i^{spd}) \in \mathrm{Gr}_{n_1,p} \times \mathrm{Sym}_{n_2}^+, i = 1, \ldots, l$ are embeddings of the tokens in answer $a$, and $\oplus_{ai} \triangleq \oplus_{ai}^1 (r = 1)$. The summation by operation $\oplus_{gr} (\oplus_{ai})$ is performed from right to left.

The embedding of question $q$ is computed by

$$\mathbf{Q} = \Big( \mathbf{B}_{gr} \oplus_{gr} \big( (\mathbf{S} \otimes_{gr}^m \mathbf{T}_1^{gr}) \oplus_{gr} \ldots \oplus_{gr} (\mathbf{S} \otimes_{gr}^m \mathbf{T}_l^{gr}) \big), T(\mathbf{T}_1^{spd} \oplus_{ai} \ldots \oplus_{ai} \mathbf{T}_l^{spd}) \oplus_{ai} \mathbf{B}_{spd} \Big),$$

where $(\mathbf{T}_i^{gr}, \mathbf{T}_i^{spd}) \in \mathrm{Gr}_{n_1,p} \times \mathrm{Sym}_{n_2}^+, i = 1, \ldots, l$ are embeddings of the tokens in question $q$, $\mathbf{S} \in \mathrm{M}_{p,n_1-p}$, $\mathbf{B}_{gr} \in \mathrm{Gr}_{n_1,p}$, and $\mathbf{B}_{spd} \in \mathrm{Sym}_{n_2}^+$ are parameters of the model, $T(.)$ is a feature transformation which can be a scaling, rotation or reflection [27]. The scaling transformation (matrix scaling) $\otimes_{spd}^m$ is defined as

$$T(\mathbf{P}) = \mathbf{A} \otimes_{spd}^m \mathbf{P} = \exp(\mathbf{A} * \log(\mathbf{P})),$$

where $\mathbf{A} \in \mathrm{Sym}_n$ is a parameter of the model, and $\mathbf{P} \in \mathrm{Sym}_n^+$. The rotation and reflection transformations are defined as following. For any $\theta \in [0, 2\pi)$ and choice of sign $\{+, -\}$, let $R^\pm(\theta)$ be the following matrix

$$R^\pm(\theta) = \begin{pmatrix} \cos\theta \mp \sin\theta \\ \sin\theta \pm \cos\theta \end{pmatrix}.$$

For any pair $i < j, i, j = 1, \ldots, n$, let $R_{ij}^\pm(\theta)$ be the matrix obtained by replacing the entries at positions $(i, i), (i, j), (j, i), (j, j)$ of $\mathbf{I}_n$ with the corresponding values of $R^\pm(\theta)$. Given a vector of angles $\vec{\theta} = (\theta_{12}, \ldots, \theta_{n-1n}) \in \mathbb{R}^{\frac{n(n-1)}{2}}$, the rotation and reflection matrices corresponding to $\vec{\theta}$ are defined as

$$\mathrm{Rot}(\vec{\theta}) = \prod_{i<j} R_{ij}^+(\theta_{ij}), \qquad \mathrm{Ref}(\vec{\theta}) = \prod_{i<j} R_{ij}^-(\theta_{ij}).$$

The rotation (reflection) transformation $T(.)$ is then obtained as

$$T(\mathbf{P}) = \mathbf{M}\mathbf{P}\mathbf{M}^T,$$

where $\mathbf{M}$ is the rotation (reflection) matrix defined above. The vector of angles $\vec{\theta}$ is a parameter of the model.

Table 7: Ablation study for the impact of $r$ on the performance of GyroAI-HAUNet.

| Dataset | HDM05 | FPHA | NTU60 |
|---|---|---|---|
| $r = 0.8$ | 77.08 | 93.56 | 95.14 |
| $r = 1.0$ | 78.14 | 96.00 | 94.72 |
| $r = 1.2$ | 82.52 | 95.64 | 93.68 |
| $r = 2.0$ | 82.93 | 94.61 | 94.32 |

Table 8: Statistics of TrecQA and WikiQA datasets.

| Dataset | Split | #Questions | #Pairs |
|---|---|---|---|
| TrecQA | TRAIN | 94 | 4718 |
| | DEV | 65 | 1117 |
| | TEST | 68 | 1442 |
| WikiQA | TRAIN | 873 | 8672 |
| | DEV | 126 | 1130 |
| | TEST | 243 | 2351 |

## 8.2 Datasets and Experimental Settings

Statistics of TrecQA and WikiQA datasets are given in Tab. 8. The networks are implemented with Pytorch framework. They are trained using binary cross-entropy loss and SGD optimizer for 300 epochs. The learning rate is set to $10^{-3}$ with weight decay of $10^{-5}$. The batch size is set to 64. The number of negative samples is set to 8. The constant $\tau$ is set to 2. We test with the number of negative samples from $\{2, 4, 6, 8, 10, 12\}$ where random sampling is used [64], and the batch size from $\{32, 64, 128, 256, 512\}$. For the SPD models of [27], we test with SPD matrices of dimensions $n \times n$ where $n \in \{8, 10, 12, 14\}$.[5] For our Grassman model GyroGR-QANet, we test with embeddings in $\mathrm{Gr}_{n,p}$ where $(n, p) \in \{(2k, k)\}, k = 5, 6, \ldots, 12$. For our models based on product manifolds, we test with $\tau$ from $\{0.5, 1, 1.5, 2, 2.5, 3\}$. Early stopping is used when the MRR score of the model on the development set does not improve after 20 epochs. We use the trecval tool[6] to compute MAP and MRR scores [64]. In all experiments, the models that obtain the best MRR scores on the development set are used for testing [64]. All experiments are conducted on a machine with Intel Core i7-9700 CPU 3.0 GHz 15GB RAM.

## 8.3 More Results

Tabs. 9, 10, and 11 present results obtained with different embedding dimensions. In each table, we compare the performance of the embeddings in $\mathrm{Gr}_{14,7} \times \mathrm{Sym}_n^+$ against that of the embeddings in $\mathrm{Gr}_{14,7}$ and $\mathrm{Sym}_n^+$, where $n = 8, 10$, and 12. In most cases, the embeddings in $\mathrm{Gr}_{14,7} \times \mathrm{Sym}_n^+$ compare favorably against the embeddings in $\mathrm{Sym}_n^+$. In some cases, the embeddings in product manifolds outperform the SPD embeddings by a large margin. For example, on TrecQA dataset, when $n = 12$, $\mathrm{SPD}_{Sca}^{F_1}$, $\mathrm{SPD}_{Rot}^{F_1}$, and $\mathrm{SPD}_{Ref}^{F_1}$ give the mean MAP scores of 47.75%, 47.39%, and 48.80%, respectively. When these networks are combined with the Grassmann embeddings, they achieve the mean MAP scores of 50.09%, 49.27%, and 50.07%, respectively. Tab. 12 presents a comparison of our models (only the one with 49 DOF and those with 85 DOF are shown) against the Euclidean model where a linear layer is used as feature transformation. The results indicate that embeddings in Grassmann manifolds and product spaces of Grassmann and SPD manifolds are more effective than those in Euclidean spaces on the two datasets.

## 9 Knowledge Graph Completion

In this section, we consider learning entity and relation embeddings in product manifolds $\mathrm{Gr}_{n_1,p} \times \mathrm{Sym}_{n_2}^+$ for knowledge graph completion.

---

[5]We did not observe better results when $n > 14$.

[6]https://trec.nist.gov/

Table 9: Effectiveness of embeddings in product manifolds. The SPD models learn embeddings in $\mathrm{Sym}_8^+$. Our model with 49 DOF learns embeddings in $\mathrm{Gr}_{14,7}$. Our models with 85 DOF learn embeddings in $\mathrm{Gr}_{14,7} \times \mathrm{Sym}_8^+$.

| DOF | Model | TrecQA | | WikiQA | |
|---|---|---|---|---|---|
| | | MAP | MRR | MAP | MRR |
| 36 | $\mathrm{SPD}_{\mathrm{Sca}}^{R}$ | $49.68 \pm 1.37$ | $58.81 \pm 1.47$ | $59.26 \pm 0.45$ | $60.08 \pm 0.42$ |
| 36 | $\mathrm{SPD}_{\mathrm{Sca}}^{F_1}$ | $49.96 \pm 0.72$ | $57.84 \pm 1.78$ | $60.21 \pm 0.78$ | $61.66 \pm 0.75$ |
| 36 | $\mathrm{SPD}_{\mathrm{Rot}}^{R}$ | $49.51 \pm 1.56$ | $58.50 \pm 1.97$ | $60.19 \pm 1.32$ | $61.31 \pm 1.69$ |
| 36 | $\mathrm{SPD}_{\mathrm{Rot}}^{F_1}$ | $50.03 \pm 0.95$ | $57.81 \pm 1.44$ | $58.95 \pm 1.42$ | $60.76 \pm 1.60$ |
| 36 | $\mathrm{SPD}_{\mathrm{Ref}}^{R}$ | $48.74 \pm 1.62$ | $57.01 \pm 3.17$ | $58.43 \pm 0.50$ | $59.57 \pm 0.67$ |
| 36 | $\mathrm{SPD}_{\mathrm{Ref}}^{F_1}$ | $48.67 \pm 1.96$ | $57.64 \pm 2.24$ | $59.32 \pm 1.20$ | $61.08 \pm 1.36$ |
| 49 | GyroGR-QANet | $50.18 \pm 1.29$ | $58.19 \pm 2.59$ | $56.69 \pm 1.45$ | $58.26 \pm 1.45$ |
| 85 | GyroGR-$\mathrm{SPD}_{\mathrm{Sca}}^{F_1}$-QANet | $50.10 \pm 0.30$ | $57.70 \pm 0.93$ | $60.62 \pm 0.25$ | $62.42 \pm 0.16$ |
| 85 | GyroGR-$\mathrm{SPD}_{\mathrm{Rot}}^{F_1}$-QANet | $50.27 \pm 0.56$ | $58.62 \pm 1.35$ | $59.78 \pm 0.15$ | $61.66 \pm 0.23$ |
| 85 | GyroGR-$\mathrm{SPD}_{\mathrm{Ref}}^{F_1}$-QANet | $48.83 \pm 1.89$ | $58.11 \pm 0.87$ | $60.41 \pm 0.39$ | $61.86 \pm 0.35$ |

Table 10: Effectiveness of embeddings in product manifolds. The SPD models learn embeddings in $\mathrm{Sym}_{10}^+$. Our model with 49 DOF learns embeddings in $\mathrm{Gr}_{14,7}$. Our models with 104 DOF learn embeddings in $\mathrm{Gr}_{14,7} \times \mathrm{Sym}_{10}^+$.

| DOF | Model | TrecQA | | WikiQA | |
|---|---|---|---|---|---|
| | | MAP | MRR | MAP | MRR |
| 55 | $\mathrm{SPD}_{\mathrm{Sca}}^{R}$ | $48.46 \pm 0.51$ | $56.89 \pm 0.85$ | $59.88 \pm 0.02$ | $61.11 \pm 0.02$ |
| 55 | $\mathrm{SPD}_{\mathrm{Sca}}^{F_1}$ | $47.91 \pm 2.09$ | $56.96 \pm 1.88$ | $59.42 \pm 0.44$ | $60.68 \pm 0.57$ |
| 55 | $\mathrm{SPD}_{\mathrm{Rot}}^{R}$ | $48.41 \pm 0.41$ | $56.13 \pm 0.67$ | $59.62 \pm 0.71$ | $60.80 \pm 0.69$ |
| 55 | $\mathrm{SPD}_{\mathrm{Rot}}^{F_1}$ | $48.33 \pm 0.95$ | $54.49 \pm 0.85$ | $60.51 \pm 0.98$ | $62.20 \pm 1.20$ |
| 55 | $\mathrm{SPD}_{\mathrm{Ref}}^{R}$ | $49.11 \pm 0.68$ | $57.53 \pm 2.02$ | $59.40 \pm 0.79$ | $60.86 \pm 0.80$ |
| 55 | $\mathrm{SPD}_{\mathrm{Ref}}^{F_1}$ | $50.34 \pm 1.08$ | $57.09 \pm 1.31$ | $61.54 \pm 1.09$ | $62.46 \pm 1.04$ |
| 49 | GyroGR-QANet | $50.18 \pm 1.29$ | $58.19 \pm 2.59$ | $56.69 \pm 1.45$ | $58.26 \pm 1.45$ |
| 104 | GyroGR-$\mathrm{SPD}_{\mathrm{Sca}}^{F_1}$-QANet | $47.99 \pm 0.75$ | $57.04 \pm 0.46$ | $59.64 \pm 0.19$ | $60.95 \pm 0.36$ |
| 104 | GyroGR-$\mathrm{SPD}_{\mathrm{Rot}}^{F_1}$-QANet | $48.65 \pm 1.17$ | $56.63 \pm 0.37$ | $60.84 \pm 0.70$ | $62.64 \pm 0.85$ |
| 104 | GyroGR-$\mathrm{SPD}_{\mathrm{Ref}}^{F_1}$-QANet | $50.48 \pm 0.87$ | $57.70 \pm 1.12$ | $61.60 \pm 0.45$ | $62.71 \pm 0.40$ |

## 9.1 Problem Formulation

A knowledge graph is a multi-relational graph representation of a collection $\mathcal{F}$ of facts in triple form $(e_s, r, e_o) \in \mathcal{E} \times \mathcal{R} \times \mathcal{E}$, where $\mathcal{E}$ is the set of entities (nodes) and $\mathcal{R}$ is the set of binary relations between them [47]. If $(e_s, r, e_o) \in \mathcal{F}$, then subject entity $e_s$ is related to object entity $e_o$ by relation $r$. Knowledge graphs are often incomplete, so the aim is to infer other true facts. A typical approach is to learn a scoring function $\phi : \mathcal{E} \times \mathcal{R} \times \mathcal{E} \to \mathbb{R}$, that assigns a score $\phi(e_s, r, e_o)$ to each triple, indicating the likelihood that a particular triple corresponds to a true fact [47].

## 9.2 Proposed Method

Our model learns a scoring function given as

$$\phi_{kgc}(e_s, r, e_o) = -d((\mathbf{A} \otimes \mathbf{S}) \oplus \mathbf{R}, \mathbf{O})^2 + b_s + b_o, \tag{14}$$

where $\mathbf{S}$ and $\mathbf{O}$ are embeddings of the subject and object entities, respectively, $\mathbf{R}$ and $\mathbf{A}$ are matrices associated with relation $r$, $b_s, b_o \in \mathbb{R}$ are scalar biases for the subject and object entities, respectively. The operation $\oplus$ is defined as

$$(\mathbf{P}_{gr}, \mathbf{P}_{spd}) \oplus (\mathbf{R}_{gr}, \mathbf{R}_{spd}) = (\mathbf{P}_{gr} \oplus_{gr} \mathbf{R}_{gr}, \mathbf{P}_{spd} \oplus_{le} \mathbf{R}_{spd}),$$

where $\mathbf{P}_{gr}, \mathbf{R}_{gr} \in \mathrm{Gr}_{n_1, p}$, $\mathbf{P}_{spd}, \mathbf{R}_{spd} \in \mathrm{Sym}_{n_2}^+$, $\oplus_{le} \triangleq \oplus_{le}^1$, and operation $\otimes$ is defined as

$$(\mathbf{A}_{gr}, \mathbf{A}_{spd}) \otimes (\mathbf{S}_{gr}, \mathbf{S}_{spd}) = (\mathbf{A}_{gr} \otimes_{gr}^m \mathbf{S}_{gr}, \mathbf{A}_{spd} \otimes_{spd}^m \mathbf{S}_{spd}),$$

Table 11: Effectiveness of embeddings in product manifolds. The SPD models learn embeddings in $\mathrm{Sym}_{12}^+$. Our model with 49 DOF learns embeddings in $\mathrm{Gr}_{14,7}$. Our models with 127 DOF learn embeddings in $\mathrm{Gr}_{14,7} \times \mathrm{Sym}_{12}^+$.

| DOF | Model | TrecQA | | WikiQA | |
|---|---|---|---|---|---|
| | | MAP | MRR | MAP | MRR |
| 78 | $\mathrm{SPD}_{\mathrm{Sca}}^{R}$ | $46.02 \pm 0.12$ | $55.66 \pm 0.13$ | $58.94 \pm 1.46$ | $60.35 \pm 1.62$ |
| 78 | $\mathrm{SPD}_{\mathrm{Sca}}^{F_1}$ | $47.75 \pm 0.12$ | $58.47 \pm 0.12$ | $59.84 \pm 0.51$ | $61.39 \pm 0.60$ |
| 78 | $\mathrm{SPD}_{\mathrm{Rot}}^{R}$ | $48.35 \pm 1.21$ | $56.35 \pm 0.90$ | $59.28 \pm 0.67$ | $60.50 \pm 0.77$ |
| 78 | $\mathrm{SPD}_{\mathrm{Rot}}^{F_1}$ | $47.39 \pm 1.29$ | $56.21 \pm 1.97$ | $61.33 \pm 0.90$ | $62.81 \pm 1.04$ |
| 78 | $\mathrm{SPD}_{\mathrm{Ref}}^{R}$ | $48.53 \pm 1.46$ | $56.73 \pm 1.22$ | $58.75 \pm 0.37$ | $60.28 \pm 0.44$ |
| 78 | $\mathrm{SPD}_{\mathrm{Ref}}^{F_1}$ | $48.80 \pm 1.22$ | $55.99 \pm 2.55$ | $61.32 \pm 0.61$ | $63.02 \pm 0.97$ |
| 49 | GyroGR-QANet | $50.18 \pm 1.29$ | $58.19 \pm 2.59$ | $56.69 \pm 1.45$ | $58.26 \pm 1.45$ |
| 127 | GyroGR-SPD$_{\mathrm{Sca}}^{F_1}$-QANet | $50.09 \pm 0.65$ | $59.26 \pm 0.43$ | $60.48 \pm 0.64$ | $62.18 \pm 1.07$ |
| 127 | GyroGR-SPD$_{\mathrm{Rot}}^{F_1}$-QANet | $49.27 \pm 1.14$ | $56.53 \pm 1.87$ | $61.36 \pm 0.73$ | $62.78 \pm 0.76$ |
| 127 | GyroGR-SPD$_{\mathrm{Ref}}^{F_1}$-QANet | $50.07 \pm 0.38$ | $57.13 \pm 1.14$ | $61.44 \pm 0.09$ | $63.33 \pm 0.61$ |

Table 12: Comparison of our models against the Euclidean model.

| DOF | Model | TrecQA | | WikiQA | |
|---|---|---|---|---|---|
| | | MAP | MRR | MAP | MRR |
| 105 | Euclidean | $46.81 \pm 0.85$ | $54.85 \pm 1.41$ | $54.60 \pm 0.62$ | $55.94 \pm 0.75$ |
| 49 | GyroGR-QANet | $50.18 \pm 1.29$ | $58.19 \pm 2.59$ | $56.69 \pm 1.45$ | $58.26 \pm 1.45$ |
| 85 | GyroGR-SPD$_{\mathrm{Sca}}^{F_1}$-QANet | $50.10 \pm 0.30$ | $57.70 \pm 0.93$ | $60.62 \pm 0.25$ | $62.42 \pm 0.16$ |
| 85 | GyroGR-SPD$_{\mathrm{Rot}}^{F_1}$-QANet | $50.27 \pm 0.56$ | $58.62 \pm 1.35$ | $59.78 \pm 0.15$ | $61.66 \pm 0.23$ |
| 85 | GyroGR-SPD$_{\mathrm{Ref}}^{F_1}$-QANet | $48.83 \pm 1.89$ | $58.11 \pm 0.87$ | $60.41 \pm 0.39$ | $61.86 \pm 0.35$ |

where $\mathbf{S}_{gr} \in \mathrm{Gr}_{n_1,p}$, $\mathbf{S}_{spd} \in \mathrm{Sym}_{n_2}^+$, $\mathbf{A}_{gr} \in \mathrm{M}_{p,n_1-p}$ and $\mathbf{A}_{spd} \in \mathrm{Sym}_{n_2}$ are matrices associated with relation $r$, and operation $\otimes_{spd}^m$ is the matrix scaling defined in Section 8.1. The distance function is computed in a similar way as in Section 4.2.1, i.e.,

$$d((\mathbf{P}_{gr}, \mathbf{P}_{spd}), (\mathbf{R}_{gr}, \mathbf{R}_{spd})) = d_{gr}(\mathbf{P}_{gr}, \mathbf{R}_{gr}) + \tau d_{spd}^{LE}(\mathbf{P}_{spd}, \mathbf{R}_{spd}),$$

where $\tau$ is a constant, and $d_{spd}^{LE}(.,.)$ is the Riemannian distance induced by the LE metric, i.e.,

$$d_{spd}^{LE}(\mathbf{P}, \mathbf{R}) = \| \log(\mathbf{P}) - \log(\mathbf{R}) \|_F.$$

## 9.3 Datasets and Experimental Settings

We use the WN18RR [52] dataset. It is a subset of WordNet [59], a hierarchical collection of relations between words, created from WN18 [50] by removing the inverse of many relations from validation and test sets to make the dataset more challenging. It contains 93003 triples with 40943 entities and 11 relations. The networks are implemented with Pytorch framework. They are trained using binary cross-entropy loss and SGD optimizer for 5000 epochs. The learning rate is set to $10^{-3}$ with weight decay of $10^{-5}$. The batch size is set to $4096$. The number of negative samples is set to $10$. These settings are taken from [29]. The constant $\tau$ is set to 2 based on our experiments in Section 8.2. Early stopping is used when the MRR score of the model on the validation set does not improve after 500 epochs. In all experiments, the models that obtain the best MRR scores on the validation set are used for testing. All experiments are conducted on a machine with Intel Core i7-9700 CPU 3.0 GHz 15GB RAM.

Table 13: Comparison of our model against the SPD models of [27] on the validation set of WN18RR dataset. The SPD models learn embeddings in $\mathrm{Sym}_6^+$. GyroGR-KGCNet learns embeddings in $\mathrm{Gr}_{5,2}$. GyroLE-KGCNet learns embeddings in $\mathrm{Sym}_5^+$. GyroGRLE-KGCNet learns embeddings in $\mathrm{Gr}_{5,2} \times \mathrm{Sym}_5^+$.

| DOF | Model | MRR | H@1 | H@3 | H@10 | Time (seconds) | |
| --- | --- | --- | --- | --- | --- | --- | --- |
| | | | | | | Train/epoch | Test |
| 21 | $\mathrm{SPD}_{\mathrm{Sca}}^{R}$ | 43.7 | 39.7 | 47.2 | 52.5 | 2.4 | 192.6 |
| | $\mathrm{SPD}_{\mathrm{Sca}}^{F_1}$ | 43.1 | 38.7 | 46.4 | 52.8 | 2.4 | 196.1 |
| | $\mathrm{SPD}_{\mathrm{Rot}}^{R}$ | 23.3 | 10.0 | 38.8 | 50.2 | 2.5 | 192.6 |
| | $\mathrm{SPD}_{\mathrm{Rot}}^{F_1}$ | 28.3 | 20.1 | 33.5 | 45.4 | 2.5 | 196.1 |
| | $\mathrm{SPD}_{\mathrm{Ref}}^{R}$ | 43.9 | 40.5 | 46.5 | 51.3 | 2.5 | 192.6 |
| | $\mathrm{SPD}_{\mathrm{Ref}}^{F_1}$ | 42.8 | 38.9 | 45.6 | 50.9 | 2.5 | 196.1 |
| | GyroGRLE-KGCNet | 44.1 | 38.8 | 48.2 | 54.9 | 1.1 | 4.5 |
| 15 | GyroLE-KGCNet | 41.4 | 37.3 | 44.9 | 50.6 | | |
| 6 | GyroGR-KGCNet | 12.7 | 7.4 | 13.8 | 27.5 | | |

Table 14: Comparison of our model against the SPD models of [27] on the test set of WN18RR dataset. The SPD models learn embeddings in $\mathrm{Sym}_6^+$. GyroGR-KGCNet learns embeddings in $\mathrm{Gr}_{5,2}$. GyroLE-KGCNet learns embeddings in $\mathrm{Sym}_5^+$. GyroGRLE-KGCNet learns embeddings in $\mathrm{Gr}_{5,2} \times \mathrm{Sym}_5^+$.

| DOF | Model | MRR | H@1 | H@3 | H@10 | Time (seconds) | |
| --- | --- | --- | --- | --- | --- | --- | --- |
| | | | | | | Train/epoch | Test |
| 21 | $\mathrm{SPD}_{\mathrm{Sca}}^{R}$ | 41.7 | 36.5 | 44.5 | 51.1 | 2.4 | 192.6 |
| | $\mathrm{SPD}_{\mathrm{Sca}}^{F_1}$ | 40.8 | 36.3 | 42.9 | 49.5 | 2.4 | 196.1 |
| | $\mathrm{SPD}_{\mathrm{Rot}}^{R}$ | 22.4 | 8.4 | 33.4 | 47.3 | 2.5 | 192.6 |
| | $\mathrm{SPD}_{\mathrm{Rot}}^{F_1}$ | 26.5 | 18.1 | 30.7 | 42.9 | 2.5 | 196.1 |
| | $\mathrm{SPD}_{\mathrm{Ref}}^{R}$ | 41.0 | 37.1 | 42.7 | 47.6 | 2.5 | 192.6 |
| | $\mathrm{SPD}_{\mathrm{Ref}}^{F_1}$ | 39.7 | 35.9 | 41.5 | 46.3 | 2.5 | 196.1 |
| | GyroGRLE-KGCNet | 41.5 | 35.3 | 44.9 | 52.1 | 1.1 | 4.5 |
| 15 | GyroLE-KGCNet | 37.8 | 33.4 | 39.9 | 45.2 | | |
| 6 | GyroGR-KGCNet | 11.5 | 5.9 | 11.1 | 25.0 | | |

## 9.4 Results

The MRR and hits at $K$ (H@K, $K = 1, 3, 10$) are used as evaluation metrics [47]. Our model[7] is compared against the SPD models[8] of [29].

We consider the case of low-dimensional embeddings where each model has 21 DOF. Results on the validation and test sets are presented in Tabs. 13 and 14, respectively. We also report results of GyroLE-KGCNet and GyroGR-KGCNet which learn embeddings in $\mathrm{Sym}_5^+$ and $\mathrm{Gr}_{5,2}$, respectively. Both models learn a scoring function given in Eq. (14). The scoring function of GyroLE-KGCNet is constructed from operations $\otimes_{spd}^m$ and $\oplus_{le}$. The scoring function of GyroGR-KGCNet is constructed from operations $\otimes_{gr}^m$ and $\oplus_{gr}$. We can notice that GyroGRLE-KGCNet improves both GyroLE-KGCNet and GyroGR-KGCNet in all the cases. The performance improvements are significant in most of the cases. On the validation set, our model achieves better MRR, H@3, and H@10 scores than the SPD models, while on the test set, our model achieves better H@3 and H@10 scores than the SPD models. Our model has clear advantage in computation time.

---

[7]Code available at https://github.com/spratmnt/kgc
[8]https://github.com/fedelopez77/gyrospd

## 10 Riemannian Geometry of SPD Manifolds

The space of SPD matrices is part of the vector space of square matrices. However, as mentioned in [3], employing the Euclidean metric for computations in this space can be problematic from both practical and theoretical points of view, i.e., the boundary problem or the tensor swelling effect. These representative works provide effective solutions to address these problems.

**Affine-Invariant Metrics** Based on the general principle of designing Riemannian metrics [60], Pennec et al. [61] proposed the AI metric that is invariant under the action of affine transformations of the underlying space, i.e.,

$$< \mathbf{A}_1 | \mathbf{A}_2 >_\mathbf{P} = < \mathbf{Q} \star \mathbf{A}_1 | \mathbf{Q} \star \mathbf{A}_2 >_{\mathbf{Q} \star \mathbf{P}},$$

where $\mathbf{P} \in \mathrm{Sym}_n^+$, $\mathbf{A}_1$ and $\mathbf{A}_2 \in \mathrm{Sym}_n$ are tangent vectors at $\mathbf{P}$, $< . | . >$ is the dot product, $\mathbf{Q} \star \mathbf{P} = \mathbf{Q}\mathbf{P}\mathbf{Q}^T$ is the action of the linear group on $\mathrm{Sym}_n^+$, and $\mathbf{Q} \star \mathbf{A}_1 = \mathbf{Q}\mathbf{A}_1\mathbf{Q}^T$ is the action of the linear group on $\mathrm{Sym}_n$[9]. The dot product at the identity is defined as $< \mathbf{A}_1 | \mathbf{A}_2 > = \mathrm{Trace}(\mathbf{A}_1\mathbf{A}_2) + \beta \mathrm{Trace}(\mathbf{A}_1) \mathrm{Trace}(\mathbf{A}_2)$ with $\beta > -\frac{1}{n}$.

The exponential map at a point can be obtained [61] as

$$\mathrm{Exp}_\mathbf{P}(\mathbf{A}) = \mathbf{P}^{\frac{1}{2}} \exp\left(\mathbf{P}^{-\frac{1}{2}}\mathbf{A}\mathbf{P}^{-\frac{1}{2}}\right)\mathbf{P}^{\frac{1}{2}}, \tag{15}$$

where $\mathbf{P} \in \mathrm{Sym}_n^+$, $\mathbf{A} \in T_\mathbf{P} \mathrm{Sym}_n^+$. By inverting the exponential map, one obtains the logarithmic map

$$\mathrm{Log}_\mathbf{P}(\mathbf{Q}) = \mathbf{P}^{\frac{1}{2}} \log\left(\mathbf{P}^{-\frac{1}{2}}\mathbf{Q}\mathbf{P}^{-\frac{1}{2}}\right)\mathbf{P}^{\frac{1}{2}}, \tag{16}$$

where $\mathbf{P}, \mathbf{Q} \in \mathrm{Sym}_n^+$. The parallel transport of a tangent vector $\mathbf{A} \in T_\mathbf{P} \mathrm{Sym}_n^+$ from $\mathbf{P}$ to $\mathbf{Q}$ along geodesics joining $\mathbf{P}$ and $\mathbf{Q}$ is given [66] by

$$\mathcal{T}_{\mathbf{P}\to\mathbf{Q}}(\mathbf{A}) = (\mathbf{Q}\mathbf{P}^{-1})^{\frac{1}{2}}\mathbf{A}\left((\mathbf{Q}\mathbf{P}^{-1})^{\frac{1}{2}}\right)^T. \tag{17}$$

**Log-Euclidean Metrics** Arsigny et al. [3] shown that the space of SPD matrices can be given a commutative Lie group structure by endowing it with the LE metric described as

$$< \mathbf{A}_1 | \mathbf{A}_2 >_\mathbf{P} = < D_\mathbf{P} \log(\mathbf{A}_1) | D_\mathbf{P} \log(\mathbf{A}_2) >_\mathbf{I},$$

where $\mathbf{P} \in \mathrm{Sym}_n^+$, $\mathbf{A}_1$ and $\mathbf{A}_2 \in \mathrm{Sym}_n$, $D_\mathbf{P} \log(\mathbf{A}_1)$ and $D_\mathbf{P} \log(\mathbf{A}_2)$ are respectively the differentials of the matrix logarithm at $\mathbf{P}$ along tangent vectors $\mathbf{A}_1$ and $\mathbf{A}_2$, and $< . | . >_\mathbf{I}$ is any metric at the tangent space at $\mathbf{I}$.

One can derive [3] the Riemannian exponential and logarithmic maps at any point as

$$\mathrm{Exp}_\mathbf{P}(\mathbf{A}) = \exp(\log(\mathbf{P}) + D_\mathbf{P} \log(\mathbf{A})), \tag{18}$$

$$\mathrm{Log}_\mathbf{P}(\mathbf{Q}) = D_{\log(\mathbf{P})} \exp(\log(\mathbf{Q}) - \log(\mathbf{P})), \tag{19}$$

where $\mathbf{P}, \mathbf{Q} \in \mathrm{Sym}_n^+$, $\mathbf{A} \in T_\mathbf{P} \mathrm{Sym}_n^+$.

While the LE metric does not yield full affine-invariance, it shares very similar properties with the AI metric. Moreover, it allows to turn Riemannian computations into Euclidean computations in the logarithmic domain that is attractive in terms of computational efficiency.

## 11 Our Theoretical Results on Gyrovector Spaces of SPD Matrices

### 11.1 AI Gyrovector Spaces

We show a hidden analogy between AI gyrovector spaces and Euclidean spaces.

**Lemma 11.1.** *Let* $\mathbf{P}_0, \mathbf{P}_1, \mathbf{Q}_0, \mathbf{Q}_1 \in \mathrm{Sym}_n^+$ *such that* $\mathrm{Log}_{\mathbf{P}_1^r}^{ai}(\mathbf{Q}_1^r)$ *is the AI parallel transport of* $\mathrm{Log}_{\mathbf{P}_0^r}^{ai}(\mathbf{Q}_0^r)$ *from* $\mathbf{P}_0^r$ *to* $\mathbf{P}_1^r$ *along geodesics connecting* $\mathbf{P}_0^r$ *and* $\mathbf{P}_1^r$. *Then*

$$\ominus_{ai}^r \mathbf{P}_1 \oplus_{ai}^r \mathbf{Q}_1 = \mathrm{gyr}_{ai}^r[\mathbf{P}_1, \ominus_{ai}^r \mathbf{P}_0](\ominus_{ai}^r \mathbf{P}_0 \oplus_{ai}^r \mathbf{Q}_0).$$

---

[9]Indeed, the action of the linear group on $\mathrm{Sym}_n^+$ is naturally extended to tangent vectors [61].

*Proof.* By assumption that $\text{Log}_{\mathbf{P}_1^r}^{ai} \mathbf{Q}_1^r$ is the AI parallel transport of $\text{Log}_{\mathbf{P}_0^r}^{ai} \mathbf{Q}_0^r$ from $\mathbf{P}_0^r$ to $\mathbf{P}_1^r$ along geodesics connecting $\mathbf{P}_0^r$ and $\mathbf{P}_1^r$, then from Eq. (17),

$$\text{Log}_{\mathbf{P}_1^r}^{ai}(\mathbf{Q}_1^r) = \mathbf{R}\,\text{Log}_{\mathbf{P}_0^r}^{ai}(\mathbf{Q}_0^r)\mathbf{R}^T,$$

where $\mathbf{R} = (\mathbf{P}_1^r\mathbf{P}_0^{-r})^{\frac{1}{2}}$.

Thus

$$\begin{aligned}
\mathbf{Q}_1^r &= \text{Exp}_{\mathbf{P}_1^r}^{ai}(\mathbf{R}\,\text{Log}_{\mathbf{P}_0^r}^{ai}(\mathbf{Q}_0^r)\mathbf{R}^T)\\
&= \mathbf{P}_1^{\frac{r}{2}}\exp(\mathbf{P}_1^{-\frac{r}{2}}\mathbf{R}\,\text{Log}_{\mathbf{P}_0^r}^{ai}(\mathbf{Q}_0^r)\mathbf{R}^T\mathbf{P}_1^{-\frac{r}{2}})\mathbf{P}_1^{\frac{r}{2}}.
\end{aligned} \tag{20}$$

Note that

$$\text{Log}_{\mathbf{P}_0^r}^{ai}(\mathbf{Q}_0^r) = \mathbf{P}_0^{\frac{r}{2}}\log(\mathbf{P}_0^{-\frac{r}{2}}\mathbf{Q}_0^r\mathbf{P}_0^{-\frac{r}{2}})\mathbf{P}_0^{\frac{r}{2}}.$$

Hence from Eq. (20),

$$\begin{aligned}
\mathbf{Q}_1^r &= \mathbf{P}_1^{\frac{r}{2}}\exp(\mathbf{U}\log(\mathbf{P}_0^{-\frac{r}{2}}\mathbf{Q}_0^r\mathbf{P}_0^{-\frac{r}{2}})\mathbf{U}^T)\mathbf{P}_1^{\frac{r}{2}}\\
&\overset{(1)}{=} \mathbf{P}_1^{\frac{r}{2}}\mathbf{U}\exp(\log(\mathbf{P}_0^{-\frac{r}{2}}\mathbf{Q}_0^r\mathbf{P}_0^{-\frac{r}{2}}))\mathbf{U}^T\mathbf{P}_1^{\frac{r}{2}}\\
&= \mathbf{P}_1^{\frac{r}{2}}\mathbf{U}\mathbf{P}_0^{-\frac{r}{2}}\mathbf{Q}_0^r\mathbf{P}_0^{-\frac{r}{2}}\mathbf{U}^T\mathbf{P}_1^{\frac{r}{2}}\\
&= \mathbf{R}\mathbf{Q}_0^r\mathbf{R}^T = (\mathbf{P}_1^r\mathbf{P}_0^{-r})^{\frac{1}{2}}\mathbf{Q}_0^r((\mathbf{P}_1^r\mathbf{P}_0^{-r})^{\frac{1}{2}})^T,
\end{aligned} \tag{21}$$

where $\mathbf{U} = \mathbf{P}_1^{-\frac{r}{2}}\mathbf{R}\mathbf{P}_0^{\frac{r}{2}}$, and (1) follows from the fact that $\mathbf{U}\mathbf{U}^T = \mathbf{I}_n$.

From Eq. (5),

$$\text{gyr}_{ai}^r[\mathbf{P}_1, \ominus_{ai}^r\mathbf{P}_0](\ominus_{ai}^r\mathbf{P}_0 \oplus_{ai}^r \mathbf{Q}_0) = \left((\mathbf{P}_1^{\frac{r}{2}}\mathbf{P}_0^{-r}\mathbf{P}_1^{\frac{r}{2}})^{-\frac{1}{2}}\mathbf{P}_1^{\frac{r}{2}}\mathbf{P}_0^{-r}\mathbf{Q}_0^r\mathbf{P}_1^{-\frac{r}{2}}(\mathbf{P}_1^{\frac{r}{2}}\mathbf{P}_0^{-r}\mathbf{P}_1^{\frac{r}{2}})^{\frac{1}{2}}\right)^{\frac{1}{r}}.$$

Let $\mathbf{B} = (\mathbf{P}_1^{\frac{r}{2}}\mathbf{P}_0^{-r}\mathbf{P}_1^{\frac{r}{2}})^{-\frac{1}{2}}\mathbf{P}_1^{\frac{r}{2}}\mathbf{P}_0^{-r}$, $\mathbf{C} = \mathbf{P}_1^{-\frac{r}{2}}(\mathbf{P}_1^{\frac{r}{2}}\mathbf{P}_0^{-r}\mathbf{P}_1^{\frac{r}{2}})^{\frac{1}{2}}$. Then

$$\text{gyr}_{ai}^r[\mathbf{P}_1, \ominus_{ai}^r\mathbf{P}_0](\ominus_{ai}^r\mathbf{P}_0 \oplus_{ai}^r \mathbf{Q}_0) = (\mathbf{B}\mathbf{Q}_0^r\mathbf{C})^{\frac{1}{r}}.$$

Hence

$$\mathbf{P}_1^{\frac{r}{2}}\left(\text{gyr}_{ai}^r[\mathbf{P}_1, \ominus_{ai}^r\mathbf{P}_0](\ominus_{ai}^r\mathbf{P}_0 \oplus_{ai}^r \mathbf{Q}_0)\right)^r\mathbf{P}_1^{\frac{r}{2}} = \mathbf{P}_1^{\frac{r}{2}}\mathbf{B}\mathbf{Q}_0^r\mathbf{C}\mathbf{P}_1^{\frac{r}{2}}. \tag{22}$$

We remark that

$$\begin{aligned}
(\mathbf{P}_1^{\frac{r}{2}}\mathbf{B})^2 &= \mathbf{P}_1^{\frac{r}{2}}\mathbf{B}\mathbf{P}_1^{\frac{r}{2}}\mathbf{B} = \mathbf{P}_1^r\mathbf{P}_0^{-r},\\
(\mathbf{C}\mathbf{P}_1^{\frac{r}{2}})^2 &= \mathbf{C}\mathbf{P}_1^{\frac{r}{2}}\mathbf{C}\mathbf{P}_1^{\frac{r}{2}} = (\mathbf{P}_1^r\mathbf{P}_0^{-r})^T.
\end{aligned}$$

Therefore

$$\mathbf{P}_1^{\frac{r}{2}}\mathbf{B}\mathbf{Q}_0^r\mathbf{C}\mathbf{P}_1^{\frac{r}{2}} = (\mathbf{P}_1^r\mathbf{P}_0^{-r})^{\frac{1}{2}}\mathbf{Q}_0^r((\mathbf{P}_1^r\mathbf{P}_0^{-r})^{\frac{1}{2}})^T. \tag{23}$$

Combining Eqs. (21), (22), and (23), we obtain

$$\mathbf{Q}_1^r = \mathbf{P}_1^{\frac{r}{2}}\left(\text{gyr}_{ai}^r[\mathbf{P}_1, \ominus_{ai}^r\mathbf{P}_0](\ominus_{ai}^r\mathbf{P}_0 \oplus_{ai}^r \mathbf{Q}_0)\right)^r\mathbf{P}_1^{\frac{r}{2}},$$

which leads to

$$\ominus_{ai}^r\mathbf{P}_1 \oplus_{ai}^r \mathbf{Q}_1 = \text{gyr}_{ai}^r[\mathbf{P}_1, \ominus_{ai}^r\mathbf{P}_0](\ominus_{ai}^r\mathbf{P}_0 \oplus_{ai}^r \mathbf{Q}_0).$$

$\square$

Lemma 11.1 reveals a strong link between the AI geometry of SPD manifolds and hyperbolic geometry, as the algebraic definition [44] of parallel transport in a gyrovector space agrees with the classical parallel transport of differential geometry. In the gyrolanguage [44, 45, 46], Lemma 11.1 states that the gyrovector $\ominus_{ai}^r\mathbf{P}_1 \oplus_{ai}^r \mathbf{Q}_1$ is the gyrovector $\ominus_{ai}^r\mathbf{P}_0 \oplus_{ai}^r \mathbf{Q}_0$ gyrated by a gyroautomorphism. This gives a characterization of the AI parallel transport that is fully analogous to that of the parallel transport in Euclidean spaces. Note that this characterization also agrees with the reinterpretation of addition and subtraction in a Riemannian manifold using logarithmic and exponential maps [61]. Thus the gyrolanguage is a powerful tool for uncovering analogies that the AI geometry of SPD manifolds shares with Euclidean geometry.

## 11.2 LE Gyrovector Spaces

A corresponding result of Lemma 11.1 for LE gyrovector spaces can also be established.

**Lemma 11.2.** *Let* $\mathbf{P}_0, \mathbf{P}_1, \mathbf{Q}_0, \mathbf{Q}_1 \in \mathrm{Sym}_n^+$ *such that* $\mathrm{Log}_{\mathbf{P}_1^r}^{le}(\mathbf{Q}_1^r)$ *is the LE parallel transport of* $\mathrm{Log}_{\mathbf{P}_0^r}^{le}(\mathbf{Q}_0^r)$ *from* $\mathbf{P}_0^r$ *to* $\mathbf{P}_1^r$ *along geodesics connecting* $\mathbf{P}_0^r$ *and* $\mathbf{P}_1^r$. *Then*

$$\ominus_{le}^r \mathbf{P}_1 \oplus_{le}^r \mathbf{Q}_1 = \mathrm{gyr}_{le}^r[\mathbf{P}_1, \ominus_{le}^r \mathbf{P}_0](\ominus_{le}^r \mathbf{P}_0 \oplus_{le}^r \mathbf{Q}_0).$$

*Proof.* Notice that

$$\begin{aligned}
\mathbf{Q}_0^r &= \mathrm{Exp}_{\mathbf{P}_0^r}^{le}(\mathrm{Log}_{\mathbf{P}_0^r}^{le}(\mathbf{Q}_0^r)) \\
&\overset{(1)}{=} \exp\left(\log(\mathbf{P}_0^r) + D_{\mathbf{P}_0^r}\log\left(\mathrm{Log}_{\mathbf{P}_0^r}^{le}(\mathbf{Q}_0^r)\right)\right) \\
&\overset{(2)}{=} \exp\left(\log(\mathbf{P}_0^r) + \mathcal{T}_{\mathbf{P}_0^r \to \mathbf{I}_n}(\mathrm{Log}_{\mathbf{P}_0^r}^{le}(\mathbf{Q}_0^r))\right),
\end{aligned}$$

where (1) and (2) follow respectively from Eqs. (18) and (30).

It is known [57] that a SPD matrix has a unique symmetric logarithm, and since $\log(\mathbf{P}_0^r) + \mathcal{T}_{\mathbf{P}_0^r \to \mathbf{I}_n}(\mathrm{Log}_{\mathbf{P}_0^r}^{le}(\mathbf{Q}_0^r))$ is symmetric, we have

$$\log(\mathbf{Q}_0^r) = \log(\mathbf{P}_0^r) + \mathcal{T}_{\mathbf{P}_0^r \to \mathbf{I}_n}(\mathrm{Log}_{\mathbf{P}_0^r}^{le}(\mathbf{Q}_0^r)).$$

We thus get

$$\log(\mathbf{Q}_0^r) - \log(\mathbf{P}_0^r) = \mathcal{T}_{\mathbf{P}_0^r \to \mathbf{I}_n}(\mathrm{Log}_{\mathbf{P}_0^r}^{le}(\mathbf{Q}_0^r)). \tag{24}$$

Similarly, we have

$$\log(\mathbf{Q}_1^r) - \log(\mathbf{P}_1^r) = \mathcal{T}_{\mathbf{P}_1^r \to \mathbf{I}_n}(\mathrm{Log}_{\mathbf{P}_1^r}^{le}(\mathbf{Q}_1^r)). \tag{25}$$

Since $\mathrm{Syn}_n^+$ are complete, simply-connected and flat manifolds [3], the parallel transport is path independent. By the assumption that $\mathrm{Log}_{\mathbf{P}_1^r}^{le}(\mathbf{Q}_1^r)$ is the LE parallel transport of $\mathrm{Log}_{\mathbf{P}_0^r}^{le}(\mathbf{Q}_0^r)$ from $\mathbf{P}_0^r$ to $\mathbf{P}_1^r$ along geodesics connecting $\mathbf{P}_0^r$ and $\mathbf{P}_1^r$, we deduce that

$$\mathcal{T}_{\mathbf{P}_0^r \to \mathbf{I}_n}(\mathrm{Log}_{\mathbf{P}_0^r}^{le}(\mathbf{Q}_0^r)) = \mathcal{T}_{\mathbf{P}_1^r \to \mathbf{I}_n}(\mathrm{Log}_{\mathbf{P}_1^r}^{le}(\mathbf{Q}_1^r)). \tag{26}$$

Combining Eqs. (24), (25), and (26) results in

$$\log(\mathbf{Q}_0^r) - \log(\mathbf{P}_0^r) = \log(\mathbf{Q}_1^r) - \log(\mathbf{P}_1^r),$$

which leads to

$$\ominus_{le}^r \mathbf{P}_1 \oplus_{le}^r \mathbf{Q}_1 = \ominus_{le}^r \mathbf{P}_0 \oplus_{le}^r \mathbf{Q}_0.$$

Therefore

$$\ominus_{le}^r \mathbf{P}_1 \oplus_{le}^r \mathbf{Q}_1 = \mathrm{gyr}_{le}^r[\mathbf{P}_1, \ominus_{le}^r \mathbf{P}_0](\ominus_{le}^r \mathbf{P}_0 \oplus_{le}^r \mathbf{Q}_0).$$

$\square$

Lemma 11.2 gives a characterization of the LE parallel transport that is fully analogous to that of the parallel transport in Euclidean spaces. This result is not surprising and stems from the fact that the space of SPD matrices with the LE geometry has a vector space structure. Lemmas 11.1 and 11.2 point out a strong link between the AI and LE geometries of SPD manifolds and hyperbolic geometry.

## 12  Proof of Lemma 3.3

*Proof.* Using Eqs. (15), (16), and (17) leads to the conclusion of the Lemma. $\square$

## 13  Proof of Lemma 3.4

*Proof.* Using Eqs. (15) and (16), it is straightforward to see that

$$t \otimes_{ai} \mathbf{P} = \mathrm{Exp}_{\mathbf{I}_n}^{ai}(t\, \mathrm{Log}_{\mathbf{I}_n}^{ai}(\mathbf{P})) = \exp(\log(\mathbf{P}))^t = \mathbf{P}^t.$$

$\square$

# 14 Proof of Theorem 3.6

*Proof.* First, note that the binary operation $\oplus^r_{ai}$ verifies the Left Cancellation Law [44, 45, 46], i.e.,
$$\ominus^r_{ai}\mathbf{P} \oplus^r_{ai} (\mathbf{P} \oplus^r_{ai} \mathbf{Q}) = \mathbf{Q},$$
for any $\mathbf{P}, \mathbf{Q} \in \mathrm{Sym}_n^+$.

The gyroautomorphism can be determined from the binary operation as in [44, 45, 46]. By axiom (G3) and the Left Cancellation Law,
$$\mathrm{gyr}^r_{ai}[\mathbf{P}, \mathbf{Q}]\mathbf{R} = \big( \ominus^r_{ai} (\mathbf{P} \oplus^r_{ai} \mathbf{Q})\big) \oplus^r_{ai} \big(\mathbf{P} \oplus^r_{ai} (\mathbf{Q} \oplus^r_{ai} \mathbf{R})\big). \tag{27}$$

Using the expression of the binary operation $\oplus^r_{ai}$ given in Lemma 3.3, we can deduce that
$$\mathrm{gyr}^r_{ai}[\mathbf{P}, \mathbf{Q}]\mathbf{R} = \big((\mathbf{P}^{\frac{r}{2}}\mathbf{Q}^r\mathbf{P}^{\frac{r}{2}})^{-\frac{1}{2}}\mathbf{P}^{\frac{r}{2}}\mathbf{Q}^{\frac{r}{2}}\mathbf{R}^r\mathbf{Q}^{\frac{r}{2}}\mathbf{P}^{\frac{r}{2}}(\mathbf{P}^{\frac{r}{2}}\mathbf{Q}^r\mathbf{P}^{\frac{r}{2}})^{-\frac{1}{2}}\big)^{\frac{1}{r}}.$$

Let $F^r_{ai}(\mathbf{P}, \mathbf{Q}) = (\mathbf{P}^{\frac{r}{2}}\mathbf{Q}^r\mathbf{P}^{\frac{r}{2}})^{-\frac{1}{2}}\mathbf{P}^{\frac{r}{2}}\mathbf{Q}^{\frac{r}{2}}$. Then
$$F^r_{ai}(\mathbf{P}, \mathbf{Q})\mathbf{Q}^{\frac{r}{2}}\mathbf{P}^{\frac{r}{2}}(\mathbf{P}^{\frac{r}{2}}\mathbf{Q}^r\mathbf{P}^{\frac{r}{2}})^{-\frac{1}{2}} = (\mathbf{P}^{\frac{r}{2}}\mathbf{Q}^r\mathbf{P}^{\frac{r}{2}})^{-\frac{1}{2}}\mathbf{P}^{\frac{r}{2}}\mathbf{Q}^r\mathbf{P}^{\frac{r}{2}}(\mathbf{P}^{\frac{r}{2}}\mathbf{Q}^r\mathbf{P}^{\frac{r}{2}})^{-\frac{1}{2}} = \mathbf{I}_n.$$

Therefore
$$\mathrm{gyr}^r_{ai}[\mathbf{P}, \mathbf{Q}]\mathbf{R} = \big(F^r_{ai}(\mathbf{P}, \mathbf{Q})\mathbf{R}^r(F^r_{ai}(\mathbf{P}, \mathbf{Q}))^{-1}\big)^{\frac{1}{r}}.$$

It is then easy to verify axioms G1,G2,G4,V1,V2,V3,V4,V5 for AI gyrovector spaces. $\qquad\square$

# 15 Proof of Lemma 3.7

*Proof.* Let $L$ be the left translation defined as
$$L_{\mathbf{P}}(\mathbf{Q}) = \exp(\log(\mathbf{P}) + \log(\mathbf{Q})).$$

Since the LE metric is a bi-invariant metric, the Levi-Civita connection coincides with the Cartan connection and the parallel transport of a tangent vector $\mathbf{V} \in T_{\mathbf{P}}\,\mathrm{Sym}_n^+$ is induced by the left translation [54, 58, 63], i.e.,
$$\mathcal{T}^{le}_{\mathbf{P}\to\mathbf{Q}}(\mathbf{V}) = D_{\mathbf{P}}L_{\mathbf{Q}\mathbf{P}^{-1}}(\mathbf{V}).$$

Thus, when $\mathbf{Q} = \mathbf{I}_n$,
$$\mathcal{T}^{le}_{\mathbf{P}\to\mathbf{I}_n}(\mathbf{V}) = D_{\mathbf{P}}L_{\mathbf{P}^{-1}}(\mathbf{V}). \tag{28}$$

Note that
$$(\log \circ L_{\mathbf{P}^{-1}})(\mathbf{R}) = \log(\mathbf{P}^{-1}) + \log(\mathbf{R}).$$

Hence
$$D_{\exp(\log(\mathbf{P}^{-1})+\log(\mathbf{R}))} \log \circ D_{\mathbf{R}}L_{\mathbf{P}^{-1}} = D_{\mathbf{R}} \log.$$

When $\mathbf{R} = \mathbf{P}$, we get
$$D_{\mathbf{P}}L_{\mathbf{P}^{-1}} = D_{\mathbf{P}} \log. \tag{29}$$

Combining Eqs. (28) and (29) leads to
$$\mathcal{T}^{le}_{\mathbf{P}\to\mathbf{I}_n}(\mathbf{V}) = D_{\mathbf{P}} \log(\mathbf{V}). \tag{30}$$

Let $\mathcal{T}^{le}_{\mathbf{I}_n\to\mathbf{P}}(\mathrm{Log}^{le}_{\mathbf{I}_n}(\mathbf{Q})) = \mathbf{V}$. Then $\mathcal{T}^{le}_{\mathbf{P}\to\mathbf{I}_n}(\mathbf{V}) = \mathrm{Log}^{le}_{\mathbf{I}_n}(\mathbf{Q}) = \log(\mathbf{Q})$. From Eq. (30), we get
$$D_{\mathbf{P}} \log(\mathbf{V}) = \log(\mathbf{Q}).$$

Then, from Eq. (18),
$$\mathrm{Exp}^{le}_{\mathbf{P}}(\mathbf{V}) = \exp(\log(\mathbf{P}) + D_{\mathbf{P}} \log(\mathbf{V})),$$
which results in
$$\mathrm{Exp}^{le}_{\mathbf{P}}(\mathbf{V}) = \exp(\log(\mathbf{P}) + \log(\mathbf{Q})).$$

We thus have
$$\mathrm{Exp}^{le}_{\mathbf{P}}(\mathcal{T}^{le}_{\mathbf{I}_n\to\mathbf{P}}(\mathrm{Log}^{le}_{\mathbf{I}_n}(\mathbf{Q}))) = \exp(\log(\mathbf{P}) + \log(\mathbf{Q})).$$

Therefore
$$\mathbf{P} \oplus^r_{le} \mathbf{Q} = \big(\exp(\log(\mathbf{P}^r) + \log(\mathbf{Q}^r))\big)^{\frac{1}{r}}.$$

$\qquad\square$

# 16 Proof of Theorem 3.10

*Proof.* First, note that the binary operation $\oplus_{le}^r$ verifies the Left Cancellation Law. From Eq. (27),

$$
\begin{aligned}
\text{gyr}_{le}^r[\mathbf{P}, \mathbf{Q}]\mathbf{R} &= \left( \ominus_{le}^r \left( \mathbf{P} \oplus_{le}^r \mathbf{Q} \right) \right) \oplus_{le}^r \left( \mathbf{P} \oplus_{le}^r \left( \mathbf{Q} \oplus_{le}^r \mathbf{R} \right) \right) \\
&\overset{(1)}{=} \left( \ominus_{le}^r \left( \mathbf{P} \oplus_{le}^r \mathbf{Q} \right) \right) \oplus_{le}^r \left( \left( \mathbf{P} \oplus_{le}^r \mathbf{Q} \right) \oplus_{le}^r \mathbf{R} \right) \\
&\overset{(2)}{=} \mathbf{R}.
\end{aligned}
\tag{31}
$$

The derivation of Eq. (31) follows.

(1) follows from the associativity of the binary operation $\oplus_{le}^r$.

(2) follows from the Left Cancellation Law.

It is then easy to verify axioms G1,G2,G4,V1,V2,V3,V4,V5 for LE gyrovector spaces.

$\square$

# 17 Riemannian Geometry of Grassmann Manifolds

The Grassmann manifold $\text{Gr}_{n,p}$ (also called Grassmannian) [48, 49, 53] is defined as the set of all p-dimensional subspaces of the Euclidean space $\mathbb{R}^n$, i.e.,

$$
\text{Gr}_{n,p} = \{ \mathcal{U} \in \mathbb{R}^n | \mathcal{U} \text{ is a subspace}, \dim(\mathcal{U}) = p \}.
$$

This set can be identified with the set of orthogonal rank-p projectors

$$
\text{Gr}_{n,p} = \{ \mathbf{P} \in M_{n,n} | \mathbf{P} = \mathbf{P}^T, \mathbf{P}^2 = \mathbf{P}, \text{rank}(\mathbf{P}) = p \}.
$$

Let $\mathbf{P} \in \text{Gr}_{n,p}$ and $\Delta \in T_{\mathbf{P}} \text{Gr}_{n,p}$, the exponential map is given by

$$
\text{Exp}_{\mathbf{P}}^{gr}(\Delta) = \exp([\Delta, \mathbf{P}]) \mathbf{P} \exp(-[\Delta, \mathbf{P}]).
\tag{32}
$$

To define the logarithmic map on $\text{Gr}_{n,p}$, we need to define the cut locus of a point on $\text{Gr}_{n,p}$. Let $\gamma_\Delta : t \to \text{Exp}_{\mathbf{P}}^{gr}(t\Delta)$. The cut time of $(\mathbf{P}, \Delta)$ is defined as

$$
t_{cut}(\mathbf{P}, \Delta) := \sup\{ b > 0 | \text{the restriction of } \gamma_\Delta \text{ to } [0, b] \text{ is minimizing} \}.
$$

The cut point of $\mathbf{P}$ along $\gamma_\Delta$ is given by $\gamma_\Delta(t_{cut}(\mathbf{P}, \Delta))$, and the cut locus of $\mathbf{P}$ is defined as

$$
\text{Cut}_{\mathbf{P}} := \{ \mathbf{F} \in \text{Gr}_{n,p} | F = \gamma_\Delta(t_{cut}(\mathbf{P}, \Delta)) \text{ for some } \Delta \in T_{\mathbf{P}} \text{Gr}_{n,p} \}.
$$

The cut locus of $\mathbf{P} = \mathbf{U}\mathbf{U}^T \in \text{Gr}_{n,p}$ is the set of all subspaces with at least one direction orthogonal to all directions in the subspace onto which $\mathbf{P}$ projects [67], i.e.,

$$
\text{Cut}_{\mathbf{P}} = \{ \mathbf{F} = \mathbf{Y}\mathbf{Y}^T \in \text{Gr}_{n,p} | \text{rank}(\mathbf{U}^T\mathbf{Y}) < p \}.
$$

The injectivity domain of $\mathbf{P}$ is defined as

$$
\text{ID}_{\mathbf{P}} := \{ \Delta \in T_{\mathbf{P}} \text{Gr}_{n,p} | \|\Delta\| < t_{cut}(\mathbf{P}, \Delta/\|\Delta\|) \}.
$$

It has been shown [65] that two points are in each other's cut locus if there is more than one shortest geodesic joining them. When restricting $\text{Exp}_{\mathbf{P}}^{gr}$ to the injectivity domain $\text{ID}_{\mathbf{P}}$, there is a unique tangent vector $\Delta \in \text{ID}_{\mathbf{P}} \subset T_{\mathbf{P}} \text{Gr}_{n,p}$ such that $\text{Exp}_{\mathbf{P}}^{gr}(\Delta) = \mathbf{F}$, for any $\mathbf{F} \in \text{Gr}_{n,p} \setminus \text{Cut}_{\mathbf{P}}$. For such a point $\mathbf{F}$, the logarithmic map is given as

$$
\text{Log}_{\mathbf{P}}^{gr}(\mathbf{F}) = [\Omega, \mathbf{P}],
\tag{33}
$$

where $[\Omega, \mathbf{P}] = \Delta \in \text{ID}_{\mathbf{P}}$, and $\Omega$ is computed by

$$
\Omega = [\Delta, \mathbf{P}] = \frac{1}{2} \log \left( (\mathbf{I}_n - 2\mathbf{F})(\mathbf{I}_n - 2\mathbf{P}) \right).
\tag{34}
$$

Let $\Gamma \in T_{\mathbf{P}} \operatorname{Gr}_{n,p}$, the parallel transport of $\Delta$ along the geodesic
$$\operatorname{Exp}_{\mathbf{P}}^{gr}(t\Gamma) = \exp(t[\Gamma, \mathbf{P}])\mathbf{P}\exp(-t[\Gamma, \mathbf{P}])$$
is given by
$$\mathcal{T}_{\mathbf{P}\to\mathbf{Q}}^{gr} = \exp(t[\Gamma, \mathbf{P}])\Delta\exp(-t[\Gamma, \mathbf{P}]), \tag{35}$$
where $\mathbf{Q} = \operatorname{Exp}_{\mathbf{P}}^{gr}(t\Gamma) \in \operatorname{Gr}_{n,p}$.

The following identities, obtained respectively from Eqs. (32) and (34), will be used extensively in our proofs:

$$\mathbf{P} = \exp([\overline{\mathbf{P}}, \mathbf{I}_{n,p}])\mathbf{I}_{n,p}\exp(-[\overline{\mathbf{P}}, \mathbf{I}_{n,p}]). \tag{36}$$

$$[\overline{\mathbf{P}}, \mathbf{I}_{n,p}] = \frac{1}{2}\log\big((\mathbf{I}_n - 2\mathbf{P})(\mathbf{I}_n - 2\mathbf{I}_{n,p})\big). \tag{37}$$

# 18 Proof of Lemma 3.11

*Proof.* Let $\tilde{\mathbf{Q}} = \mathcal{T}_{\mathbf{I}_{n,p}\to\mathbf{P}}(\operatorname{Log}_{\mathbf{I}_{n,p}}^{gr}(\mathbf{Q}))$. From Eq. (35),
$$\tilde{\mathbf{Q}} = \exp([\overline{\mathbf{P}}, \mathbf{I}_{n,p}])\overline{\mathbf{Q}}\exp(-[\overline{\mathbf{P}}, \mathbf{I}_{n,p}]).$$

Hence
$$\begin{aligned}
\mathbf{P} \oplus_{gr} \mathbf{Q} &= \operatorname{Exp}_{\mathbf{P}}^{gr}(\tilde{\mathbf{Q}}) \\
&\overset{(1)}{=} \exp([\tilde{\mathbf{Q}}, \mathbf{P}])\mathbf{P}\exp(-[\tilde{\mathbf{Q}}, \mathbf{P}]) \\
&\overset{(2)}{=} \exp([\tilde{\mathbf{Q}}, \mathbf{P}])\exp([\overline{\mathbf{P}}, \mathbf{I}_{n,p}])\mathbf{I}_{n,p}\exp(-[\overline{\mathbf{P}}, \mathbf{I}_{n,p}])\exp(-[\tilde{\mathbf{Q}}, \mathbf{P}]),
\end{aligned} \tag{38}$$
where (1) and (2) follow respectively from Eqs. (32) and (36).

Note that
$$\begin{aligned}
[\tilde{\mathbf{Q}}, \mathbf{P}] &\overset{(1)}{=} [\exp([\overline{\mathbf{P}}, \mathbf{I}_{n,p}])\overline{\mathbf{Q}}\exp(-[\overline{\mathbf{P}}, \mathbf{I}_{n,p}]), \exp([\overline{\mathbf{P}}, \mathbf{I}_{n,p}])\mathbf{I}_{n,p}\exp(-[\overline{\mathbf{P}}, \mathbf{I}_{n,p}])] \\
&= \exp([\overline{\mathbf{P}}, \mathbf{I}_{n,p}])\overline{\mathbf{Q}}\mathbf{I}_{n,p}\exp(-[\overline{\mathbf{P}}, \mathbf{I}_{n,p}]) - \exp([\overline{\mathbf{P}}, \mathbf{I}_{n,p}])\mathbf{I}_{n,p}\overline{\mathbf{Q}}\exp(-[\overline{\mathbf{P}}, \mathbf{I}_{n,p}]) \\
&= \exp([\overline{\mathbf{P}}, \mathbf{I}_{n,p}])[\overline{\mathbf{Q}}, \mathbf{I}_{n,p}]\exp(-[\overline{\mathbf{P}}, \mathbf{I}_{n,p}]),
\end{aligned}$$
where (1) follows from Eq. (36).

Thus
$$\begin{aligned}
\exp([\tilde{\mathbf{Q}}, \mathbf{P}]) &= \exp\big(\exp([\overline{\mathbf{P}}, \mathbf{I}_{n,p}])[\overline{\mathbf{Q}}, \mathbf{I}_{n,p}]\exp(-[\overline{\mathbf{P}}, \mathbf{I}_{n,p}])\big) \\
&\overset{(1)}{=} \exp([\overline{\mathbf{P}}, \mathbf{I}_{n,p}])\exp([\overline{\mathbf{Q}}, \mathbf{I}_{n,p}])\exp(-[\overline{\mathbf{P}}, \mathbf{I}_{n,p}]),
\end{aligned} \tag{39}$$
where (1) follows from the fact that $\exp([\overline{\mathbf{P}}, \mathbf{I}_{n,p}])\exp(-[\overline{\mathbf{P}}, \mathbf{I}_{n,p}]) = \mathbf{I}_n$.

Combining Eqs. (38) and (39), we get
$$\begin{aligned}
\mathbf{P} \oplus_{gr} \mathbf{Q} &= \exp([\overline{\mathbf{P}}, \mathbf{I}_{n,p}])\exp([\overline{\mathbf{Q}}, \mathbf{I}_{n,p}])\mathbf{I}_{n,p}\exp(-[\overline{\mathbf{Q}}, \mathbf{I}_{n,p}])\exp(-[\overline{\mathbf{P}}, \mathbf{I}_{n,p}]) \\
&\overset{(1)}{=} \exp([\overline{\mathbf{P}}, \mathbf{I}_{n,p}])\mathbf{Q}\exp(-[\overline{\mathbf{P}}, \mathbf{I}_{n,p}]),
\end{aligned}$$
where (1) follows from Eq. (36).

$\square$

# 19 Proof of Lemma 3.12

*Proof.* We have
$$\begin{aligned}
\mathbf{P} \oplus_{gr} \mathbf{Q} &\overset{(1)}{=} \exp([\overline{\mathbf{P}}, \mathbf{I}_{n,p}])\mathbf{Q}\exp(-[\overline{\mathbf{P}}, \mathbf{I}_{n,p}]) \\
&\overset{(2)}{=} \exp\left(\frac{1}{2}\log\big((\mathbf{I}_n - 2\mathbf{P})(\mathbf{I}_n - 2\mathbf{I}_{n,p})\big)\right)\mathbf{Q}\exp\left(-\frac{1}{2}\log\big((\mathbf{I}_n - 2\mathbf{P})(\mathbf{I}_n - 2\mathbf{I}_{n,p})\big)\right),
\end{aligned}$$
where (1) and (2) follow respectively from Eqs. (8) and (37).

$\square$

## 20 Proof of Lemma 3.13

*Proof.* We have

$$t \otimes \mathbf{P} = \mathrm{Exp}^{gr}_{\mathbf{I}_{n,p}}(t \, \mathrm{Log}^{gr}_{\mathbf{I}_{n,p}}(\mathbf{P}))$$
$$= \mathrm{Exp}^{gr}_{\mathbf{I}_{n,p}}(t\overline{\mathbf{P}})$$
$$\overset{(1)}{=} \exp([t\overline{\mathbf{P}}, \mathbf{I}_{n,p}])\mathbf{I}_{n,p}\exp(-[t\overline{\mathbf{P}}, \mathbf{I}_{n,p}]),$$

where (1) follows from Eq. (32).

$\square$

## 21 Proof of Lemma 3.14

*Proof.* We have

$$t \otimes \mathbf{P} \overset{(1)}{=} \exp([t\overline{\mathbf{P}}, \mathbf{I}_{n,p}])\mathbf{I}_{n,p}\exp(-[t\overline{\mathbf{P}}, \mathbf{I}_{n,p}])$$
$$= \exp(t[\overline{\mathbf{P}}, \mathbf{I}_{n,p}])\mathbf{I}_{n,p}\exp(-t[\overline{\mathbf{P}}, \mathbf{I}_{n,p}])$$
$$\overset{(2)}{=} \exp\left(\frac{t}{2}\log\left((\mathbf{I}_n - 2\mathbf{P})(\mathbf{I}_n - 2\mathbf{I}_{n,p})\right)\right)\mathbf{I}_{n,p}\exp\left(-\frac{t}{2}\log\left((\mathbf{I}_n - 2\mathbf{P})(\mathbf{I}_n - 2\mathbf{I}_{n,p})\right)\right),$$

where (1) and (2) follow respectively from Eqs. (9) and (37).

$\square$

## 22 Proof of Lemma 3.19

*Proof.* For the first identity, we have (see Section 17)

$$\mathrm{Log}^{gr}_{\mathbf{F}}(\mathbf{P}) = [\mathbf{\Omega}_1, \mathbf{F}], \mathbf{\Omega}_1 = \frac{1}{2}\log\left((\mathbf{I}_n - 2\mathbf{P})(\mathbf{I}_n - 2\mathbf{F})\right),$$
$$\mathrm{Log}^{gr}_{\mathbf{OFO}^T}(\mathbf{OPO}^T) = [\mathbf{\Omega}_2, \mathbf{OFO}^T], \mathbf{\Omega}_2 = \frac{1}{2}\log\left((\mathbf{I}_n - 2\mathbf{OPO}^T)(\mathbf{I}_n - 2\mathbf{OFO}^T)\right).$$

Notice that

$$\mathbf{\Omega}_2 = \frac{1}{2}\log\left((\mathbf{I}_n - 2\mathbf{OPO}^T)(\mathbf{I}_n - 2\mathbf{OFO}^T)\right) = \frac{1}{2}\log\left((\mathbf{OI}_n\mathbf{O}^T - 2\mathbf{OPO}^T)(\mathbf{I}_n - 2\mathbf{OFO}^T)\right)$$
$$= \frac{1}{2}\log\left(\mathbf{O}(\mathbf{I}_n - 2\mathbf{P})\mathbf{O}^T(\mathbf{I}_n - 2\mathbf{OFO}^T)\right)$$
$$= \frac{1}{2}\log\left(\mathbf{O}(\mathbf{I}_n - 2\mathbf{P})(\mathbf{O}^T\mathbf{I}_n - 2\mathbf{FO}^T)\right)$$
$$= \frac{1}{2}\log\left(\mathbf{O}(\mathbf{I}_n - 2\mathbf{P})(\mathbf{I}_n\mathbf{O}^T - 2\mathbf{FO}^T)\right)$$
$$= \frac{1}{2}\log\left(\mathbf{O}(\mathbf{I}_n - 2\mathbf{P})(\mathbf{I}_n - 2\mathbf{F})\mathbf{O}^T\right)$$
$$\overset{(1)}{=} \frac{1}{2}\mathbf{O}\log\left((\mathbf{I}_n - 2\mathbf{P})(\mathbf{I}_n - 2\mathbf{F})\right)\mathbf{O}^T$$
$$= \mathbf{O}\mathbf{\Omega}_1\mathbf{O}^T,$$

where (1) follows from the fact that matrix $\mathbf{O}$ is orthogonal and $\log\left((\mathbf{I}_n - 2\mathbf{P})(\mathbf{I}_n - 2\mathbf{F})\right)$ is defined.
Therefore

$$\mathrm{Log}^{gr}_{\mathbf{OFO}^T}(\mathbf{OPO}^T) = [\mathbf{\Omega}_2, \mathbf{OFO}^T]$$
$$= [\mathbf{O}\mathbf{\Omega}_1\mathbf{O}^T, \mathbf{OFO}^T]$$
$$= \mathbf{O}\mathbf{\Omega}_1\mathbf{FO}^T - \mathbf{OF}\mathbf{\Omega}_1\mathbf{O}^T$$
$$= \mathbf{O}[\mathbf{\Omega}_1, \mathbf{F}]\mathbf{O}^T$$
$$= \mathbf{O}\,\mathrm{Log}^{gr}_{\mathbf{F}}(\mathbf{P})\mathbf{O}^T.$$

For the second identity, note that

$$\mathrm{Exp}^{gr}_{\mathbf{OPO}^T}(\mathbf{O}\Delta\mathbf{O}^T) \overset{(1)}{=} \exp([\mathbf{O}\Delta\mathbf{O}^T, \mathbf{OPO}^T])\mathbf{OPO}^T \exp(-[\mathbf{O}\Delta\mathbf{O}^T, \mathbf{OPO}^T])$$
$$= \exp(\mathbf{O}\Delta\mathbf{PO}^T - \mathbf{OP}\Delta\mathbf{O}^T)\mathbf{OPO}^T \exp\big(-(\mathbf{O}\Delta\mathbf{PO}^T - \mathbf{OP}\Delta\mathbf{O}^T)\big)$$
$$= \exp(\mathbf{O}[\Delta, \mathbf{P}]\mathbf{O}^T)\mathbf{OPO}^T \exp(-\mathbf{O}[\Delta, \mathbf{P}]\mathbf{O}^T)$$
$$\overset{(2)}{=} \mathbf{O}\exp([\Delta, \mathbf{P}])\mathbf{O}^T\mathbf{OPO}^T\mathbf{O}\exp(-[\Delta, \mathbf{P}])\mathbf{O}^T$$
$$= \mathbf{O}\exp([\Delta, \mathbf{P}])\mathbf{P}\exp(-[\Delta, \mathbf{P}])\mathbf{O}^T$$
$$\overset{(3)}{=} \mathbf{O}\,\mathrm{Exp}^{gr}_{\mathbf{P}}(\Delta)\mathbf{O}^T.$$
$$\tag{40}$$

The derivation of Eq. (40) follows.

(1) follows from Eq. (32).

(2) follows from the fact that $\mathbf{O}$ is orthogonal.

(3) follows from Eq. (32).

The third identity can be proved by induction on $m$. First, it is easy to see that the identity holds for $m = 0$. Assuming that it holds for $m = k$, i.e.,

$$[\Delta, \mathbf{P}]^m = (\mathbf{I}_n - 2\mathbf{P})(-[\Delta, \mathbf{P}])^m(\mathbf{I}_n - 2\mathbf{P}). \tag{41}$$

Then we have

$$[\Delta, \mathbf{P}]^{m+1} = (\mathbf{I}_n - 2\mathbf{P})(-[\Delta, \mathbf{P}])^m(\mathbf{I}_n - 2\mathbf{P})[\Delta, \mathbf{P}]$$
$$= (\mathbf{I}_n - 2\mathbf{P})(-[\Delta, \mathbf{P}])^m(\mathbf{I}_n\Delta\mathbf{P} - \mathbf{I}_n\mathbf{P}\Delta - 2\mathbf{P}\Delta\mathbf{P} + 2\mathbf{P}^2\Delta)$$
$$\overset{(1)}{=} (\mathbf{I}_n - 2\mathbf{P})(-[\Delta, \mathbf{P}])^m(\Delta\mathbf{P} + \mathbf{P}\Delta - 2\mathbf{P}\Delta\mathbf{P})$$
$$\overset{(2)}{=} (\mathbf{I}_n - 2\mathbf{P})(-[\Delta, \mathbf{P}])^m(\mathbf{P}\Delta\mathbf{I}_n - 2\mathbf{P}\Delta\mathbf{P} - \Delta\mathbf{PI}_n + 2\Delta\mathbf{P}^2)$$
$$= (\mathbf{I}_n - 2\mathbf{P})(-[\Delta, \mathbf{P}])^m(-[\Delta, \mathbf{P}])(\mathbf{I}_n - 2\mathbf{P})$$
$$= (\mathbf{I}_n - 2\mathbf{P})(-[\Delta, \mathbf{P}])^{m+1}(\mathbf{I}_n - 2\mathbf{P}),$$

where (1) and (2) follow from the fact that $\mathbf{P}^2 = \mathbf{P}$.

For the fourth identity, from Eq. (41) with $m = 1$,

$$s[\Delta, \mathbf{P}] = (\mathbf{I}_n - 2\mathbf{P})(-s[\Delta, \mathbf{P}])(\mathbf{I}_n - 2\mathbf{P}),$$

for any $s \in \mathbb{R}$. Therefore

$$\exp(s[\Delta, \mathbf{P}]) = \exp\big((\mathbf{I}_n - 2\mathbf{P})(-s[\Delta, \mathbf{P}])(\mathbf{I}_n - 2\mathbf{P})\big)$$
$$\overset{(1)}{=} (\mathbf{I}_n - 2\mathbf{P})\exp(-s[\Delta, \mathbf{P}])(\mathbf{I}_n - 2\mathbf{P}),$$

where (1) follows from the fact that $(\mathbf{I}_n - 2\mathbf{P})(\mathbf{I}_n - 2\mathbf{P}) = \mathbf{I}_n$.

$\square$

## 23   Proof of Theorem 3.20

*Proof.* We need to show that spaces $(\mathrm{Gr}_{n,p}, \oplus_{gr}, \otimes_{gr})$ satisfy axioms (G1), (G2), (G3), V(1), V(2), V(3), V(4), and V(5) (under certain conditions of the theorem).

### Axiom (G1)

*Proof.* The verification of axiom (G1) follows directly from Eq. (8). It is also easy to see that $\mathbf{I}_{n,p}$ is a right identity of any $\mathbf{P} \in \mathrm{Gr}_{n,p}$. $\square$

**Axiom (G2)**

*Proof.* We first recall a property of the matrix logarithm.

**Lemma 23.1** ([56])**.** *Let $\mathbf{A}$ be a square matrix that has no negative real eigenvalues. Then*

$$\log(\mathbf{A}^{-1}) = -\log(\mathbf{A}).$$

Let $\mathrm{Log}^{gr}_{\mathbf{I}_{n,p}}(\mathbf{P}) = [\Omega, \mathbf{I}_{n,p}]$. Then from Eq. (33) we have

$$
\begin{aligned}
\Omega &= \frac{1}{2}\log\big((\mathbf{I}_n - 2\mathbf{P})(\mathbf{I}_n - 2\mathbf{I}_{n,p})\big) \\
&= \frac{1}{2}\log\big((\mathbf{I}_n - 2\,\mathrm{Exp}^{gr}_{\mathbf{I}_{n,p}}(\mathrm{Log}^{gr}_{\mathbf{I}_{n,p}}(\mathbf{P})))(\mathbf{I}_n - 2\mathbf{I}_{n,p})\big) \\
&\overset{(1)}{=} \frac{1}{2}\log\big((\mathbf{I}_n - 2\exp([\bar{\mathbf{P}}, \mathbf{I}_{n,p}])\mathbf{I}_{n,p}\exp(-[\bar{\mathbf{P}}, \mathbf{I}_{n,p}]))(\mathbf{I}_n - 2\mathbf{I}_{n,p})\big) \\
&= \frac{1}{2}\log\big(\exp([\bar{\mathbf{P}}, \mathbf{I}_{n,p}])(\mathbf{I}_n - 2\mathbf{I}_{n,p})\exp(-[\bar{\mathbf{P}}, \mathbf{I}_{n,p}])(\mathbf{I}_n - 2\mathbf{I}_{n,p})\big) \\
&\overset{(2)}{=} \frac{1}{2}\log\big(\exp(2[\bar{\mathbf{P}}, \mathbf{I}_{n,p}])\big),
\end{aligned}
\tag{42}
$$

where (1) follows from Eq. (32), and (2) follows from the fourth identity of Lemma 3.19.

Since $\mathbf{P}$ and $\mathbf{I}_{n,p}$ are not in each other's cut locus, they can be joined by a unique geodesic. Therefore, there is an implicit condition on $\mathbf{P}$, i.e., matrix $(\mathbf{I}_n - 2\mathbf{P})(\mathbf{I}_n - 2\mathbf{I}_{n,p})$ has no negative real eigenvalues [48]. From the above chain of equations we have

$$(\mathbf{I}_n - 2\mathbf{P})(\mathbf{I}_n - 2\mathbf{I}_{n,p}) = \exp(2[\bar{\mathbf{P}}, \mathbf{I}_{n,p}]),$$

which means that $\exp(2[\bar{\mathbf{P}}, \mathbf{I}_{n,p}])$ has no negative real eigenvalues.

Let $\mathrm{Log}^{gr}_{\mathbf{I}_{n,p}}(\ominus_{gr}\mathbf{P}) = [\Omega', \mathbf{I}_{n,p}]$. Then from Eq. (33) we have

$$
\begin{aligned}
\Omega' &= \frac{1}{2}\log\big((\mathbf{I}_n - 2\ominus_{gr}\mathbf{P})(\mathbf{I}_n - 2\mathbf{I}_{n,p})\big) \\
&= \frac{1}{2}\log\big((\mathbf{I}_n - 2\,\mathrm{Exp}^{gr}_{\mathbf{I}_{n,p}}(-\mathrm{Log}^{gr}_{\mathbf{I}_{n,p}}(\mathbf{P})))(\mathbf{I}_n - 2\mathbf{I}_{n,p})\big) \\
&\overset{(1)}{=} \frac{1}{2}\log\big((\mathbf{I}_n - 2\exp([-\bar{\mathbf{P}}, \mathbf{I}_{n,p}])\mathbf{I}_{n,p}\exp(-[-\bar{\mathbf{P}}, \mathbf{I}_{n,p}]))(\mathbf{I}_n - 2\mathbf{I}_{n,p})\big) \\
&= \frac{1}{2}\log\big((\mathbf{I}_n - 2\exp(-[\bar{\mathbf{P}}, \mathbf{I}_{n,p}])\mathbf{I}_{n,p}\exp([\bar{\mathbf{P}}, \mathbf{I}_{n,p}]))(\mathbf{I}_n - 2\mathbf{I}_{n,p})\big) \\
&= \frac{1}{2}\log\big(\exp(-[\bar{\mathbf{P}}, \mathbf{I}_{n,p}])(\mathbf{I}_n - 2\mathbf{I}_{n,p})\exp([\bar{\mathbf{P}}, \mathbf{I}_{n,p}])(\mathbf{I}_n - 2\mathbf{I}_{n,p})\big) \\
&\overset{(2)}{=} \frac{1}{2}\log\big(\exp(-2[\bar{\mathbf{P}}, \mathbf{I}_{n,p}])\big) \\
&= \frac{1}{2}\log\big(\exp(2[\bar{\mathbf{P}}, \mathbf{I}_{n,p}])^{-1}\big) \\
&\overset{(3)}{=} -\frac{1}{2}\log\big(\exp(2[\bar{\mathbf{P}}, \mathbf{I}_{n,p}])\big).
\end{aligned}
\tag{43}
$$

The derivation of Eq. (43) follows.

(1) follows from Eq. (32),

(2) follows from the fourth identity of Lemma 3.19.

(3) follows from the fact that $\exp(2[\bar{\mathbf{P}}, \mathbf{I}_{n,p}])$ has no negative real eigenvalues and Lemma 23.1.

We thus obtain

$$
\begin{aligned}
\overline{\ominus_{gr}\mathbf{P}} &= \mathrm{Log}_{\mathbf{I}_{n,p}}^{gr}(\ominus_{gr}\mathbf{P}) \\
&= [\Omega', \mathbf{I}_{n,p}] \\
&= [-\Omega, \mathbf{I}_{n,p}] \\
&= -[\Omega, \mathbf{I}_{n,p}] \\
&= -\mathrm{Log}_{\mathbf{I}_{n,p}}^{gr}(\mathbf{P}) \\
&= -\overline{\mathbf{P}}.
\end{aligned}
\tag{44}
$$

Therefore

$$
\begin{aligned}
\ominus_{gr}\mathbf{P} \oplus_{gr} \mathbf{P} &\overset{(1)}{=} \exp([\overline{\ominus_{gr}\mathbf{P}}, \mathbf{I}_{n,p}])\mathbf{P}\exp(-[\overline{\ominus_{gr}\mathbf{P}}, \mathbf{I}_{n,p}]) \\
&\overset{(2)}{=} \exp([-\overline{\mathbf{P}}, \mathbf{I}_{n,p}])\mathbf{P}\exp(-[-\overline{\mathbf{P}}, \mathbf{I}_{n,p}]) \\
&= \exp(-[\overline{\mathbf{P}}, \mathbf{I}_{n,p}])\mathbf{P}\exp([\overline{\mathbf{P}}, \mathbf{I}_{n,p}]) \\
&\overset{(3)}{=} \exp(-[\overline{\mathbf{P}}, \mathbf{I}_{n,p}])\exp([\overline{\mathbf{P}}, \mathbf{I}_{n,p}])\mathbf{I}_{n,p}\exp(-[\overline{\mathbf{P}}, \mathbf{I}_{n,p}])\exp([\overline{\mathbf{P}}, \mathbf{I}_{n,p}]) \\
&= \mathbf{I}_{n,p},
\end{aligned}
$$

where (1), (2), and (3) follow respectively from Eqs. (8), (44), and (36).

*Remark.* It is easy to show that $\ominus_{gr}\mathbf{P}$ is a right inverse of $\mathbf{P}$. Indeed, we have

$$
\begin{aligned}
\mathbf{P} \oplus_{gr} (\ominus_{gr}\mathbf{P}) &\overset{(1)}{=} \exp([\overline{\mathbf{P}}, \mathbf{I}_{n,p}])(\ominus_{gr}\mathbf{P})\exp(-[\overline{\mathbf{P}}, \mathbf{I}_{n,p}]) \\
&= \exp([\overline{\mathbf{P}}, \mathbf{I}_{n,p}])\mathrm{Exp}_{\mathbf{I}_{n,p}}^{gr}(-\mathrm{Log}_{\mathbf{I}_{n,p}}^{gr}(\mathbf{P}))\exp(-[\overline{\mathbf{P}}, \mathbf{I}_{n,p}]) \\
&\overset{(2)}{=} \exp([\overline{\mathbf{P}}, \mathbf{I}_{n,p}])\exp([-\overline{\mathbf{P}}, \mathbf{I}_{n,p}])\mathbf{I}_{n,p}\exp(-[-\overline{\mathbf{P}}, \mathbf{I}_{n,p}])\exp(-[\overline{\mathbf{P}}, \mathbf{I}_{n,p}]) \\
&= \mathbf{I}_{n,p},
\end{aligned}
$$

where (1) and (2) follow respectively from Eqs. (8) and (32).

$\square$

### Axiom (G3)

*Proof.* From Eq. (8),

$$
(\mathbf{P} \oplus_{gr} \mathbf{Q}) \oplus_{gr} \mathrm{gyr}_{gr}[\mathbf{P}, \mathbf{Q}]\mathbf{R} = \exp([\overline{\mathbf{P} \oplus_{gr} \mathbf{Q}}, \mathbf{I}_{n,p}])\,\mathrm{gyr}_{gr}[\mathbf{P}, \mathbf{Q}]\mathbf{R}\exp(-[\overline{\mathbf{P} \oplus_{gr} \mathbf{Q}}, \mathbf{I}_{n,p}]).
\tag{45}
$$

Replacing $\mathrm{gyr}_{gr}[\mathbf{P}, \mathbf{Q}]\mathbf{R}$ in the right-hand side of Eq. (45) with its expression in Eq. (10), we obtain

$$
\begin{aligned}
(\mathbf{P} \oplus_{gr} \mathbf{Q}) \oplus_{gr} \mathrm{gyr}_{gr}[\mathbf{P}, \mathbf{Q}]\mathbf{R} &= \exp([\overline{\mathbf{P}}, \mathbf{I}_{n,p}])\exp([\overline{\mathbf{Q}}, \mathbf{I}_{n,p}])\mathbf{R}\exp(-[\overline{\mathbf{Q}}, \mathbf{I}_{n,p}])\exp(-[\overline{\mathbf{P}}, \mathbf{I}_{n,p}]) \\
&\overset{(1)}{=} \exp([\overline{\mathbf{P}}, \mathbf{I}_{n,p}])(\mathbf{Q} \oplus_{gr} \mathbf{R})\exp(-[\overline{\mathbf{P}}, \mathbf{I}_{n,p}]) \\
&\overset{(2)}{=} \mathbf{P} \oplus_{gr} (\mathbf{Q} \oplus_{gr} \mathbf{R}),
\end{aligned}
$$

where (1) and (2) follow from Eq. (8).

$\square$

### Gyrocommutative Law

*Proof.* First, we need to prove the two following lemmas.

**Lemma 23.2.** *Let* $\mathbf{A} = \exp([\overline{\mathbf{P}}, \mathbf{I}_{n,p}])$ *where* $\mathbf{P} \in \mathrm{Gr}_{n,p}$. *Then* $\mathbf{A}$ *is orthogonal and has the following form:*

$$
\mathbf{A} = \begin{bmatrix} \mathbf{A}_{11} & \mathbf{A}_{12} \\ -\mathbf{A}_{12}^T & \mathbf{A}_{22} \end{bmatrix},
$$

*where* $\mathbf{A}_{11} \in \mathrm{Sym}_p$, $\mathbf{A}_{12} \in \mathrm{M}_{p,n-p}$, *and* $\mathbf{A}_{22} \in \mathrm{Sym}_{n-p}$.

*Proof.* Let $\Omega = [\bar{\mathbf{P}}, \mathbf{I}_{n,p}]$. Since $T_{\mathbf{I}_{n,p}} \operatorname{Gr}_{n,p}$ contains a subset of symmetric matrices [48], $\bar{\mathbf{P}}$ is symmetric, $\Omega$ is skew-symmetric, and thus

$$\exp([\bar{\mathbf{P}}, \mathbf{I}_{n,p}]) = \exp(\Omega)$$

is orthogonal.

To prove the second part, note that $[\bar{\mathbf{P}}, \mathbf{I}_{n,p}]$ has the following form:

$$[\bar{\mathbf{P}}, \mathbf{I}_{n,p}] = \begin{bmatrix} 0 & \mathbf{K} \\ -\mathbf{K}^T & 0 \end{bmatrix},$$

where $\mathbf{K} \in \mathrm{M}_{p,n-p}$. By induction on $n$, it is easy to show that for $n \geq 0$ and $n$ is even, $[\bar{\mathbf{P}}, \mathbf{I}_{n,p}]^n$ and $[\bar{\mathbf{P}}, \mathbf{I}_{n,p}]^{n+1}$ have the following forms

$$[\bar{\mathbf{P}}, \mathbf{I}_{n,p}]^n = \begin{bmatrix} \mathbf{K}_1 & 0 \\ 0 & \mathbf{K}_2 \end{bmatrix}, [\bar{\mathbf{P}}, \mathbf{I}_{n,p}]^{n+1} = \begin{bmatrix} 0 & \mathbf{K}_3 \\ -\mathbf{K}_3^T & 0 \end{bmatrix},$$

where $\mathbf{K}_1 \in \mathrm{Sym}_p$, $\mathbf{K}_2 \in \mathrm{Sym}_{n-p}$, and $\mathbf{K}_3 \in \mathrm{M}_{p,n-p}$. Therefore

$$\exp([\bar{\mathbf{P}}, \mathbf{I}_{n,p}]) = \sum_{k=0}^{\infty} \frac{1}{k!} [\bar{\mathbf{P}}, \mathbf{I}_{n,p}]^k = \begin{bmatrix} \mathbf{A}_{11} & \mathbf{A}_{12} \\ -\mathbf{A}_{12}^T & \mathbf{A}_{22} \end{bmatrix},$$

where $\mathbf{A}_{11} \in \mathrm{Sym}_p$, $\mathbf{A}_{12} \in \mathrm{M}_{p,n-p}$, and $\mathbf{A}_{22} \in \mathrm{Sym}_{n-p}$. $\qquad\square$

**Lemma 23.3.** *Let* $\mathbf{A}, \mathbf{B}, \mathbf{C} \in \mathrm{O}_n$ *such that*

$$\mathbf{A} = \begin{bmatrix} \mathbf{A}_{11} & \mathbf{A}_{12} \\ -\mathbf{A}_{12}^T & \mathbf{A}_{22} \end{bmatrix}, \mathbf{B} = \begin{bmatrix} \mathbf{B}_{11} & \mathbf{B}_{12} \\ -\mathbf{B}_{12}^T & \mathbf{B}_{22} \end{bmatrix}, \mathbf{C} = \begin{bmatrix} \mathbf{C}_{11} & \mathbf{C}_{12} \\ -\mathbf{C}_{12}^T & \mathbf{C}_{22} \end{bmatrix},$$

*where* $\mathbf{A}_{11} \in \mathrm{Sym}_p$, $\mathbf{A}_{12} \in \mathrm{M}_{p,n-p}$, $\mathbf{A}_{22} \in \mathrm{Sym}_{n-p}$, $\mathbf{B}_{11} \in \mathrm{Sym}_p$, $\mathbf{B}_{12} \in \mathrm{M}_{p,n-p}$, $\mathbf{B}_{22} \in \mathrm{Sym}_{n-p}$, $\mathbf{C}_{11} \in \mathrm{Sym}_p$, $\mathbf{C}_{12} \in \mathrm{M}_{p,n-p}$, *and* $\mathbf{C}_{22} \in \mathrm{Sym}_{n-p}$. *Let* $\mathbf{O} \in \mathrm{O}_n$ *such that* $\mathbf{O}\mathbf{I}_{n,p}\mathbf{O}^{-1} = \mathbf{I}_{n,p}$ *and* $\mathbf{C} = \mathbf{ABO}$. *Then*

$$\mathbf{C}^{-1}\mathbf{ABBAI}_{n,p}\mathbf{A}^{-1}\mathbf{B}^{-1}\mathbf{B}^{-1}\mathbf{A}^{-1}\mathbf{C} = \mathbf{CI}_{n,p}\mathbf{C}^{-1}.$$

*Proof.* Let $o_{ij}, i, j = 1, \ldots, n$ be the entry at the $i^{th}$ row and $j^{th}$ column of $\mathbf{O}$. The equality $\mathbf{O}\mathbf{I}_{n,p}\mathbf{O}^{-1} = \mathbf{I}_{n,p}$ implies that $\mathbf{O}\mathbf{I}_{n,p} = \mathbf{I}_{n,p}\mathbf{O}$. We thus have $o_{ij} = o_{ji} = 0, i = p + 1, \ldots, n, j = 1, \ldots, p$. Therefore, $\mathbf{O}$ has the following form:

$$\mathbf{O} = \begin{bmatrix} \mathbf{O}_1 & 0 \\ 0 & \mathbf{O}_2 \end{bmatrix},$$

where $\mathbf{O}_1 \in \mathrm{M}_{p,p}$ and $\mathbf{O}_2 \in \mathrm{M}_{n-p,n-p}$. Some simple computations show that $\mathbf{O}^T\mathbf{B}^T\mathbf{A}^T$ and $\mathbf{O}^T\mathbf{BA}$ have the following forms:

$$\mathbf{O}^T\mathbf{B}^T\mathbf{A}^T = \begin{bmatrix} \mathbf{L}_{11} & \mathbf{L}_{12} \\ \mathbf{L}_{21} & \mathbf{L}_{22} \end{bmatrix},$$

$$\mathbf{O}^T\mathbf{BA} = \begin{bmatrix} \mathbf{L}_{11} & -\mathbf{L}_{12} \\ -\mathbf{L}_{21} & \mathbf{L}_{22} \end{bmatrix},$$

where $\mathbf{L}_{11} \in \mathrm{M}_{p,p}$, $\mathbf{L}_{12} \in \mathrm{M}_{p,n-p}$, $\mathbf{L}_{21} \in \mathrm{M}_{n-p,p}$, and $\mathbf{L}_{22} \in \mathrm{M}_{n-p,n-p}$. Now the equality $\mathbf{C}^T = \mathbf{O}^T\mathbf{B}^T\mathbf{A}^T$ implies that

$$\mathbf{C}_{11} = \mathbf{L}_{11}, -\mathbf{C}_{12} = \mathbf{L}_{12}, \mathbf{C}_{12}^T = \mathbf{L}_{21}, \mathbf{C}_{22} = \mathbf{L}_{22},$$

which leads to $\mathbf{C} = \mathbf{O}^T\mathbf{BA}$, and therefore

$$\mathbf{C}^{-1}\mathbf{ABBAI}_{n,p}\mathbf{A}^{-1}\mathbf{B}^{-1}\mathbf{B}^{-1}\mathbf{A}^{-1}\mathbf{C} = \mathbf{O}^{-1}\mathbf{B}^{-1}\mathbf{A}^{-1}\mathbf{ABBAI}_{n,p}\mathbf{A}^{-1}\mathbf{B}^{-1}\mathbf{B}^{-1}\mathbf{A}^{-1}\mathbf{ABO}$$

$$= \mathbf{O}^{-1}\mathbf{BAI}_{n,p}\mathbf{A}^{-1}\mathbf{B}^{-1}\mathbf{O}$$

$$= \mathbf{CI}_{n,p}\mathbf{C}^{-1}.$$

$\square$

Let $\mathbf{C} = \exp([\overline{\mathbf{P} \oplus_{gr} \mathbf{Q}}, \mathbf{I}_{n,p}])$, $\mathbf{A} = \exp([\overline{\mathbf{P}}, \mathbf{I}_{n,p}])$, $\mathbf{B} = \exp([\overline{\mathbf{Q}}, \mathbf{I}_{n,p}])$. Using Eqs. (10) and (11), we have

$$
\begin{aligned}
\mathrm{gyr}_{gr}[\mathbf{P}, \mathbf{Q}](\mathbf{Q} \oplus_{gr} \mathbf{P}) &= \mathbf{C}^{-1}\mathbf{A}\mathbf{B}(\mathbf{Q} \oplus_{gr} \mathbf{P})\mathbf{B}^{-1}\mathbf{A}^{-1}\mathbf{C} \\
&\stackrel{(1)}{=} \mathbf{C}^{-1}\mathbf{A}\mathbf{B}\exp([\overline{\mathbf{Q}}, \mathbf{I}_{n,p}])\mathbf{P}\exp(-[\overline{\mathbf{Q}}, \mathbf{I}_{n,p}])\mathbf{B}^{-1}\mathbf{A}^{-1}\mathbf{C} \\
&\stackrel{(2)}{=} \mathbf{C}^{-1}\mathbf{A}\mathbf{B}\exp([\overline{\mathbf{Q}}, \mathbf{I}_{n,p}])\exp([\overline{\mathbf{P}}, \mathbf{I}_{n,p}])\mathbf{I}_{n,p}\exp(-[\overline{\mathbf{P}}, \mathbf{I}_{n,p}])\exp(-[\overline{\mathbf{Q}}, \mathbf{I}_{n,p}])\mathbf{B}^{-1}\mathbf{A}^{-1}\mathbf{C} \\
&= \mathbf{C}^{-1}\mathbf{A}\mathbf{B}\mathbf{B}\mathbf{A}\mathbf{I}_{n,p}\mathbf{A}^{-1}\mathbf{B}^{-1}\mathbf{B}^{-1}\mathbf{A}^{-1}\mathbf{C},
\end{aligned}
\tag{46}
$$

where (1) and (2) follow respectively from Eqs. (8) and (36).

Notice that

$$
\begin{aligned}
\exp([\overline{\mathbf{P} \oplus_{gr} \mathbf{Q}}, \mathbf{I}_{n,p}])\mathbf{I}_{n,p}\exp(-[\overline{\mathbf{P} \oplus_{gr} \mathbf{Q}}, \mathbf{I}_{n,p}]) &\stackrel{(1)}{=} \mathbf{P} \oplus_{gr} \mathbf{Q} \\
&\stackrel{(2)}{=} \exp([\overline{\mathbf{P}}, \mathbf{I}_{n,p}])\mathbf{Q}\exp(-[\overline{\mathbf{P}}, \mathbf{I}_{n,p}]) \\
&\stackrel{(3)}{=} \exp([\overline{\mathbf{P}}, \mathbf{I}_{n,p}])\exp([\overline{\mathbf{Q}}, \mathbf{I}_{n,p}])\mathbf{I}_{n,p}\exp(-[\overline{\mathbf{Q}}, \mathbf{I}_{n,p}])\exp(-[\overline{\mathbf{P}}, \mathbf{I}_{n,p}]),
\end{aligned}
\tag{47}
$$

where (1), (2), and (3) follow respectively from Eqs. (36), (8), and (36).

Therefore

$$
\mathbf{C}\mathbf{I}_{n,p}\mathbf{C}^{-1} = \mathbf{A}\mathbf{B}\mathbf{I}_{n,p}\mathbf{B}^{-1}\mathbf{A}^{-1}.
\tag{48}
$$

Let $\mathbf{O} = (\mathbf{A}\mathbf{B})^{-1}\mathbf{C}$. Then $\mathbf{C} = \mathbf{A}\mathbf{B}\mathbf{O}$ and from Eq. (48) we deduce that

$$
\mathbf{A}\mathbf{B}\mathbf{O}\mathbf{I}_{n,p}\mathbf{O}^{-1}\mathbf{B}^{-1}\mathbf{A}^{-1} = \mathbf{A}\mathbf{B}\mathbf{I}_{n,p}\mathbf{B}^{-1}\mathbf{A}^{-1},
$$

which leads to $\mathbf{O}\mathbf{I}_{n,p}\mathbf{O}^{-1} = \mathbf{I}_{n,p}$. Based on Lemma 23.3, we get

$$
\mathbf{C}^{-1}\mathbf{A}\mathbf{B}\mathbf{B}\mathbf{A}\mathbf{I}_{n,p}\mathbf{A}^{-1}\mathbf{B}^{-1}\mathbf{B}^{-1}\mathbf{A}^{-1}\mathbf{C} = \mathbf{C}\mathbf{I}_{n,p}\mathbf{C}^{-1}.
\tag{49}
$$

Combining Eqs. (46), (47), (48) and (49) leads to

$$
\mathrm{gyr}_{gr}[\mathbf{P}, \mathbf{Q}](\mathbf{Q} \oplus_{gr} \mathbf{P}) = \mathbf{P} \oplus_{gr} \mathbf{Q}.
\tag{50}
$$

$\square$

## Nonreductive Gyrogroups

*Proof.* We need to show that groupoids $(\mathrm{Gr}_{n,p}, \oplus_{gr})$ do not satisfy the Left Reduction Property. We will prove this by contradiction. Assuming that for all $\mathbf{P}, \mathbf{Q}$, and $\mathbf{R} \in \mathrm{Gr}_{n,p}$,

$$
\mathrm{gyr}_{gr}[\mathbf{P}, \mathbf{Q}]\mathbf{R} = \mathrm{gyr}_{gr}[\mathbf{P} \oplus_{gr} \mathbf{Q}, \mathbf{Q}]\mathbf{R}.
$$

In particular, for $\mathbf{P} = \mathbf{I}_{n,p}$,

$$
\begin{aligned}
\mathrm{gyr}_{gr}[\mathbf{I}_{n,p}, \mathbf{Q}]\mathbf{R} &= \mathrm{gyr}_{gr}[\mathbf{I}_{n,p} \oplus_{gr} \mathbf{Q}, \mathbf{Q}]\mathbf{R} \\
&= \mathrm{gyr}_{gr}[\mathbf{Q}, \mathbf{Q}]\mathbf{R} \\
&\stackrel{(1)}{=} F_{gr}[\mathbf{Q}, \mathbf{Q}]\mathbf{R}(F_{gr}[\mathbf{Q}, \mathbf{Q}])^{-1},
\end{aligned}
\tag{51}
$$

where (1) follows from Eq. (10).

From Eq. (11),

$$
F_{gr}(\mathbf{I}_{n,p}, \mathbf{Q}) = \exp(-[\overline{\mathbf{I}_{n,p} \oplus_{gr} \mathbf{Q}}, \mathbf{I}_{n,p}])\exp([\overline{\mathbf{I}_{n,p}}, \mathbf{I}_{n,p}])\exp([\overline{\mathbf{Q}}, \mathbf{I}_{n,p}]) = \exp(-[\overline{\mathbf{Q}}, \mathbf{I}_{n,p}])\mathbf{I}_n\exp([\overline{\mathbf{Q}}, \mathbf{I}_{n,p}]) = \mathbf{I}_n.
$$

Hence, from Eq. (10), we get

$$
\mathrm{gyr}_{gr}[\mathbf{I}_{n,p}, \mathbf{Q}]\mathbf{R} = \mathbf{R}.
\tag{52}
$$

Combining Eqs. (51) and (52) leads to

$$F_{gr}[\mathbf{Q}, \mathbf{Q}]\mathbf{R}(F_{gr}[\mathbf{Q}, \mathbf{Q}])^{-1} = \mathbf{R}.$$

Therefore

$$F_{gr}[\mathbf{Q}, \mathbf{Q}]\mathbf{R} = \mathbf{R}F_{gr}[\mathbf{Q}, \mathbf{Q}].$$

In other words, $F_{gr}[\mathbf{Q}, \mathbf{Q}]$ commutes with all matrices in $\mathrm{Gr}_{n,p}$. In particular, it commutes with any diagonal matrix $\mathbf{D}$ with $p$ diagonal entries equal to one and $n - p$ diagonal entries equal to zero. Let $i_1, \ldots, i_p \in \{1, \ldots, n\}$ be the indices of the diagonal entries of $\mathbf{D}$ equal to one. Then some simple computations of $F_{gr}[\mathbf{Q}, \mathbf{Q}]\mathbf{D}$ and $\mathbf{D}F_{gr}[\mathbf{Q}, \mathbf{Q}]$ leads to $f_{ij} = f_{ji} = 0, i \in \{i_1, \ldots, i_p\}, j \in \{1, \ldots, n\} \setminus \{i_1, \ldots, i_p\}$, where $f_{ij}$ is the entry at the $i^{th}$ row and $j^{th}$ column of $F_{gr}[\mathbf{Q}, \mathbf{Q}]$. Since $p < n$, there always exists a set of indices $i_1, \ldots, i_p$ such that $i \in \{i_1, \ldots, i_p\}, j \in \{1, \ldots, n\} \setminus \{i_1, \ldots, i_p\}$ for any given $i, j = 1, \ldots, n$. We can thus conclude that $f_{ij} = f_{ji} = 0$ for any $i, j = 1, \ldots, n, i \neq j$, i.e., $F_{gr}[\mathbf{Q}, \mathbf{Q}]$ is diagonal. However, for $n = 3, k = 2$, and $\mathbf{Q}$ is given by

$$\mathbf{Q} = \begin{bmatrix} \frac{1}{\sqrt{3}} & \frac{1}{\sqrt{2}} \\ \frac{1}{\sqrt{3}} & 0 \\ \frac{1}{\sqrt{3}} & -\frac{1}{\sqrt{2}} \end{bmatrix},$$

$F_{gr}[\mathbf{Q}, \mathbf{Q}]$ is given by

$$F_{gr}[\mathbf{Q}, \mathbf{Q}] = \begin{bmatrix} 0.6 & 0.8 & 0 \\ 0.8 & -0.6 & 0 \\ 0 & 0 & -1 \end{bmatrix},$$

which is not a diagonal matrix.

$\square$

**Axiom (V1)**

*Proof.* By the definition of the scalar multiplication, it is trivial to verify axiom (V1). $\square$

**Axiom (V2)**

*Proof.* We first recall a property of the matrix logarithm.

**Lemma 23.4** ([56]). *Let $\mathbf{A}$ be a square matrix such that $|\operatorname{Im}(\lambda_i)| < \pi$ for every eigenvalue $\lambda_i$ of $\mathbf{A}$.* *Then*

$$\log(\exp(\mathbf{A})) = \mathbf{A}.$$

We have

$$s \otimes_{gr} \mathbf{P} \oplus_{gr} t \otimes_{gr} \mathbf{P} \overset{(1)}{=} \exp([\overline{s \otimes_{gr} \mathbf{P}}, \mathbf{I}_{n,p}])(t \otimes_{gr} \mathbf{P}) \exp(-[\overline{s \otimes_{gr} \mathbf{P}}, \mathbf{I}_{n,p}])$$

$$\overset{(2)}{=} \exp\left(\frac{1}{2} \log\left((\mathbf{I}_n - 2(s \otimes_{gr} \mathbf{P}))(\mathbf{I}_n - 2\mathbf{I}_{n,p})\right)\right)(t \otimes_{gr} \mathbf{P})$$

$$\exp\left(-\frac{1}{2} \log\left((\mathbf{I}_n - 2(s \otimes_{gr} \mathbf{P}))(\mathbf{I}_n - 2\mathbf{I}_{n,p})\right)\right),$$

where (1) and (2) follow respectively from Eqs. (8) and (37).

Note that

$$(\mathbf{I}_n - 2(s \otimes_{gr} \mathbf{P}))(\mathbf{I}_n - 2\mathbf{I}_{n,p}) = (\mathbf{I}_n - 2\operatorname{Exp}_{\mathbf{I}_{n,p}}^{gr}(s\bar{\mathbf{P}}))(\mathbf{I}_n - 2\mathbf{I}_{n,p})$$

$$\overset{(1)}{=} (\mathbf{I}_n - 2\exp([s\bar{\mathbf{P}}, \mathbf{I}_{n,p}])\mathbf{I}_{n,p} \exp(-[s\bar{\mathbf{P}}, \mathbf{I}_{n,p}]))(\mathbf{I}_n - 2\mathbf{I}_{n,p})$$

$$= (\mathbf{I}_n - 2\exp(s[\bar{\mathbf{P}}, \mathbf{I}_{n,p}])\mathbf{I}_{n,p} \exp(-s[\bar{\mathbf{P}}, \mathbf{I}_{n,p}]))(\mathbf{I}_n - 2\mathbf{I}_{n,p})$$

$$= \exp(s[\bar{\mathbf{P}}, \mathbf{I}_{n,p}])(\mathbf{I}_n - 2\mathbf{I}_{n,p}) \exp(-s[\bar{\mathbf{P}}, \mathbf{I}_{n,p}])(\mathbf{I}_n - 2\mathbf{I}_{n,p})$$

$$\overset{(2)}{=} \exp(2s[\bar{\mathbf{P}}, \mathbf{I}_{n,p}]),$$

$$\tag{53}$$

where (1) follows from Eq. (32), and (2) follows from the fourth identity of Lemma 3.19.

Hence

$$s \otimes_{gr} \mathbf{P} \oplus_{gr} t \otimes_{gr} \mathbf{P} = \exp\left(\frac{1}{2}\log\left(\exp(2s[\overline{\mathbf{P}}, \mathbf{I}_{n,p}])\right)\right)(t \otimes_{gr} \mathbf{P})\exp\left(-\frac{1}{2}\log\left(\exp(2s[\overline{\mathbf{P}}, \mathbf{I}_{n,p}])\right)\right).$$

The assumption on $\mathbf{P}$ and $s$ implies that $2s[\overline{\mathbf{P}}, \mathbf{I}_{n,p}]$ satisfies the condition of Lemma 23.4. Thus

$$
\begin{aligned}
s \otimes_{gr} \mathbf{P} \oplus_{gr} t \otimes_{gr} \mathbf{P} &= \exp(s[\overline{\mathbf{P}}, \mathbf{I}_{n,p}])(t \otimes_{gr} \mathbf{P})\exp(-s[\overline{\mathbf{P}}, \mathbf{I}_{n,p}]) \\
&\overset{(1)}{=} \exp(s[\overline{\mathbf{P}}, \mathbf{I}_{n,p}])\exp([t\overline{\mathbf{P}}, \mathbf{I}_{n,p}])\mathbf{I}_{n,p}\exp(-[t\overline{\mathbf{P}}, \mathbf{I}_{n,p}])\exp(-s[\overline{\mathbf{P}}, \mathbf{I}_{n,p}]) \\
&= \exp(s[\overline{\mathbf{P}}, \mathbf{I}_{n,p}])\exp(t[\overline{\mathbf{P}}, \mathbf{I}_{n,p}])\mathbf{I}_{n,p}\exp(-t[\overline{\mathbf{P}}, \mathbf{I}_{n,p}])\exp(-s[\overline{\mathbf{P}}, \mathbf{I}_{n,p}]) \\
&\overset{(2)}{=} \exp((s+t)[\overline{\mathbf{P}}, \mathbf{I}_{n,p}])\mathbf{I}_{n,p}\exp(-(s+t)[\overline{\mathbf{P}}, \mathbf{I}_{n,p}]) \\
&= \exp([(s+t)\overline{\mathbf{P}}, \mathbf{I}_{n,p}])\mathbf{I}_{n,p}\exp(-[(s+t)\overline{\mathbf{P}}, \mathbf{I}_{n,p}]) \\
&\overset{(3)}{=} (s+t) \otimes_{gr} \mathbf{P}.
\end{aligned}
$$

(54)

The derivation of Eq. (54) follows.

(1) follows from Eq. (9).

(2) follows from the fact that $s[\overline{\mathbf{P}}, \mathbf{I}_{n,p}]$ and $t[\overline{\mathbf{P}}, \mathbf{I}_{n,p}]$ commute.

(3) follows from Eq. (9).

$\square$

**Axiom (V3)**

*Proof.* We have

$$
\begin{aligned}
s \otimes_{gr} (t \otimes_{gr} \mathbf{P}) &\overset{(1)}{=} \exp([s\overline{t \otimes_{gr} \mathbf{P}}, \mathbf{I}_{n,p}])\mathbf{I}_{n,p}\exp(-[s\overline{t \otimes_{gr} \mathbf{P}}, \mathbf{I}_{n,p}]) \\
&= \exp(s[\overline{t \otimes_{gr} \mathbf{P}}, \mathbf{I}_{n,p}])\mathbf{I}_{n,p}\exp(-s[\overline{t \otimes_{gr} \mathbf{P}}, \mathbf{I}_{n,p}]) \\
&\overset{(2)}{=} \exp\left(\frac{s}{2}\log\left((\mathbf{I}_n - 2(t \otimes_{gr} \mathbf{P}))(\mathbf{I}_n - 2\mathbf{I}_{n,p})\right)\right)\mathbf{I}_{n,p}\exp\left(-\frac{s}{2}\log\left((\mathbf{I}_n - 2(t \otimes_{gr} \mathbf{P}))(\mathbf{I}_n - 2\mathbf{I}_{n,p})\right)\right) \\
&\overset{(3)}{=} \exp\left(\frac{s}{2}\log\left(\exp(2t[\overline{\mathbf{P}}, \mathbf{I}_{n,p}])\right)\right)\mathbf{I}_{n,p}\exp\left(-\frac{s}{2}\log\left(\exp(2t[\overline{\mathbf{P}}, \mathbf{I}_{n,p}])\right)\right).
\end{aligned}
$$

(55)

The derivation of Eq. (55) follows.

(1) follows from Eq. (9).

(2) follows from Eq. (37).

(3) follows from Eq. (53) where $s$ is replaced with $t$.

The assumption on $\mathbf{P}$ and $t$ implies that $2t[\overline{\mathbf{P}}, \mathbf{I}_{n,p}]$ satisfies the condition of Lemma 23.4. Thus

$$
\begin{aligned}
s \otimes_{gr} (t \otimes_{gr} \mathbf{P}) &= \exp(st[\overline{\mathbf{P}}, \mathbf{I}_{n,p}])\mathbf{I}_{n,p}\exp(-st[\overline{\mathbf{P}}, \mathbf{I}_{n,p}]) \\
&= \exp([st\overline{\mathbf{P}}, \mathbf{I}_{n,p}])\mathbf{I}_{n,p}\exp(-[st\overline{\mathbf{P}}, \mathbf{I}_{n,p}]) \\
&\overset{(1)}{=} (st) \otimes_{gr} \mathbf{P},
\end{aligned}
$$

where (1) follows from Eq. (9).

$\square$

## Axiom (V4)

*Proof.* We first need to prove the following lemma.

**Lemma 23.5.** *Let* $t \in \mathbb{R}, \mathbf{P} \in \mathrm{Gr}_{n,p}$, *and* $\mathbf{O} \in \mathrm{O}_n$ *such that* $\mathbf{OI}_{n,p}\mathbf{O}^T = \mathbf{I}_{n,p}$. *Then*

$$t \otimes_{gr} (\mathbf{OPO}^T) = \mathbf{O}(t \otimes_{gr} \mathbf{P})\mathbf{O}^T.$$

*Proof.* We have

$$
\begin{aligned}
t \otimes_{gr} (\mathbf{OPO}^T) &\overset{(1)}{=} \exp([t\overline{\mathbf{OPO}^T}, \mathbf{I}_{n,p}])\mathbf{I}_{n,p} \exp(-[t\overline{\mathbf{OPO}^T}, \mathbf{I}_{n,p}]) \\
&\overset{(2)}{=} \exp([t\mathbf{O}\overline{\mathbf{P}}\mathbf{O}^T, \mathbf{I}_{n,p}])\mathbf{I}_{n,p} \exp(-[t\mathbf{O}\overline{\mathbf{P}}\mathbf{O}^T, \mathbf{I}_{n,p}]) \\
&= \exp([t\mathbf{O}\overline{\mathbf{P}}\mathbf{O}^T, \mathbf{OI}_{n,p}\mathbf{O}^T])\mathbf{I}_{n,p} \exp(-[t\mathbf{O}\overline{\mathbf{P}}\mathbf{O}^T, \mathbf{OI}_{n,p}\mathbf{O}^T]) \\
&= \exp(\mathbf{O}[t\overline{\mathbf{P}}, \mathbf{I}_{n,p}]\mathbf{O}^T)\mathbf{I}_{n,p} \exp(-\mathbf{O}[t\overline{\mathbf{P}}, \mathbf{I}_{n,p}]\mathbf{O}^T) \qquad (56) \\
&\overset{(3)}{=} \mathbf{O}\exp([t\overline{\mathbf{P}}, \mathbf{I}_{n,p}])\mathbf{O}^T\mathbf{I}_{n,p}\mathbf{O}\exp(-[t\overline{\mathbf{P}}, \mathbf{I}_{n,p}])\mathbf{O}^T \\
&= \mathbf{O}\exp([t\overline{\mathbf{P}}, \mathbf{I}_{n,p}])\mathbf{I}_{n,p}\exp(-[t\overline{\mathbf{P}}, \mathbf{I}_{n,p}])\mathbf{O}^T \\
&\overset{(4)}{=} \mathbf{O}(t \otimes_{gr} \mathbf{P})\mathbf{O}^T.
\end{aligned}
$$

The derivation of Eq. (56) follows.

(1) follows from Eq. (9).

(2) follows from the first identity of Lemma 3.19 and the fact that $\mathbf{OI}_{n,p}\mathbf{O}^T = \mathbf{I}_{n,p}$.

(3) follows from the fact that matrix $\mathbf{O}$ is orthogonal.

(4) follows from Eq. (9).

$\square$

Let $\mathbf{C} = \exp([\overline{\mathbf{P} \oplus_{gr} \mathbf{Q}}, \mathbf{I}_{n,p}])$, $\mathbf{A} = \exp([\overline{\mathbf{P}}, \mathbf{I}_{n,p}])$, $\mathbf{B} = \exp([\overline{\mathbf{Q}}, \mathbf{I}_{n,p}])$. As shown in Eq. (48),

$$\mathbf{CI}_{n,p}\mathbf{C}^T = \mathbf{ABI}_{n,p}\mathbf{B}^T\mathbf{A}^T.$$

Hence

$$\mathbf{C}^T\mathbf{ABI}_{n,p}\mathbf{B}^T\mathbf{A}^T\mathbf{C} = \mathbf{I}_{n,p}.$$

Let $\mathbf{O} = \mathbf{C}^T\mathbf{AB}$, then $\mathbf{OI}_{n,p}\mathbf{O}^T = \mathbf{I}_{n,p}$. Using the result in Lemma 23.5 for $t \in \mathbb{R}$ and $\mathbf{R} \in \mathrm{Gr}_{n,p}$, we get

$$t \otimes_{gr} (\mathbf{C}^T\mathbf{ABRB}^T\mathbf{A}^T\mathbf{C}) = \mathbf{C}^T\mathbf{AB}(t \otimes_{gr} \mathbf{R})\mathbf{B}^T\mathbf{A}^T\mathbf{C}.$$

From Eq. (11),

$$F_{gr}(\mathbf{P}, \mathbf{Q}) = \exp(-[\overline{\mathbf{P} \oplus_{gr} \mathbf{Q}}, \mathbf{I}_{n,p}])\exp([\overline{\mathbf{P}}, \mathbf{I}_{n,p}])\exp([\overline{\mathbf{Q}}, \mathbf{I}_{n,p}]) = \mathbf{C}^T\mathbf{AB}.$$

Therefore

$$t \otimes_{gr} \mathrm{gyr}_{gr}[\mathbf{P}, \mathbf{Q}]\mathbf{R} = \mathrm{gyr}_{gr}[\mathbf{P}, \mathbf{Q}](t \otimes_{gr} \mathbf{R}).$$

$\square$

## Axiom (V5)

*Proof.* From Eq. (11),

$$
\begin{aligned}
F_{gr}(s \otimes_{gr} \mathbf{P}, t \otimes_{gr} \mathbf{P}) &= \exp(-[\overline{s \otimes_{gr} \mathbf{P} \oplus_{gr} t \otimes_{gr} \mathbf{P}}, \mathbf{I}_{n,p}])\exp([\overline{s \otimes_{gr} \mathbf{P}}, \mathbf{I}_{n,p}])\exp([\overline{t \otimes_{gr} \mathbf{P}}, \mathbf{I}_{n,p}]) \\
&\overset{(1)}{=} \exp(-[\overline{(s+t) \otimes_{gr} \mathbf{P}}, \mathbf{I}_{n,p}])\exp([\overline{s \otimes_{gr} \mathbf{P}}, \mathbf{I}_{n,p}])\exp([\overline{t \otimes_{gr} \mathbf{P}}, \mathbf{I}_{n,p}]),
\end{aligned}
$$

where (1) follows from the assumption on $\mathbf{P}$ and $s$, and axiom (V2).

Using similar manipulations in the proof of axiom (V2), and by the assumption on $\mathbf{P}$, $s$, and $t$, we obtain

$$\exp(-[\overline{(s+t) \otimes_{gr} \mathbf{P}}, \mathbf{I}_{n,p}]) = \exp(-(s+t)[\overline{\mathbf{P}}, \mathbf{I}_{n,p}]).$$

$$\exp([\overline{s \otimes_{gr} \mathbf{P}}, \mathbf{I}_{n,p}]) = \exp(s[\overline{\mathbf{P}}, \mathbf{I}_{n,p}]).$$

$$\exp([\overline{t \otimes_{gr} \mathbf{P}}, \mathbf{I}_{n,p}]) = \exp(t[\overline{\mathbf{P}}, \mathbf{I}_{n,p}]).$$

Therefore

$$F_{gr}(s \otimes_{gr} \mathbf{P}, t \otimes_{gr} \mathbf{P}) = \exp(-(s+t)[\overline{\mathbf{P}}, \mathbf{I}_{n,p}]) \exp(s[\overline{\mathbf{P}}, \mathbf{I}_{n,p}]) \exp(t[\overline{\mathbf{P}}, \mathbf{I}_{n,p}])$$
$$= \mathbf{I}_n,$$

which leads to $\mathrm{gyr}_{gr}[s \otimes_{gr} \mathbf{P}, t \otimes_{gr} \mathbf{P}] = \mathrm{Id}$.

$\square$

## 24 Proof of Corollary 3.21

*Proof.* The fact that $\|[\overline{\mathbf{P}}, \mathbf{I}_{n,p}]\| \geq \max_{\lambda_i}\{|\mathrm{Im}(\lambda_i)|\}$ leads to the conclusion of the Corollary. $\square$

## 25 Proof of Corollary 3.22

*Proof.* By assumption that $\mathbf{P}$ and $\mathbf{I}_{n,p}$ are not in each other's cut locus, they can be joined by a unique geodesic. Therefore, there is an implicit condition on $\mathbf{P}$, i.e., matrix $(\mathbf{I}_n - 2\mathbf{P})(\mathbf{I}_n - 2\mathbf{I}_{n,p})$ has no negative real eigenvalues [48]. Thus the eigenvalues of the principle logarithm of $(\mathbf{I}_n - 2\mathbf{P})(\mathbf{I}_n - 2\mathbf{I}_{n,p})$ lie in the strip $\{z : -\pi < \mathrm{Im}(z) < \pi\}$. From Eq. (34),

$$\Omega = [\overline{\mathbf{P}}, \mathbf{I}_{n,p}] = \frac{1}{2}\log\left((\mathbf{I}_n - 2\mathbf{P})(\mathbf{I}_n - 2\mathbf{I}_{n,p})\right). \tag{57}$$

We deduce that

$$|\mathrm{Im}(\lambda_i)| < \frac{\pi}{2}.$$

$\square$

If $|s| \leq 1$, then $|s| < \frac{\pi}{2|\mathrm{Im}(\lambda_i)|}$ for any $\lambda_i$. By Theorem 3.20, we can conclude that gyrocommutative and gyrononreductive gyrogroups $(\mathrm{Gr}_{n,p}, \oplus_{gr})$ with the scalar multiplication $\otimes_{gr}$ satisfy axiom (V2). Similar arguments can be used to verify axioms (V3) and (V5).

$\square$

## 26 Derivation of Our SPD Neural Networks

**Definition 26.1** (**Gyroderivative in AI and LE Gyrovector Spaces**). *Let* $(\mathrm{Sym}_n^+, \oplus, \odot)$ *be a gyrovector space, and* $h : \mathbb{R} \to \mathrm{Sym}_n^+$ *be a map. If the limit*

$$\frac{dh}{dt}(t) = \lim_{\delta t \to 0} \frac{1}{\delta t} \odot (\ominus h(t) \oplus h(t + \delta t))$$

*exists for any* $t \in \mathbb{R}$, *then the map* $h$ *is said to be differentiable on* $\mathbb{R}$, *and the gyroderivative of* $h(t)$ *is* $\frac{dh}{dt}(t)$.

Note that the gyroderivative considered here is different from the derivative used for computing tangent vectors to a curve on manifolds [2], which is a map from a set of smooth real-valued functions to $\mathbb{R}$. From Definition 26.1, we can derive the chain rule in AI and LE gyrovector spaces similar to the Gyro-chain-rule [12] in hyperbolic spaces.

**Lemma 26.2** (**Gyro-chain-rule in AI and LE Gyrovector Spaces**). *Let $g : \mathbb{R} \to \mathbb{R}$ be a differentiable map, and $h : \mathbb{R} \to \mathrm{Sym}_n^+$ be a map with a well-defined gyroderivative in a gyrovector space $(\mathrm{Sym}_n^+, \oplus, \odot)$. If $f := h \circ g$, then we have*

$$\frac{df}{dt}(t) = \frac{dg}{dt}(t) \odot \frac{dh}{dt}(g(t)),$$

*where $\frac{dg}{dt}(t)$ is the ordinary derivative.*

*Proof.* The Lemma can be proved by applying the techniques in [12].

$$\frac{df}{dt}(t) = \lim_{\delta t \to 0} \frac{1}{\delta t} \odot (\ominus f(t) \oplus f(t + \delta t))$$

$$= \lim_{\delta t \to 0} \frac{1}{\delta t} \odot (\ominus h(g(t)) \oplus h(g(t) + \delta t(g'(t) + \mathcal{O}(\delta t)))).$$

Let $l_1 = \frac{g'(t)}{\delta t(g'(t) + \mathcal{O}(\delta t))}$, $l_2 = \frac{\mathcal{O}(\delta t)}{\delta t(g'(t) + \mathcal{O}(\delta t))}$. Then $\frac{1}{\delta t} = l_1 + l_2$ and we have

$$\frac{df}{dt}(t) = \lim_{\delta t \to 0} (l_1 + l_2) \odot (\ominus h(g(t)) \oplus h(g(t) + \delta t(g'(t) + \mathcal{O}(\delta t)))).$$

Let $L_1 = \lim_{\delta t \to 0} l_1 \odot (\ominus h(g(t)) \oplus h(g(t) + \delta t(g'(t) + \mathcal{O}(\delta t))))$, $L_2 = \lim_{\delta t \to 0} l_2 \odot (\ominus h(g(t)) \oplus h(g(t) + \delta t(g'(t) + \mathcal{O}(\delta t))))$. Then we get

$$\frac{df}{dt}(t) \overset{(1)}{=} L_1 \oplus L_2$$

$$= L_1 \oplus 0 \odot (\ominus h(g(t)) \oplus h(g(t) + \delta t(g'(t) + \mathcal{O}(\delta t))))$$

$$\overset{(2)}{=} L_1 \oplus \mathbf{I}_n$$

$$\overset{(3)}{=} L_1,$$

where (1), (2), and (3) follow respectively from axioms (V2), (V1), and (G1).

Let $u = \delta t(g'(t) + \mathcal{O}(\delta t))$. Then we have

$$\frac{df}{dt}(t) = \lim_{u \to 0} \frac{g'(t)}{u} \odot (\ominus h(g(t)) \oplus h(g(t) + u))$$

$$= \lim_{u \to 0} g'(t) \frac{1}{g(t) + u - g(t)} \odot (\ominus h(g(t)) \oplus h(g(t) + u))$$

$$\overset{(1)}{=} \frac{dg}{dt}(t) \odot \frac{dh}{dt}(g(t)),$$

where (1) follows from axiom (V3).

$\square$

We consider a class of models that are invariant to time rescaling. Following [12, 42], we first study time transformations in the continuous-time setting and then translate continuous-time models back to the discrete-time setting. In the following, we use indices $h_t$ for discrete time and brackets $h(t)$ for continuous time. From Definition 26.1, axiom (V3), and the Left Cancellation Law, we have

$$h(t + \delta t) \approx h(t) \oplus \delta t \odot \frac{dh}{dt}(t) \tag{58}$$

for small $\delta t$. Let $T$ be a time variable and $H(T) = h(\alpha T)$, $X(T) = x(\alpha T)$. Using the chain rule in AI and LE gyrovector spaces, we obtain

$$\frac{dH}{dT}(T) = \alpha \odot \frac{dh}{dT}(\alpha T). \tag{59}$$

Let $h(t + 1) = \phi(h(t), x(t))$.[10] Note that Eq. (58) is equivalent to

$$\ominus h(t) \oplus h(t + \delta t) \approx \delta t \odot \frac{dh}{dt}(t).$$

---

[10]We drop the model parameters to simplify notations.

With discretization step $\delta t = 1$, we have

$$\frac{dh}{dT}(\alpha T) = \ominus H(T) \oplus h(\alpha T + 1)$$
$$= \ominus H(T) \oplus \phi(H(T), X(T)).$$

Eq. (59) now becomes

$$\frac{dH}{dT}(T) = \alpha \odot \big( \ominus H(T) \oplus \phi(H(T), X(T)) \big).$$

By renaming $H$ to $h$, $X$ to $x$, and $T$ to $t$, we obtain

$$\frac{dh}{dt}(t) = \alpha \odot \big( \ominus h(t) \oplus \phi(h(t), x(t)) \big),$$

which results in

$$h(t) \oplus \frac{dh}{dt}(t) = h(t) \oplus \alpha \odot \big( \ominus h(t) \oplus \phi(h(t), x(t)) \big).$$

From Eq. (58), we have $h(t + 1) = h(t) \oplus \frac{dh}{dt}(t)$. Then

$$h(t + 1) = h(t) \oplus \alpha \odot \big( \ominus h(t) \oplus \phi(h(t), x(t)) \big). \tag{60}$$

Now setting $\phi(h(t), x(t)) = \varphi^{\otimes_a}(\mathbf{W}_h \otimes_{spd}^v h(t) + \mathbf{W}_x \otimes_{spd}^v x(t))$ and translating Eq. (60) back to discrete-time models, we obtain the following recurrent equations:

$$\mathbf{P}_t = \varphi^{\otimes_a}(\mathbf{W}_h \otimes_{spd}^v \mathbf{H}_{t-1} + \mathbf{W}_x \otimes_{spd}^v \mathbf{X}_t),$$

$$\mathbf{H}_t = \mathbf{H}_{t-1} \oplus \alpha \odot ((\ominus \mathbf{H}_{t-1}) \oplus \mathbf{P}_t),$$

where $\mathbf{X}_t, \mathbf{P}_t, \mathbf{H}_{t-1}, \mathbf{H}_t \in Sym_n^+$, $\mathbf{W}_h, \mathbf{W}_x \in \mathbb{R}^n$, and $\alpha \in \mathbb{R}$ are learnable parameters.

*Remark.* Using the above technique, one can also derive a class of RNNs on nonreductive gyrovector spaces that verify the Left Cancellation Law.