# OpenReview forum: "The Gyro-Structure of Some Matrix Manifolds"
_NeurIPS.cc/2022/Conference — NeurIPS 2022 Accept_

### Official Review · Reviewer_65HJ · 2022-07-06

**Rating:** 7
**Confidence:** 2
**Soundness:** 4 excellent
**Presentation:** 2 fair
**Contribution:** 4 excellent

**Summary:**

This paper studied potential gyrovector space structures for SPD manifolds with AI and LE geometries and the Grassman manifolds. The authors defined gyroautomorphisms for these geometries and then verified that SPD manifolds can form gyrovector spaces with AI and LE geometries. For Grassman manifold, the gyrogroup is not gyroreductive, but it is gyrocommunitative. From these defined gyrovector spaces, the authors derived operations that can be used for neural networks. They empirically validated that such manifold networks can be applied to applications such as multi-time-series data analysis and word embeddings.

**Questions:**

- Contributions (2) and (3), Why AI geometry is left out?
- Definition 3.1, How is r useful?

**Strengths And Weaknesses:**

Strengths:

1. The paper contains some original ideas.
2. The theoretical analyses seem to be well supported by the empirical results provided.

Weaknesses:

- This paper is difficult to read, partly because of the prerequisite concepts, and partly because of the authors' writing style. The authors can help the reader by making sure that new concepts are properly defined and motivated. The information in this paper can be better organised so that the key messages are clear and consistent.

---

> ### Author Response · Authors · 2022-08-02
> **Response Reviewer 65HJ**
>
> Regarding our statement in contributions (2) and (3): As noted in Section 3.1.1, the expressions for the basic operations in AI gyrovector spaces already appear in [1,16,17,22,23,24,25]. Therefore, we consider that our contributions in Section 3.1.1 are less important than those in Sections 3.1.2 and 3.2. It is worth mentioning that in Section 3.1.1 we have made two contributions with respect to [1,16,17,22,23,24,25]. First, we show how one can construct the basic operations from the Riemannian geometry of SPD manifolds (exponential map, logarithmic map, and parallel transport). Second, we uncover a hidden analogy related to the parallel transport between AI gyrovector spaces and Euclidean spaces (see the supplemental material, Section 11). Please also refer to the beginning of our response to Reviewer L1ia for more details on Section 3.1.1.
>
> Results of GyroAI-HAUNet with different settings of $r$ are given in the table below:
>
> | Dataset    | HDM05 | FPHA | NTU60 |
> | ------------- | ---------- | ---------| ----------|
> | $r=0.8$    | 77.08      | 93.56 | 95.14 |
> | $r=1.0$    | 78.14      | 96.00 | 94.72 |
> | $r=1.2$    | 82.52      | 95.64 | 93.68 |
> | $r=2.0$    | 82.93      | 94.61 | 94.32 |
>
> Results show that $r$ can have an important impact on our network performance. For example, on HDM05, the performance gap between the two settings $r=1.0$ and $r=2.0$ is 4.79\%.

---

### Official Review · Reviewer_L1ia · 2022-07-13

**Rating:** 6
**Confidence:** 3
**Soundness:** 3 good
**Presentation:** 1 poor
**Contribution:** 3 good

**Summary:**

This paper investigates the space of SPD matrices and Grassmann manifolds as gyro-vector spaces. The authors showed the basic definitions of these algebraic structures and proved that the SPD manifold satisfies the axioms in two different ways, i.e., the so-called AI and LE geometries.  The authors also showed that Grassmann manifolds satisfy similar properties.

As applications, the authors rebuilt SPD networks using the proposed operations of SPD matrices and showed performance improvements over SPDNets on actions recognization. The authors also use Grassmann manifolds for word embedding, showing marginal improvements over SPD models.

**Questions:**

I am convinced that the authors imported new geometric tools into deep learning, which can be potentially useful. However, it is not clear why they are needed, and how they are used in the applications.

Section 3 introduced the gyro-vector space structures of SPD and Grassmann manifolds. The author gives a list of definitions and properties. It should be better emphasized which of these definitions are new. If these are existing definitions, proper citations are needed in the body of the definitions. Similar to the formal statements: does section 3 merely review existing properties of such spaces or there are new discoveries?

The author vaguely mentioned that "We note that the expressions of the binary operation, scalar multiplication ... have already been found. However, none of them pointed out the relations in Eqs. (1) and (2)". It is not clear what "relations" this sentence refers to.

More importantly, why the gyrovector spaces are important to redefine the structure of such manifolds? What are the motivations from the application perspective? What limitations of existing techniques call for these gyrovector spaces?

Section 4 Applications. There should be details in implementing the modified SPD networks, which are missing. I suggest the authors explain in detail and formally the background problem (classification), the original SPD net, and how the gyro-vector space is plugged into this framework.

**Limitations:**

This work is mostly thoretical. In the applications, the authors proposed variations of existing deep learning techniques for the applications of word embedding and action recognition. The authors' experiments do not advance the state-of-the-art performance scores from the application perspective. The negative societal impact is therefore limited by these existing techniques.

**Strengths And Weaknesses:**

Pro:
- new geometric tools for deep learning

Con:
- the writing lacks motivation and clarity. (see the questions section below)
- some implementation details are missing in section 4

---

> ### Author Response · Authors · 2022-08-02
> **Response Reviewer L1ia**
>
> We appreciate the observation about the presentation in Section 3. This helps us to better clarify which parts of the paper are new with respect to the literature.
> * Eqs. (1) and (2) generalize the work of [11] to the matrix manifold setting. We will add this information when we introduce the method in Section 3.
> * Section 3.1.1: The expressions in Eqs. (3) and (4) already appear in [1,16,17]. However, the authors of [1,16,17] consider a different set of axioms to study structures referred to as generalized gyrovector spaces. A generalized gyrovector space is not necessarily a gyrovector space as defined in our work, and vice versa. Eq. (5) already appears in the Kim’s works [22,23,24,25] but only for 2 particular values of $r$, i.e., $r=1$ and $r=2$. Kim also considers a set of axioms for gyrovector spaces that are more loose than the ones considered in our work (see Section 2.1). For the above reasons, we think it would be suitable to refer to [1,16,17,22,23,24,25] without citing them in the definitions and properties.
> * Sections 3.1.2 and 3.2: The definitions and properties are new as far as we know.
>
> Regarding our statement about the relations in Eqs. (1) and (2), we are referring to the connection between the basic operations and the Riemannian geometry of the considered manifold (exponential map, logarithmic map, and parallel transport).
>
> We approach our problem with the gyrovector space formalism for the following reason: We seek generalizations of basic operations on vector spaces, e.g., vector addition and subtraction in order to generalize Euclidean neural networks to SPD and Grassmann manifolds. Furthermore, we aim to uncover hidden analogies that these manifolds share with Euclidean spaces in the same way that one uncovers hidden analogies that hyperbolic spaces share with Euclidean spaces. Such analogies have been shown to be the key ingredients for successfully generalizing Euclidean neural networks to the hyperbolic setting [11]. The theory of gyrovector spaces provides an elegant framework to achieve our goals. Gyrovector spaces are algebraic structures that provide a natural generalization of vector spaces. The theory of gyrovector spaces is also armed with powerful tools and techniques for revealing analogies between hyperbolic and Euclidean spaces. By studying the structures of the considered manifolds under the framework of gyrovector spaces, we aim to adapt such tools and techniques for use in SPD and Grassmann geometries.
>
> From the application perspective, our work addresses some limitations of existing works:
> * SPD manifold learning: Most existing works [5,7,19] do not define the basic operations of gyrovector spaces on SPD manifolds. As a result, it is not trivial for them to generalize models that require arithmetic operations on Euclidean spaces (such as vector addition and subtraction), like the MuRE model [44] to the SPD manifold setting. The work of [27] develops SPD models based on the AI geometry and does not investigate the gyro-structure of SPD manifolds with the LE geometry. The LE metric often leads to much simpler and faster computations compared to the AI metric [3]. Also, it is not clear how a discriminative model built from the work of [27] performs in human action recognition applications.
> * Grassmann manifold learning: To the best of our knowledge, our work is the first to construct a binary operation and a scalar multiplication on Grassmann manifolds by using the theory of gyrovector spaces. This allows us to generalize methods mentioned above to the Grassmann manifold setting. In contrast, it is not trivial for existing works to develop such generalizations. Our approach could also avoid a training issue of the discriminative Grassmann network in [21]. For feature transformations, this network relies on the FRmap layer, followed by the ReOrth layer to preserve the properties of the input matrix of the FRmap layer (orthonormal matrix). During the backward pass through the ReOrth layer, backprop requires the inverse of upper-triangular matrices. In practice, these matrices are often ill-conditioned, making the network impossible to train. Building a discriminative Grassmann network from the projector perspective [46] with our proposed operations as feature transformations could effectively address that training issue.
>
> Concerning the implementation details and problem formulations in Section 4: For human action recognition applications, we propose to add some implementation details from Section 6.1 of the supplemental material. We will add the formulation of the classification problem. For question answering applications, we propose to add the problem formulation from Section 7.1 of the supplemental material. We will also add the implementation details of our Grassmann model. We would like to mention that our SPD model for human action recognition is new and its architecture is not based on that of SPDNet from the work of Huang and Gool [19].

---

### Official Review · Reviewer_DGz9 · 2022-07-15

**Rating:** 7
**Confidence:** 3
**Soundness:** 3 good
**Presentation:** 4 excellent
**Contribution:** 3 good

**Summary:**

The paper studies the gyro-structure of several matrix manifolds. In the process, the authors show that the affine-invariant (AI) and log-Euclidean (LE) geometries of SPD manifolds are closely related to hyperbolic geometries, and Grassmann manifolds share similar properties with gyrovector spaces. Several interesting applications based on these geometries are presented and compared against other models.

**Questions:**

I have developed some uncertainty about the magnitude of significance of such works. The gyrovector operations seem to be basically standard geometric operations that have been understood for quite a while. The packaging of such standard operations in the gyrovector formalism seems reasonably straightforward, albeit necessary in certain contexts. There does not appear to be much particularly profound new analysis or geometry which makes this translation into the gyro-structure language possible. This may be a misunderstanding on my part which may only be clarified by a thorough reading of the supplementary materials. Unfortunately, given the volume of such supplementary material and the reviewing demands of the conference, it is often not possible to perform such a detailed reading in the available time.

If the authors could provide a response to dispel my impression noted above, I would be grateful. They may want to emphasise it in the main body of the paper as well.

**Limitations:**

These need to be discussed clearly in the main paper and not just as supplementary material.

**Strengths And Weaknesses:**

The paper is well-structured and is written in a highly readable and professional style. It builds on several recent works on the topic effectively.  The applications and associated experimental results are interesting and appear convincing.

---

> ### Author Response · Authors · 2022-08-02
> **Response Reviewer DGz9**
>
> The theory of gyrovector spaces has been developed in a series of Ungar’s works [38,39,40] for quite some time. These works have shown that gyrovector spaces algebraically regulate many typical examples of analytic hyperbolic geometry. While it is clear from the Ungar’s works that gyrovector spaces provide the setting for hyperbolic geometry in the same way that vector spaces provide the setting for Euclidean geometry, many questions remain open: Which hidden analogies that matrix manifolds, e.g., SPD and Grassmann manifolds share with hyperbolic and Euclidean spaces ? Can one adapt tools and techniques from the theory of gyrovector spaces for revealing such analogies ? How should one generalize Euclidean neural networks to the matrix manifold setting from such analogies ? We would like to find the answers to these questions and we hope this work contributes towards our goals. The contributions of this work are detailed below:
>
> * SPD manifold learning: We define a binary operation and a scalar multiplication on SPD manifolds with the LE geometry, and verify the gyro-structure of such manifolds, when equipped with the proposed operations. We show that the algebraic definition of parallel transport in a gyrovector space [38] agrees with the classical parallel transport of differential geometry on SPD manifolds with the AI and LE geometries (see the supplemental material, Section 11). We also generalize the matrix scaling and pointwise nonlinearity to the SPD manifold setting, and develop a new class of discriminative SPD neural networks. For human activity recognition applications, our method yields significant improvement over state-of-the-art SPD neural networks. Namely, our best network outperforms the best state-of-the-art SPD network by 1.89\%, 4.17\%, and 15.2\% on HDM05, FPHA, and NTU60 datasets, respectively. Furthermore, our networks have smaller size than state-of-the-art SPD networks.
> * Grassmann manifold learning: To the best of our knowledge, our work is the first that studies the structure of Grassmann manifolds under the framework of gyrovector spaces. We define a binary operation and a scalar multiplication on Grassmann manifolds represented from the projector perspective [46]. Under certain conditions, we show that these manifolds, when equipped with the proposed operations, form nonreductive gyrovector spaces. Such spaces share remarkable analogies with gyrovector spaces. The analogies work well that Grassmann manifolds verify several important properties of gyrovector spaces, e.g., the Left Cancellation Law and Left Gyrotranslation Law [40]. One could then consider adapting powerful tools and techniques [40] for translating classical results in Euclidean geometry into novel results in Grassmann geometry. Thus, our work could open a new direction for generalizing some traditional machine learning models to those on Grassmann manifolds. We also generalize the matrix scaling to the Grassmann manifold setting, and showcase our approach on the task of question answering. In almost all cases, our Grassmann model outperforms the SPD models of [27] in terms of mean accuracy on TreeQA dataset, sometimes by a large margin, while being much more computationally efficient and requiring smaller numbers of DOF. The representative work [21] proposes a discriminative Grassmann neural network from the ONB perspective [46]. Although our work does not show how to build discriminative Grassmann neural networks, it provides the basic ingredients to develop such networks from the projector perspective [46]. Therefore, our study and further extensions could enable a comparison of the two perspectives in the context of deep learning on Grassmann manifolds.
> * We empirically show the benefits of embeddings in product manifolds $\operatorname{Gr}_{n,p} \times \operatorname{Sym}_m^+$ in downstream tasks, i.e., question answering and knowledge graph completion. For question answering, in almost all cases, our best model that learns embeddings in product manifolds outperforms the SPD models of [27] in terms of mean accuracy and standard deviation. For knowledge graph completion, our model that learns embeddings in product manifolds is competitive to the best SPD model, while being much more computationally efficient.
>
> Regarding a discussion about the limitations of our work, we propose to move the discussion from Section 9 of the supplemental material to the main paper.

---

### Meta-Review · Area_Chair_HoLH · 2022-08-28

**Recommendation:** Accept
**Confidence:** Certain

**Metareview:**

The paper studies the gyrovector space structure of a few matrix manifolds. This is of broader interest to practitioners who may be interested in using geometric tools in applications. As the reviewers mention that although interesting, the analysis is per se straightforward for many manifolds. Having said that, this AC believes it is a worthwhile effort to compile such results in one place and show the practical benefits. This paper is a good attempt in this direction. In the final version, please include all the suggestions both in explanations and as well as the new results presented in the discussion period.


**Award:**

No

---

### Decision · Program_Chairs · 2022-09-14

Accept